



# SO₂ and BrO emissions of Masaya volcano from 2014–2020

Florian Dinger[1,2], Timo Kleinbek[2], Steffen Dörner[1], Nicole Bobrowski[1,2], Ulrich Platt[1,2],
Thomas Wagner[1,2], Martha Ibarra[3], and Eveling Espinoza[3]

[1]Max Planck Institut for Chemistry, Mainz, Germany
[2]Institute for Environmental Physics, University of Heidelberg, Germany
[3]Instituto Nicaragüense de Estudios Territoriales, Managua, Nicaragua

**Correspondence:** Florian Dinger (florian.dinger@mpic.de)

**Abstract.** Masaya volcano (Nicaragua, 12.0°N, 86.2°W, 635 m a.s.l.) is one of the few volcanoes hosting a lava lake, today. We present continuous time series of $SO_2$ emission fluxes and $BrO/SO_2$ molar ratios in the gas plume of Masaya from March 2014 to March 2020. This study has two foci: (1) discussing the state of the art of long-term $SO_2$ emission flux monitoring on the example of Masaya and (2) the provision and discussion of a continuous dataset on volcanic gas data unique in its temporal

coverage, which poses a major extension of the empirical data base for studies on the volcanologic as well as atmospheric bromine chemistry.

Our $SO_2$ emission flux retrieval is based on a comprehensive investigation of various aspects of the spectroscopic retrievals, the wind conditions, and the plume height. Our retrieved $SO_2$ emission fluxes are on average a factor of 1.4 larger than former estimates based on the same data. We furthermore observed a correlation between the $SO_2$ emission fluxes and the wind speed

when several of our retrieval extensions are not applied. We make plausible that such a correlation is not expected and present a partial correction of this artefact via applying dynamic estimates for plume height as a function of the wind speed (resulting in a vanishing correlation for wind speeds larger than 10 m/s).

Our empirical data set covers the three time periods (1) before the lava lake elevation, (2) period of high lava lake activity (December 2015 − May 2018), (3) after the period of high lava lake activity. For these three time periods, we report average

$SO_2$ emission fluxes of $1000 \pm 200\,\mathrm{t\,d^{-1}}$, $1000 \pm 300\,\mathrm{t\,d^{-1}}$, and $700 \pm 200\,\mathrm{t\,d^{-1}}$ and average $BrO/SO_2$ molar ratios of $(2.9 \pm 1.5) \cdot 10^{-5}$, $(4.8 \pm 1.9) \cdot 10^{-5}$, and $(5.5 \pm 2.6) \cdot 10^{-5}$. These variations indicate that the two gas proxies provide complementary information: the $BrO/SO_2$ molar ratios were susceptible in particular for the transition between the two former periods while the $SO_2$ emission fluxes were in particular susceptible for the transition between the two latter time periods.

We observed an extremely significant annual cyclicity for the $BrO/SO_2$ molar ratios (amplitudes between $1.4–2.6 \cdot 10^{-5}$) with

a weak semi-annual modulation. We suggest that this cyclicity might be a manifestation of meteorological cycles. We found an anti-correlation between the $BrO/SO_2$ molar ratios and the atmospheric water concentration (correlation coefficient of $-47\%$) but in contrast to that neither a correlation with the ozone mixing ratio ($+21\%$) nor systematic dependencies between the $BrO/SO_2$ molar ratios and the atmospheric plume age for an age range of 2–20 min after the release from the volcanic edifice. The two latter observations indicate an early stop of the autocatalytic partial transformation of bromide $Br^-$ solved in aerosol

particles to atmospheric BrO.

Further patterns in the $BrO/SO_2$ time series were (1) a step increase by $0.7 \cdot 10^{-5}$ in late 2015, (2) a linear trend of $1.2 \cdot 10^{-5}$

per year from December 2015 to March 2018, and (3) a linear trend of $-0.8 \cdot 10^{-5}$ per year from June 2018 to March 2020. The step increase in 2015 coincided with the elevation of the lava lake and was thus most likely caused by a change in the magmatic system. The linear trend between late 2015 and early 2018 may indicate the evolution of the magmatic gas phase

during the ascent of juvenile gas-rich magma whereas the linear trend from June 2018 on may indicate a decreasing bromine abundance in the magma.

## 1 Introduction

Volcanic gas emissions consist predominantly of water ($H_2O$), followed in abundance by carbon dioxide ($CO_2$) and sulphur dioxide ($SO_2$), as well as by a large number of trace gases such as halogen compounds (Giggenbach, 1996; Aiuppa, 2009;

Oppenheimer et al., 2014; Bobrowski and Platt, 2015).

Monitoring magnitude or chemical composition of volcanic gas emissions can help to forecast volcanic eruptions (Carroll and Holloway, 1994; Oppenheimer et al., 2014). $SO_2$ emission fluxes, carbon to sulphur ratios, and halogen to sulphur ratios turned out to be powerful tools enabling the detection of events of magma influx at depth, and respectively the arrival of magma in shallow zones of the magmatic system (e.g. Edmonds et al., 2001; Métrich et al., 2004; Allard et al., 2005; Aiuppa et al., 2005;

Burton et al., 2007; Bobrowski and Giuffrida, 2012).

Monitoring of volcanic gas emissions furthermore allows a quantification of the global volcanic volatile emission fluxes (e.g. Carn et al., 2017; Fischer et al., 2019), is thus an important tool for the validation of satellite data (e.g. Theys et al., 2019), provides empirical data on the impact of volcanoes on the chemistry in the local atmosphere (e.g. Bobrowski and Platt, 2015), and is one of the rare possibilities to learn something about the interior of the Earth (e.g. Oppenheimer et al., 2014,

https://deepcarbon.net/).

The magnitude of volcanic gas emissions can be determined by passive remote sensing techniques such as Differential Optical Absorption Spectroscopy (DOAS) (Platt et al., 1980; Platt and Stutz, 2008; Kern, 2009) which allow the recording of semi-continuous (only during daytime) long-term time series (e.g. Galle et al., 2010). In particular, $SO_2$ emission fluxes are considered to be relatively easy to obtain because of the high spectroscopic selectivity for $SO_2$, the typically negligible atmo-

spheric $SO_2$ background, and a typical atmospheric lifetime of $SO_2$ of at least 1 day. The accuracy of the $SO_2$ emission fluxes depends, however, strongly on the accuracy of the available information on the wind conditions and the altitude of the volcanic plume as well as on the radiative transport conditions. The emission fluxes of other volcanic gas species are usually retrieved by scaling the $SO_2$ emission fluxes with the abundance of these species relative to $SO_2$.

The chemical composition of volcanic gas plumes can be determined for many different gas species by in-situ sampling and

subsequent sample analysis in the laboratory. More recently, automatised in-situ "Multi-Gas" sensors are installed in the field, which measure and transmit the concentration of volcanic gases in the ambient atmosphere with an hourly to daily resolution (e.g. Shinohara, 2005; Aiuppa et al., 2006; Roberts et al., 2017). Chemical composition data retrieved by in-situ methods may, however, not be representative for the bulk gas emissions. Furthermore, in-situ methods are rather labour-intensive and dangerous for the scientist and the instruments due to their vulnerability to destruction by a volcanic explosion and the permanent





contact with poisonous and corrosive volcanic gases.

A retrieval of variations in the chemical composition directly via remote sensing overcomes these limitations. For the remote sensing of a molar ratio at least one additional gas species besides $SO_2$ is required. The desired candidates are $H_2O$ or $CO_2$, however, it has not yet been possible to retrieve their volcanic contributions by remote sensing routinely due to their rather high atmospheric backgrounds—although some recent developments succeeded for special conditions (La Spina et al., 2013;

Kern et al., 2017; Butz et al., 2017; Queisser et al., 2017). Other obvious candidates are chlorine and fluorine compounds. Remote sensing techniques allow a retrieval of hydrogen chloride (HCl) and hydrogen fluorine (HF) via Fourier Transform InfraRed (FTIR) spectroscopy (e.g. Mori and Notsu, 1997; Mori et al., 2002), and chlorine dioxide (OClO) via UV-DOAS (e.g. Bobrowski et al., 2007; Donovan et al., 2014; Gliß et al., 2015; Kern and Lyons, 2018). FTIR systems, however, require stronger light sources than passive DOAS (i.e. usually diffuse solar radiation is not sufficient) and are significantly more ex-

pensive, which is the reason why no continuous monitoring of chlorine or fluorine species has been established, except for the remote-controlled FTIR scanner systems installed at Stromboli volcano since 2009 and at Popocatepetl volcano since 2012 (La Spina et al., 2013; Taquet et al., 2019).

A further emitted halogen — but with a much lower abundance — is hydrogen bromine (HBr). HBr is rapidly converted in the atmosphere by photochemistry to several bromine species by the so called bromine explosion process (Platt and Lehrer, 1997;

Wennberg, 1999; von Glasow, 2010). One of these secondary species is bromine monoxide (BrO), which can be retrieved from the same UV-spectra used for the retrieval of the $SO_2$ emission fluxes. $BrO/SO_2$ time series are thus in principle available or retrievable for all volcanoes which are monitored for $SO_2$ emission fluxes by UV-spectrometers. In consequence, although BrO is not on the list of the most desired plume constituent species, time series of the $BrO/SO_2$ molar ratios in volcanic gas plumes are the best accessible gas proxy for volcanic processes so far (besides the $SO_2$ emission fluxes).

The volcanological interpretation of $BrO/SO_2$ molar ratios is yet difficult and much work is still required in order to use them as a fully reliable proxy for volcanic activity variations. The challenge is the interplay of the manifold physical and chemical behaviour of bromine causing the relative Br/S abundance ratio to be significantly altered by virtually any involved compartment of the volcanic system. The two major sources of uncertainty are the bromine partitioning between the magmatic melt phase and the magmatic gas phase and the bromine chemistry in the volcanic gas plume. With respect to the latter, a robust

understanding of the quantitative link between the emitted HBr and the observed BrO is crucial in order to quantify the total volcanic bromine emissions. This link has been studied by empirical observations (Oppenheimer et al., 2006; Bobrowski and Giuffrida, 2012; Gliß et al., 2015; Roberts, 2018; Rüdiger et al., 2020), theoretical models and simulations (Bobrowski et al., 2007; Roberts et al., 2009, 2014; Roberts, 2018; von Glasow, 2010), and lab experiments (Rüdiger et al., 2018). Gutmann et al. (2018) summarised the current state of the art in their review article. The HBr $\rightleftharpoons$ BrO conversion rate and the stationary

equilibrium HBr/BrO ratio may depend on the chemical plume composition and on the atmospheric conditions such as the solar irradiance, the absolute/relative humidity, the tropospheric background ozone level, and the in-mixing rate of air in the volcanic plume. Based on empirical observations, the equilibrium of the HBr $\rightleftharpoons$ BrO conversion is typically reached within the first 2–10 min after the release of HBr to the atmosphere and remains constant for the next at least 30 min (Bobrowski and Giuffrida, 2012; Lübcke et al., 2014; Platt and Bobrowski, 2015; Gliß et al., 2015). Model simulations have proposed relative





BrO equilibrium fractions between BrO/Br$_{total}$ = 10–50% (von Glasow, 2010; Roberts et al., 2014).

Masaya volcano (12.0°N, 86.2°W, 635 m a.s.l.) is located on the Nicaraguan portion of the Central American Volcanic Arc. Its volcanic complex consists of an older shield volcano now hosting a 6 km x 11 km caldera created by three highly explosive basaltic eruptions during the last 6,000 years: a VEI6 eruption at ∼ 6 ka, a VEI5 eruption at 2.1 ka, and a VEI5 eruption at ∼ 1.8 ka. (VEI: volcanic explosivity index, Williams, 1983; Pérez et al., 2009, Smithsonian Institution). There is a nearly

continuous historic record of its activity since the arrival of the Spanish conquistadors in 1524 (de Oviedo, 1855; McBirney, 1956; Rymer et al., 1998). The Smithonian Institution lists two major eruptions which occurred in 1670 (VEI3) and 1772 (VEI2) and 28 eruptions (mainly VEI1 and some VEI2) since 1852. Masaya's currently active Santiago pit crater formed in 1858/1859 and hosted since occasionally incandescence vents and lava lakes usually lasting several years (McBirney, 1956; Rymer et al., 1998). Masaya's most recent lava lake cycle started in late 2015 when its lava lake elevated to shallower levels

(Aiuppa et al., 2018) and stopped in May 2018 when Masaya's thermal activity decreased back to relatively low levels (Smithsonian Institution, 2018). Masaya is one of the strongest degassing volcanoes in the Central American Volcanic Arc (Martin et al., 2010; Aiuppa et al., 2014, 2018). The volcanic gas plume often hovers close to the ground causing serious issues to the local agriculture and health conditions of the local population (Baxter et al., 1982; Delmelle et al., 2002; van Manen, 2014).

This manuscript reports and discusses time series of the SO$_2$ emission fluxes and BrO/SO$_2$ molar ratios in the gas plume of

Masaya volcano from 2014–2020. We present a comprehensive investigation of frequently ignored topics in the SO$_2$ emission flux retrieval and propose a set of technical extensions which aim to reduce the impact of these systematics. Next, we discuss the impact of the meteorology on the bromine chemistry in the volcanic gas plume. Finally, we compare the retrieved gas data with the general volcanic changes from 2014–2020.

## 2   Measurement side and meteorology

The SO$_2$ and BrO emissions of Masaya are monitored by the Instituto Nicaragüense de Estudios Territoriales (INETER) which is part of the Network for Observation of Volcanic and Atmospheric Change (NOVAC, Galle et al., 2010). The NOVAC instruments are automated remote-sensing UV-spectrometers whose design is simplistic in order to reduce their power consumption, costs, and maintenance (e.g. the spectrometers are not actively temperature-controlled). First NOVAC measurements were conducted at Masaya from April–June 2007 and from September 2008 to February 2009 (these data are not presented

in this manuscript but listed for completeness in Table 4). From March 2014 to March 2020 (end of this study), the NOVAC station "Caracol" (instrument: D2J2124_0, see Figure 1 and Table 1) operated continuously, except for two data gaps of several months. From March–October 2014, a second NOVAC station "Nancital" (D2J2375_0) operated quite close to Caracol (Figure 1).

No direct measurements of the meteorological conditions in the volcanic gas plume were available (except for the plume

heights and the wind directions from March–October 2014 retrieved from the NOVAC data via a triangulation, see next section). Therefore, we accessed the meteorological conditions at Masaya as a best guess by European Centre for Medium-Range Weather Forecasts (ECMWF) ERA-Interim model data for an altitude of 700 m a.s.l. (Figure 3), by operational ECMWF re-





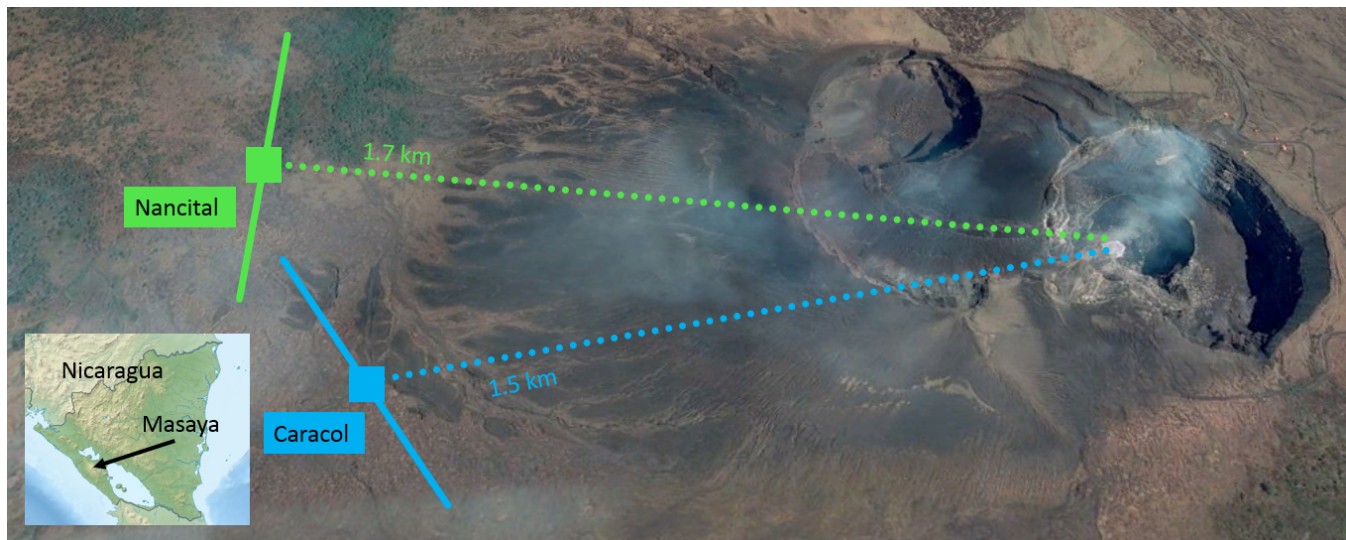

**Figure 1.** Location and scan geometries of the NOVAC stations Caracol and Nancital at Masaya volcano. Coloured squares indicate the location and solid lines the scan direction. For further parameters see Table 1. The map was created with graphical material from https://commons.wikimedia.org/wiki/File:Nicaragua_relief_location_map.jpg and from © Google Earth.

analysis data for an altitude of 700 m a.s.l. (Figure 4), and by ground-based data from Managua airport, which is located 15 km north of Masaya (Figure B3, data from https://mesonet.agron.iastate.edu).

**Table 1.** Spatial set-up of the NOVAC stations at Masaya: altitude $A$, horizontal distance $D$ to and angular orientation $\sigma$ to the volcanic edifice, orientation of the scan plane $\beta$ (see Figures 1 and 2).

| station | coordinates | $A$ | $D$ | $\sigma$ | $\beta$ |
|---------|-------------|-----|-----|----------|---------|
| Caracol | 11.98, -86.18 | 382 m a.s.l. | 1.5 km | 75° | 54° |
| Nancital | 11.99, -86.18 | 340 m a.s.l. | 1.7 km | 95° | 100° |

**ECMWF ERA-Interim data**

The ECMWF ERA-Interim dataset has an original spatial resolution of about 0.7°x0.7° (at and close to the equator), a temporal resolution of 6 h (0:00, 6:00, 12:00, and 18:00 UTC), and 60 hybrid pressure layers which follow the terrain close to ground and in the lower atmosphere and are constant in the higher atmosphere. They reach up to 10 Pa, i.e. about 66 km. The presented ECMWF data are vertically interpolated to an altitude of 700 m a.s.l. and horizontally gridded on a 1°x1° grid (i.e. about 110km 135 x 110km at 12°N). The preparation of the ERA-Interim data on such a grid has been chosen for in general better compatibility with other global data. For local studies using the original 0.7°x0.7° data may be more appropriate though this distinction





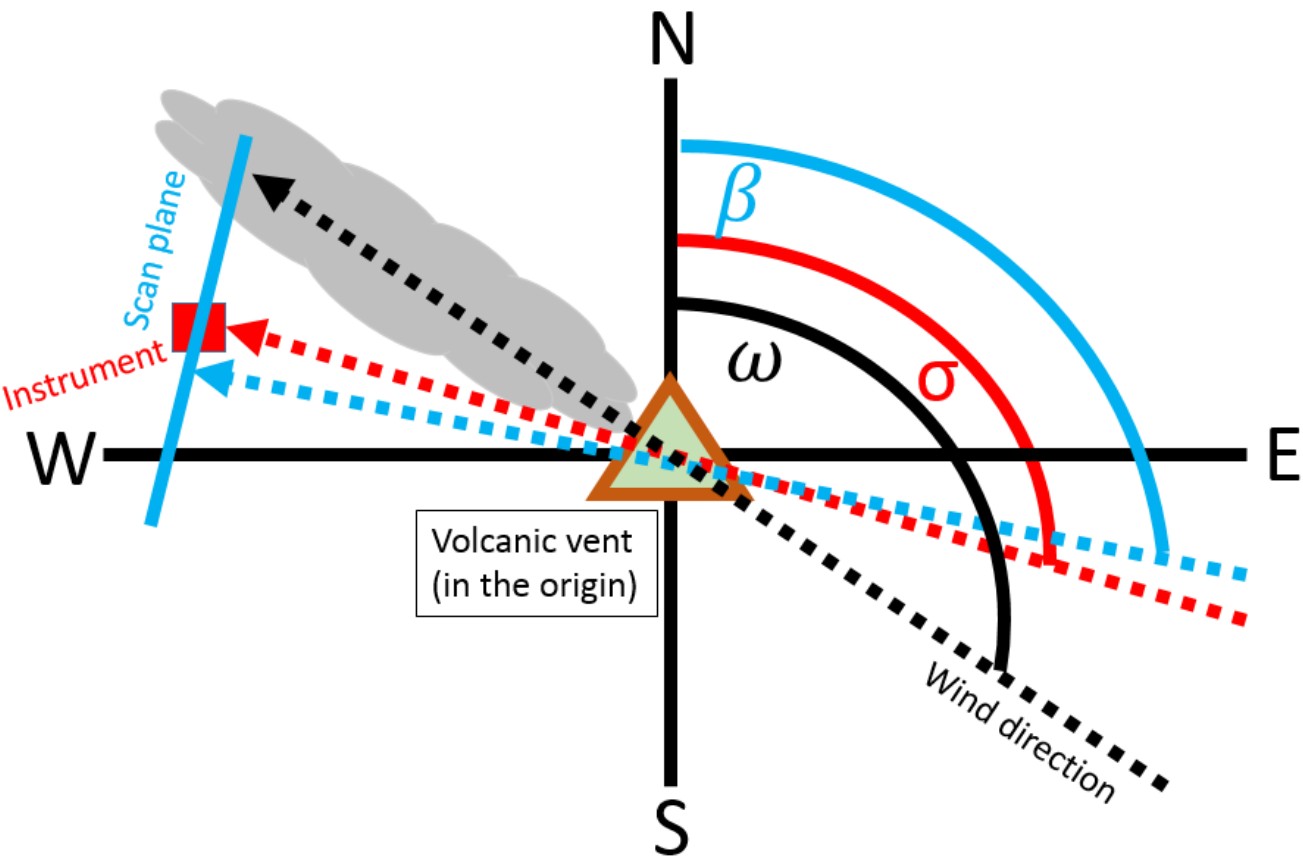

**Figure 2.** Sketch of the geometric relations which are used to calculate the $SO_2$ emission fluxes, to conduct the plume centre triangulation, and to estimate the plume age.

becomes mostly obsolete due to our local calibration approach (see below).

The following meteorological parameters are presented and discussed in this study: (1) the wind speed and (2) the wind direction in order to reconstruct the plume propagation, (3) the barometric pressure, (4) the ambient air temperature, (5) the water vapour concentration, (6) the relative humidity, and (7) the ozone mixing ratio in order to investigate their possible influence on the plume chemistry, and (8) the total cloud cover (i.e. the fraction of the ground pixel area hidden from direct solar irradiation by visible clouds anywhere between ground and the top of the model domain) as a proxy for the radiative conditions.

The following quantitative discussion focuses on the two-weekly moving average of ECMWF data around noon time, though a discussion of the unfiltered time series comes to similar results (see red and blue lines in Figure 3). The time series indicate for all analysed parameters annual cycles, however, with different timing and spacing of their extrema and different significance of their amplitudes. The total cloud cover varied between two clearly distinguishable plateaus with mostly clear skies (values of $0.1 \pm 0.1$) from December–March and predominantly dense coverage (values of $0.8 \pm 0.1$) from May–October, indicating that





**Figure 3.** Meteorological conditions at Masaya volcano retrieved from ECMWF ERA-Interim data with resolutions of 6 h and $1° \times 1°$ and interpolated to an altitude of 700 m a.s.l. **Grey lines:** 6-hourly data. **Blue lines:** running means over the 6-hourly data with an averaging window of $\pm 7$ days. **Black dots:** around noon (18:00 UTC) data. **Red lines:** running means over the around noon data with an averaging window of $\pm 7$ days. Absolute variations are almost the same for the full data set and the around noon data only, except for the air temperature and the relative humidity which follow their expected diurnal cycles.





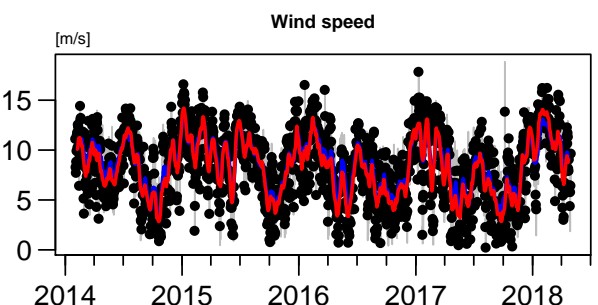
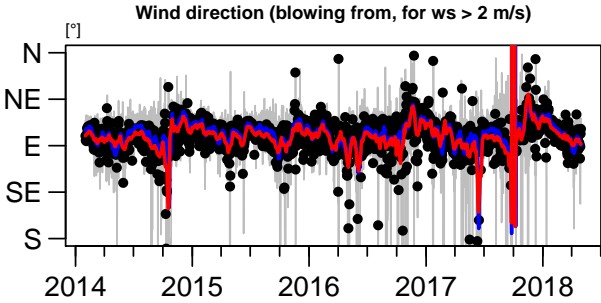

**Figure 4.** Meteorological conditions at Masaya volcano retrieved from the operational ECMWF reanalysis data. See Figure 3 for details.

Masaya is affected by the intertropical convergence zone (ITCZ) during that latter time interval. The wind speed varied between

$3$–$17\,\frac{m}{s}$, with maxima in January/February, and weaker secondary maxima in July, and with minima in June and in October. The
average of the wind direction mainly indicates easterlies $(75\pm28)°$ subject to an about linear trend every year with a step change
every year towards east-northeasterly in October followed an about linear trend towards easterly in September. In addition, the
wind conditions are rather unstable every year in June and in October (which corresponds to the times when Masaya is located
at the edge of the ITCZ). When those exceptions are ignored by limiting the circular statistics to $40$–$110°$ (which contains 88%

of the raw data), the wind conditions were rather stable with almost exclusively east–northeasterly winds from $(75\pm10)°$ almost
all the year. The barometric pressure varied between 931–935 mbar, with weak minima in October/November. The ambient air
temperature varied between 294–298 K with maxima in April and minima in January. The water vapour concentration and the
relative humidity varied between $4$–$6\cdot10^{17}\,\frac{molec}{cm^3}$ and 60–90%, respectively, with minima in February/March and maxima from
June–October. The ozone mixing ratio varied between 20–50 ppbv with minima usually around October and maxima in March.

**Operational ECMWF reanalysis data**

The accuracy of weather data especially in the mountainous regions around the volcano is directly related to the accuracy of the
model topography. Therefore an enhanced (fundamental) model resolution should result in general in more representative data.
For this purpose, we investigated the meteorological conditions at Masaya also by using the operational ECMWF reanalysis
data with a higher spatial resolution of $0.14°$x$0.14°$ (modelled in spectral domain with a truncation of T1279, interpolated to

the Gaussian grid N640). The underlying model of the operational ECMWF reanalysis data is, however, updated frequently
(e.g. since March 8 2016 the model uses a finer horizontal resolution of $0.07°$ instead of $0.14°$) and thus there are potentially
artificial jumps in its time series. Because this study analysis a long time series, such artificial changes in the model setup are
not suitable for direct application. A full list of modifications to the model setup in addition to the change in horizontal model
resolution can be found at https://www.ecmwf.int/en/forecasts/documentation-and-support/changes-ecmwf-model.



**Ground-based data from Managua airport**

The ground-based data from Managua airport are recorded at an altitude of 59 m a.s.l. and thus their quantitative values are expected to differ from the meteorological data modelled for 700 m a.s.l. Nevertheless, the ground base data agree with the ERA-Interim data in the general patterns and relative variations, with variations in the ambient temperature between 302–307 K, variations in the relative humidity between 40–70%, variations in the wind speeds between $2\text{–}8\,\frac{m}{s}$, and a mean wind direction of $(86 \pm 31)°$ (Figure B3). The wind directions differ significantly with $(102 \pm 29)°$ when considering all times of the day.

## 3    Methods

We derived semi-continuous (only during daytime) time series of the differential slant column densities (dSCD) of $SO_2$ and BrO via Multi-Axis Differential Optical Absorption Spectroscopy (MAX-DOAS) applied on UV-spectra of the diffuse solar irradiation recorded by the NOVAC stations (e.g. Edmonds et al., 2003; Galle et al., 2003; Bobrowski et al., 2003). The $SO_2$ emission fluxes and the $BrO/SO_2$ molar ratios in the volcanic plume were then derived from the dSCD data.

The spectroscopic retrieval of the $SO_2$ and BrO dSCDs as well as the spectroscopic and post-spectroscopic retrieval of the $BrO/SO_2$ molar ratios applied in this manuscript follow in large parts the evaluation described by Dinger (2019) which itself follows mainly Lübcke et al. (2014) and Dinger et al. (2018).

Our retrieval of the $SO_2$ emissions fluxes is based on the standard NOVAC approach described by Johansson et al. (2009) but (1) extended by a set of data filters which aim to reduce the amount of potentially problematic measurement conditions, (2) information on the plume height and the wind conditions has been assessed partially via a triangulation of NOVAC observations, and (3) the spectroscopic retrieval has been adapted to the high $SO_2$ emission fluxes at Masaya. One of these filters uses the absolute $SO_2$ background calibration method described by Lübcke et al. (2016).

In this section, we describe the applied retrieval steps and data filters (see Table 2). A summary and critical assessment of our retrieval steps can be found in the discussion section of this manuscript.

**Spectroscopic retrieval of the $SO_2$ dSCD distribution**

The NOVAC data were recorded by UV-spectrometers which scan across the sky from horizon to horizon in steps of 3.6° by means of a small field-of-view telescope yielding a mean temporal resolution of about 5–15 min per scan. Each scan circumvents 53 spectra: an initial zenith spectrum, a dark current spectrum, and 51 measurement spectra (Galle et al., 2010). Prior to the spectroscopic retrieval, the individual spectra of a scan were checked for their spectroscopic quality. A scan was rejected if its initial zenith spectra was either over- or underexposed (accept only spectra whose channel with the highest number of counts has recorded within 12–88% of the maximum possible count number, where the lower boundary is assessed after the dark current correction) or the single exposure time was unreliable (less than 20 ms or more than 2 s). For a passing scan, all its measurement spectra were checked for over- or underexposure and individually rejected if necessary. Furthermore, all spectra recorded at zenith angles $\varepsilon$ larger than $|\varepsilon| > 76°$ were rejected in order to avoid large light paths or spectroscopic




**Table 2.** Applied filters in chronological order (if condition fulfilled, reject data). See text for details. For comparison, the standard NOVAC retrieval applies the five following filter conditions (Johansson et al., 2009): (1) overexposure of (any) spectrum by more than 99%, (2) underexposure of (any) spectrum by less than 2.5%, (3) $\chi^2_{SO_2} > 0.9$ for a measurement spectrum, (4) number of passing spectra $< 10$, (5) a "plume completeness filter" which rejects a scan if most of the large dSCDs are close to the margin of the effective angle range.

| Filter | Filter condition |
|---|---|
| **Zenith spectrum** | |
| exposure time | $< 20\,\mathrm{ms}$ or $> 2\,\mathrm{s}$ |
| overexposure | $> 88\%$ in any channel |
| underexposure | $< 12\%$ in any channel |
| **Measurement spectra** | |
| overexposure | $> 88\%$ in any channel |
| underexposure | $< 12\%$ in any channel |
| zenith angle | $|\varepsilon| > 76°$ |
| **SO$_2$ VCD distribution** | |
| number of spectra | $< 30$ |
| maximum VCD | at margin of effective angle range |
| relative background | $> 2 \cdot 10^{17} \frac{molec}{cm^2}$ |
| Gaussian fit ($b = 0$) | did not converge or negative amplitude |
| Gaussian fit ($b$ free) | $b < -1 \cdot 10^{17} \frac{molec}{cm^2}$ |
| Gauss vs. Discrete | $I^{discr}_{SO_2} \notin 0.8\text{–}1.6 \cdot I^{fit}_{SO_2}$ |
| absolute background | $> 5 \cdot 10^{17} \frac{molec}{cm^2}$ |
| **Meteorological conditions** | |
| wind speed | $< 5\,\mathrm{m/s}$ |

artefacts due to obstacles in the light path. Accordingly, at most 43 measurement spectra could pass the quality filters. The scan was entirely rejected if less than 30 spectra passed.

For every scan passing the filters, SO$_2$ DOAS fits were applied on each of the passing spectra where the initial zenith spectrum of the respective scan was used as reference spectrum (the DOAS fit scenarios are summarised in Table 3). The result was a distribution of SO$_2$ dSCDs w.r.t. the zenith spectrum depending on the viewing direction. The SO$_2$ distributions were used for three purposes: (1) the calculation of the SO$_2$ emission fluxes, (2) the triangulation of the plume centre position for a retrieval of the plume height and the wind direction, and (3) the identification of the plume region as a preparation for the BrO/SO$_2$ retrieval.





**Table 3.** Applied DOAS fit scenarios. The two lowest lines give the parameter ranges of the Levenberg–Marquardt fit routine.

|  | $SO_2$ fit | BrO fit |
|---|---|---|
| Fit range | 314.8–326.8 nm | 330.6–352.75 nm |
| *(Pseudo-)Absorption cross sections:* | | |
| $SO_2$ | Vandaele et al. (2009), @298 K | (same) |
| $O_3$ | Burrows et al. (1999), @221 K | (same) |
| BrO | | Fleischmann et al. (2004), @298 K |
| $O_4$ | | Hermans et al. (2003) |
| $NO_2$ | | Vandaele et al. (1998), @294 K |
| $CH_2O$ | | Meller and Moortgat (2000), @298 K |
| Ring spectrum (Grainer and Ring, 1962) calculated from the particular reference spectrum | | |
| Ring spectrum multiplied with the wavelength[4] (see Wagner et al., 2009) | | |
| *Further DOAS fit parameters:* | | |
| Polynomial of order $n = 3$ in the optical depth space | | |
| Stray light polynomial of order $n = 0$ in the intensity space | | |
| Reference spectrum (+ 2 Ring spectra): $a_{shift} \in \pm 0.2\,\text{nm}$ and $a_{squeeze} \in 1 \pm 0.02$ | | |
| Absorption cross sections (linked together): $a_{shift} \in \pm 0.2\,\text{nm}$ and $a_{squeeze} \in 1 \pm 0.02$ | | |

**Choice of the wavelength range in the $SO_2$ DOAS fit**

The choice of the wavelength range (and in particular its lower limit) used in the $SO_2$ DOAS fit can cause major deviations in the spectroscopic results. The standard NOVAC evaluation routine uses 310 nm for the lower limit because this value is optimum to detect low $SO_2$ dSCDs of several $10^{17}\,\frac{\text{molec}}{\text{cm}^2}$ (which is the case for volcanoes with a low to moderate degassing strength or considerable distances of the DOAS instrument to the emission source). As a drawback, such a short wavelength for the lower limit makes the $SO_2$ DOAS fit susceptible to saturation effects resulting in an underestimations of $SO_2$ dSCDs larger than

$1 \cdot 10^{18}\,\frac{\text{molec}}{\text{cm}^2}$ (see Figure 5a and e.g. Bobrowski et al., 2010). In contrast to that, our choice of 314.9–326.8 nm can be considered to be hardly affected by saturation effects up to $SO_2$ SCDs of $1$–$2 \cdot 10^{18}\,\frac{\text{molec}}{\text{cm}^2}$ and still of acceptable accuracy at $SO_2$ SCDs of $3 \cdot 10^{18}\,\frac{\text{molec}}{\text{cm}^2}$ (less than 10% underestimation, see Figure 5b). We observed at Masaya, nevertheless, a significant amount of data with $SO_2$ SCDs above $3 \cdot 10^{18}\,\frac{\text{molec}}{\text{cm}^2}$ which were underestimated also by our retrieval (Figure 5b and 9). A separated retrieval of those data with an alternative fit range starting, e.g., at 319 nm would in general result in more accurate estimates for these

particularly large $SO_2$ SCDs. Fickel and Delgado Granados (2017) proposed such an approach for Popocatépetl volcano using even three wavelength ranges at 310–322 nm, 314.7–326.7 nm, and 322–334 nm. The risk of such a compound retrieval would be, however, artificial jumps along the chosen thresholds. We therefore hesitated to use such an approach but encourage further investigations, e.g., whether an interpolation between the results retrieved by several fit ranges could avoid this risk while





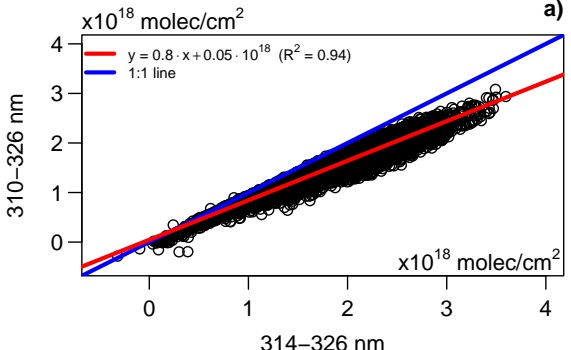
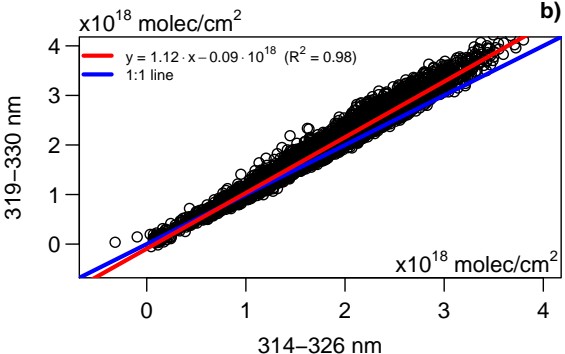

**Figure 5.** Comparison of the chosen wavelength range (x-axis) with two other wavelength ranges as retrieved for Caracol station from 2014–2020. For statistical interpretation of the 314–326 nm data: 9% of the $SO_2$ dSCDs were lower than $1 \cdot 10^{18} \frac{molec}{cm^2}$, 63% were between $1$–$2 \cdot 10^{18} \frac{molec}{cm^2}$, 26% were between $2$–$3 \cdot 10^{18} \frac{molec}{cm^2}$, 2% were larger than $3 \cdot 10^{18} \frac{molec}{cm^2}$.

enhancing the accuracy of large $SO_2$ SCDs (see Theys et al., 2017, for the implementation of such an approach in satellite

observations).

**Retrieval of the background $SO_2$ slant column density**

The subsequent data analysis requires the absolute $SO_2$ slant column density (SCD) distribution rather than the $SO_2$ dSCD distribution, where $SO_2\,SCD = SO_2\,dSCD + SCD_{SO_2,ref}$. An accurate estimate for the absolute slant column density $SCD_{SO_2,ref}$ of the reference spectrum (here: of the initial zenith spectrum) is non-trivial.

A pragmatic approach is the assumption that the scan included viewing directions which were not at all affected by a contamination with volcanic gases. Following this assumption, for the background direction $\varepsilon_{bg}$ held $dSCD_{SO_2}(\varepsilon_{bg}) = -SCD_{SO_2,ref}$. In order to be less susceptible to negative outliers, we calculated $SCD_{SO_2,ref}$ as the mean of the 8 lowest dSCDs (orange squares in Figure 6).

It has been observed, however, that the assumption of a non-contaminated background direction is not always justified (e.g.

Lübcke et al., 2016). Another approach which does not rely on that assumption is the direct retrieval of $SCD_{SO_2,ref}$ via a $SO_2$ DOAS fit of the zenith spectrum against a solar-atlas spectrum (see Salerno et al., 2009; Lübcke et al., 2016; Chance and Kurucz, 2010). This approach requires, however, to retrieve the instrument characteristics (e.g., via a principal component analysis of the residual spectroscopic structure), which is not only a time expensive procedure but also prone to introducing systematics when not carefully applied.

A third approach is the hybrid of these two approaches with the following subsequent steps: (1) apply the first approach to identify the viewing directions of the 8 lowest $SO_2$ dSCDs, (2) co-add these 8 spectra, (3) apply the second approach (i.e. evaluate against a solar-atlas spectrum) on this "added-reference-spectrum" (instead of on the zenith spectrum, see Appendix A for details). In comparison to the pure second approach, the absolute retrieval step in this hybrid approach faces in general





low—and mostly negligible—$SO_2$ SCDs. Therefore, the $SO_2$ DOAS fit of the absolute calibration retrieval can start at a wave-
length of 310 nm or even lower, resulting in general in lower statistical fit errors and weaker effects from possible spectroscopic
interferences of the $SO_2$ absorption cross section with, e.g., the imperfect estimation of the instrument characteristics. The re-
sults of the hybrid approach could be used either as a filter or for correcting the SCD data w.r.t. the retrieved background $SO_2$
SCD.

A fourth approach extends the third approach by (1) checking for a background contamination but then (2) using a reference
spectrum from another time where a background contamination has been ruled out (e.g. from the previous day at the same time)
for a subsequent iteration of the $SO_2$ fits. Both, the third and the fourth approach, have in common that the chosen reference
spectrum has not been recorded under the same conditions as the measurement spectrum. The advantage of the fourth approach
w.r.t. the third approach would be that the chosen reference spectra are expected to be recorded at least under similar conditions
as the measurement spectra (at least when the time-of-day and the ambient temperature have been considered for the selection).
The drawback of the fourth approach is the temporal variation of these systematics while the third approach would cause the
same systematics to all contaminated scans.

We applied the third approach to the data but used the results of the absolute calibration only for a rather conservative data
filtering: the absolute background $SO_2$ VCD which was derived as the product of the absolute background $SO_2$ SCD times
the mean air mass factor $\text{mean}(\cos(\varepsilon_{i \in \text{bg}}))$ but corrected by $-5 \cdot 10^{16} \frac{\text{molec}}{\text{cm}^2}$ for Caracol station and $+5 \cdot 10^{16} \frac{\text{molec}}{\text{cm}^2}$ for Nancital
station (such that the peaks of the histograms match zero $SO_2$, see Figure 8a). A scan was rejected if its (corrected) absolute
background $SO_2$ VCD exceeded $5 \cdot 10^{17} \frac{\text{molec}}{\text{cm}^2}$.

**Calculation of the $SO_2$ emission fluxes**

The retrieval of the background $SO_2$ SCD allows the calculation of the vertical $SO_2$ column densities

$$V_{SO_2}(\varepsilon) = \cos(\varepsilon) \cdot [\text{dSCD}_{SO_2}(\varepsilon) + \text{SCD}_{SO_2,\text{ref}}] \tag{1}$$

associated to the coordinates within the scan plane where the horizontal distance w.r.t. the instrument is $H(\varepsilon) \cdot \tan(\varepsilon)$ and
the mean plume height $H(\varepsilon)$ (above the horizon of the instrument) can in general vary horizontally. We highlight that eq. 1
assumes geometric air mass factors while the real air mass factors could deviate due to angle-dependent atmospheric radiative
transport effects (e.g. Mori et al., 2006; Kern et al., 2010). Examples for retrieved $SO_2$ VCD distributions are shown in the
Figures 6 and B2.

Precise information on the plume height is usually not available—not to mention spatially resolved variations of the plume
height. A commonly applied pragmatic approach is thus the assumption of a plume height constant in space and time. Assuming
that the plume height $H(\varepsilon) = H$ is constant at least in space, the $SO_2$ emission fluxes $F_{SO_2}$ for a particular time can be
calculated via

$$F_{SO_2} = M_{SO_2} \cdot v \cdot \cos(\omega - \beta) \cdot H \cdot \int_{-\infty}^{\infty} V_{SO_2}(\varepsilon) \, d(\tan(\varepsilon)) \tag{2}$$





with the molar mass of SO$_2$ $M_{SO_2} = 64$ g/mol, the absolute wind speed $v$, and the relative angle $|\omega - \beta| < \frac{\pi}{2}$ between the wind direction and the scan plane (see Figure 2). We highlight that the integral can be understood as a spatial integral which integrates along a straight horizontal line by steps of $d(H \cdot |\tan(\varepsilon)|)$.

The angular integral $I_{SO_2} = \int_{-\infty}^{\infty} V_{SO_2}(\varepsilon) \, d(\tan(\varepsilon))$ can be calculated in good approximation by a discrete summation of the spectroscopically retrieved SO$_2$ VCD distribution via

$$I_{SO_2} \approx \sum_{i=1}^{n-1} \frac{V_i + V_{i+1}}{2} \cdot [\tan(\varepsilon_{i+1}) - \tan(\varepsilon_i)] \tag{3}$$

where $n$ is the number of all individual spectra (i.e. individual viewing directions) which passed the filters discussed above and the $V_i$ are the vertical column densities calculated according to eq. 1 and associated to the horizontal coordinates $H \cdot \tan(\varepsilon_i)$ as explained above.

Up to here, the paragraph followed the standard NOVAC approach (Johansson et al., 2009). This approach tacitly assumes
that the measurement conditions have not changed significantly during one scan. This assumption could be frequently not justified for several causes, e.g. unstable wind conditions or intra-minute variations in the volcanic degassing source strength (e.g. Pering et al., 2019). In the next paragraph, we present a set of filters which reject data which is potentially influenced by unstable measurement conditions. Our approaches to estimate the wind conditions as well as the plume height are discussed starting in the next but one paragraph.

**Filtering of unstable conditions**

For stable meteorological and radiative conditions as well as a constant SO$_2$ emission strength, the horizontal broadening of a volcanic plume is caused predominantly by turbulent diffusion. Under such ideal measurement conditions, the SO$_2$ VCD distribution would be a Gaussian distribution w.r.t. the relative distance $H \cdot \tan(\varepsilon_i)$. A Gaussian shape is indeed observed in good approximation for the large part of the scans where exactly one plume has been identified (e.g. Figures 6 and B2d).
However, there is also a significant amount of scans where the retrieved SO$_2$ VCD distribution differs significantly from an ideal Gaussian shape. The predominant reason for such deviations is an apparent secondary plume (presumably either because there is another plume or because the wind direction has changed during the scan) but also less well defined, rather random shapes have been observed (see Figure B2a–c).

As motivated above, scans with unstable conditions should be rejected. For this purpose, we fitted a Gaussian distributions

$$V_i(\varepsilon_i) = a \cdot \exp\left[-\left(\frac{\tan(\varepsilon_i) - \mu}{w}\right)^2\right] + b \tag{4}$$

as a function of $\tan(\varepsilon)$ to the SO$_2$ VCD distribution (with the fit parameters $a$, $\mu$, $w$, $b$) as a tool to semi-quantitatively assess the "degree of stability" during that scan. Actually, two Gaussian distributions have been fitted to the SO$_2$ VCD distribution, once with a fixed $b = 0$ and once with a free parameter $b$ (see the solid and dashed lines in Figure B2, while both lines perfectly overlap in Figure 6).

The Gaussian fits (and other criteria) were used to filter the data in six subsequent steps (F1)–(F6), a scan was rejected if:





(F1) its highest SO$_2$ VCD was retrieved at the margin of the effective angle range (which is usually at $\pm 75.6°$) because in such a case at least half of the plume area was not included, (F2) the discrete SO$_2$ VCD offset exceeded $2 \cdot 10^{17} \frac{\text{molec}}{\text{cm}^2}$ because for stable conditions that offset should be negative by construction, (F3) the Gaussian fit with fixed $b = 0$ did not converge or proposed a negative amplitude (20% and 5% of the scans were rejected by these filters for Caracol station and Nancital station, respectively).

Next, we highlight that the Gaussian fit would tend to propose a positive offset parameter $b > 0$, e.g. if a secondary plume elevates the apparent background level. In contrast to that, significant negative values for $b$ indicate that the effective scan range does not include the gas-free background. Accordingly, a scan was rejected if (F4) $b < -1 \cdot 10^{17} \frac{\text{molec}}{\text{cm}^2}$ (3% and 1% of scans rejected).

Next, we compared the Gaussian integral $I_{\text{SO}_2}^{\text{fit}} = \sqrt{2 \cdot \pi} \cdot a \cdot w$ (retrieved for $b = 0$) and the discrete integral $I_{\text{SO}_2}^{\text{discr}}$. On the one hand, the Gaussian integral is also for $b = 0$ usually smaller than the discrete integral (e.g. because of a secondary peak or an asymmetric plume shape). On the other hand, our filtering of data with $|\varepsilon_i| > 76°$ implies the tacit assumption of $V_i = 0$ for $|\varepsilon_i| > 76°$ for eq. 3. When this assumption does not hold true, the discrete integral underestimates the overall SO$_2$ amount while the Gaussian integral could correctly include those contributions. Furthermore, a scan was rejected if its two integrals differed rather strongly, that is, a scan passed only if (F5) $I_{\text{SO}_2}^{\text{discr}} \in 0.8\text{--}1.6 \cdot I_{\text{SO}_2}^{\text{fit}}$ (20% and 10% of scans rejected, with 12% and 5% due to the lower threshold and 8% and 5% due to the upper threshold). Furthermore, for a passing scan the higher value $I_{\text{SO}_2}^{\text{finally}} = \max(I_{\text{SO}_2}^{\text{discr}}, I_{\text{SO}_2}^{\text{fit}})$ of the two integrals was chosen (where $I_{\text{SO}_2}^{\text{fit}}$ was chosen for 28% and 23% of the scans).

As a last filter (F6), a scan was rejected when its absolute SO$_2$ background VCD exceeded $5 \cdot 10^{17} \frac{\text{molec}}{\text{cm}^2}$ (see above, 8% and 3% of scans rejected).

In total, 57% and 82% of the scans passed the filters for Caracol station and Nancital station, respectively. We consider this a good compromise between the lack of temporal resolution and reducing the risk of systematics due to unstable measurement conditions.

**Triangulation of the plume centre position**

When the volcanic gas plume is observed simultaneously by two NOVAC stations, the two associated viewing directions towards the plume centre can be used for a triangulation of the spatial position of the plume centre. The relationship between the plume height $H_s$ above the horizon of the NOVAC station $s$ and the wind direction $\omega$ is given by

$$H_s + A_s = D_s \cdot \left| \frac{\sin(\omega - \sigma_s)}{\cos(\omega - \beta_s) \cdot \tan(\varepsilon_s)} \right| \tag{5}$$

where the fixed station geometry parameters $A$, $D$, $\sigma$, $\beta$ are summarised in Table 1 and the horizontal geometrical considerations are sketched in Figure 2. We highlight that the total plume altitude $H_s + A_s$ is the same for both instruments. If NOVAC station $s$ observes $\varepsilon_s = 0°$, the wind direction is trivially given by $\omega = \sigma_s$ and the plume height can be retrieved by applying eq. 5 on the other station.

We used the peak positions of the above introduced Gaussian fits (with $b = 0$) as estimates for the plume centre position. A practical limitation of the triangulation was that the NOVAC stations did not measure exactly simultaneously. Therefore a tem-



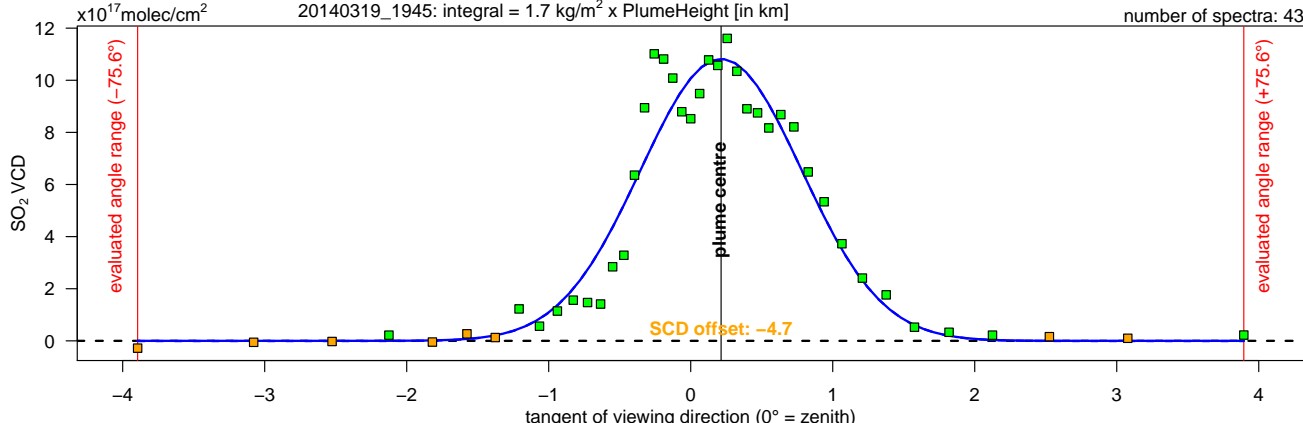

**Figure 6.** $SO_2$ VCD distribution retrieved from the scan starting at 2014-03-19 19:45 UTC recorded at Caracol station. The green and orange squares give the retrieved angular $SO_2$ VCD distribution. Only data for zenith angles between $-75.6°$ and $+75.6°$ are considered. The orange squares are used for the retrieval of the background SCD (the set of the lowest SCDs does not necessarily match perfectly with the set of the lowest VCDs). The (negative) background SCD is given in orange. The plume centre is retrieve via the Gaussian fit (solid blue line).

poral binning of their data was required. We calculated bins of 30 min. A bin was rejected if the plume centre varied for one
instrument such that the standard deviation exceeded $20°$ (within these 30 min).

The plume centre triangulation proposed an average wind direction of $(84 \pm 3)°$ and average total plume altitudes of $H + A_s = (755 \pm 102)$ m which implied an average plume height above horizon of $(373 \pm 102)$ m for Caracol station and $(415 \pm 102)$ m for Nancital station (Figure 7).

**Estimates for the wind speed and the wind direction**

We based our analysis in general on the meteorological parameters from the ECMWF ERA-Interim data because this dataset allows an analysis consistent in time, i.e. without potential jumps in the time series. Nevertheless, the ERA-Interim data have to be expected to provide only limited accuracy in particular in the complex topology around volcanoes. Therefore, we applied the following local calibrations of the ERA-Interim data.

We compared the wind speeds $v_{era}$ provided by the ERA-Interim dataset and $v_{oad}$ provided by the operational ECMWF re-
analysis data. We consider the latter to be in general more accurate estimates due to its higher spatial resolution. Despite that the wind speeds of both datasets are highly correlated (coefficient of 89%), their scatter plot significantly deviates from a proportional relationship with

$$v_{oad} = 0.53 \cdot v_{era} + \sqrt{v_{era}} \qquad \left( \equiv v_{calibrated} \right) \qquad (6)$$

being apparently a good fit (Figure 8b). All wind speed data used in our further evaluation steps were retrieved from the ERA-
Interim data but calibrated according to eq. 6.







**Figure 7.** Results of the plume centre triangulation. Data has been binned in 30 min intervals and the means of the plume centre have been compared. A bin has been rejected if the plume centre varied for one instrument such that the standard deviation exceeded 20°. **a)** Histograms of the retrieved wind directions. **b)** Scatter plot of the retrieved plume altitudes and the retrieved wind directions. The altitudes of the Caracol and Nancital stations and of the volcanic edifice are marked by blue, green, and grey lines. The effective field of view $|\varepsilon| < 76°$ of the two instruments is given by the curvy blue and green lines. The dominant bulk of observations centred at approximately (84°, 756 m a.s.l.) refers to observations when both instruments nearly simultaneously recorded the sample plume while the "wings" to the upper left and upper right corners refer to observation where both instruments recorded at the same time different plumes. The wings are presumably artefacts caused by the simplicity of the triangulation approach given in equation 5. **c)** Histogram of the retrieved plume altitudes.

**Figure 8.** Summary of several empirical observations used for filtering and estimations. **a)** Histograms of the absolute $SO_2$ SCD of the background spectrum for both instruments for their respective total time series. **b)** Comparison of the wind speeds from the ECMWF ERA-Interim data and the operational ECMWF reanalysis data. **c)** Comparison of the distribution of the wind directions estimated by four different methods. **d)** Scatter plot of the triangulated plume height and the calibrated wind speed. The red dotted line indicates the relationship between wind speed and plume height. **e+f)** Correlation between the retrieved $SO_2$ emission fluxes and the wind speed. The plots compare daily $SO_2$ means and the means of the wind speed a the respective measurement times. **e)** Original ERA-Interim wind speeds versus the non-calibrated $SO_2$ emission fluxes calculated with original ERA-Interim wind conditions and a fixed plume altitude of 635 m a.s.l. **f)** Calibrated wind speeds versus the $SO_2$ emission fluxes calculated with the calibrated wind conditions and a dynamic plume altitude as a function of the wind speed.





We consider the wind directions retrieved via the triangulation of the NOVAC results as the "ground truth". The operational ECMWF reanalysis data matches well with these data, what further supports this assumption. The ERA-Interim data, however, provides wind directions which are further to the East-Northeast by 11° (Figure 8c). All wind direction data used in our further evaluation steps were retrieved from the ERA-Interim data but calibrated according to eq. 7:

$$\omega_{\mathrm{calibrated}} = \omega_{\mathrm{era}} + 11° \tag{7}$$

(that is a shift from east-northeastery towards easterly).

These two calibrations could not be expected to improve the accuracy for every single data point but result arguably in average in a more accurate data set.

**Estimates for the plume height**

The plume altitude is a major source of uncertainty in the calculation of the $SO_2$ emission fluxes. When lacking visual observations, the plume altitude is usually assumed to be fixed to the altitude level of the volcano summit or the expected effective plume height of the volcanic plume.

The plume height can be considered to vary significantly and depending on the wind conditions. The initial buoyancy of the volcanic plume is just one mechanism which links the plume height and the wind speed. The volcanic plume is usually hotter 370 than the ambient atmosphere and thus rises until its temperature is equilibrated due to adiabatic cooling and mixing with ambient air. Accordingly, higher wind speeds should result in average in lower observed plume heights for at least two reasons: First, the higher the wind speed the larger is the atmospheric turbulence, the larger is the cooling rate of the plume, and thus the lower is the effective plume height. Second, the higher the (horizontal) wind speed, the smaller has been the propagation time between release and observation, and thus the lower is the probability that measured plume has already reached its effective 375 plume height.

The comparison of the triangulated plume height with the wind speed (calibrated as explained above) confirmed such a causal link between the plume height and the wind speed with a weak linear relationship of (Figure 8d)

$$H_s[\text{in m}] + A_s[\text{in m}] = 907\,\text{m} - 13\,\text{m} \cdot v_{\mathrm{calibrated}}[\text{in m/s}]. \tag{8}$$

We used this relationship for dynamic estimates of the plume height as a function of the wind speed. We are aware that 380 this relationship is subject to a large scatter, though we consider it a better best guess than applying a fixed plume height. In particular, using a fixed plume height could result in apparent seasonal variations in the $SO_2$ emission fluxes which are, however, possibly only inherited artefacts due to seasonal variations of the wind speed (see next paragraph).

**Correlation of $SO_2$ emission fluxes and wind speeds**

We observed a strong correlation between the $SO_2$ emission fluxes and the wind speeds when none of our estimation approaches 385 for the wind speed, the wind direction, or the plume height were applied (correlation coefficient of 82% when all wind speeds are considered and of 53% when only wind speeds larger than 10 m/s are considered, Figure 8e). This correlation was lower for





the calibrated data (correlation coefficient of 69% when all wind speeds are considered) and in particular basically vanished for wind speeds larger than 10 m/s (correlation coefficient of 19%, Figure 8f).

The $SO_2$ emission fluxes are of magmatic origin and thus no causal link to the meteorological conditions would be expected.
For better readability, we postponed the discussion of this observation to section 5 "Discussion of the $SO_2$ emission flux retrieval". There our conclusions were that the observed correlation between the $SO_2$ emission fluxes and the wind speed is rather not a real observation but is more likely caused by misestimations of the $SO_2$ emission fluxes and in particular due to neglecting the variations in the plume height.

On the one hand, our proposed retrieval extension were able to correct this spurious correlation for wind speeds larger than
10 m/s (Figure 8f). On the other hand, we were not able to explain or correct for that correlation for wind speeds smaller than 10 m/s. Accordingly, it could be appropriate to reject all data with wind speeds smaller than 10 m/s but this would massively reduce our dataset. As a compromise between data reliability and temporal resolution we thus applied a more conservative filter and reject only those data with wind speeds smaller than 5 m/s. This subsequent filter rejected 15% and 22% of the remaining scans for Caracol and Nancital station, respectively (but 72% and 78% when rejecting all data with wind speeds smaller than
10 m/s).

**Retrieval of the BrO/SO$_2$ molar ratios**

The optical density of BrO in a volcanic gas plume is at least one order of magnitude smaller than for $SO_2$ and thus a higher photon statistic is required for sufficiently precise BrO results beyond the detection limit. At manually controlled measurements, this is realised by a sufficiently large number of consecutive exposures (and besides, typical state-of-the-art campaign
instruments are much more precise than NOVAC instruments due better spectrometers and an active temperature stabilisation). For NOVAC data optimised w.r.t. the $SO_2$ retrieval requirements, the required larger number of exposures per spectrum can be realised by a subsequent spectral adding of spectra which are recorded in the temporal proximity and in the same or at least similar viewing direction.

For this purpose, the retrieved $SO_2$ dSCD distribution was used to identify all spectra which were predominantly part of the
volcanic plume and then these spectra are added in order to get one "added-plume-spectrum" per scan. Analogously, the spectra which were associated with the 10 lowest $SO_2$ dSCDs are added in order to get one "added-reference-spectrum". The drawback of this method is the loss of spatial information because the retrieval derives only one averaged value for the BrO dSCDs and thus for the BrO/SO$_2$ molar ratios. Accordingly, this approach does not allow to investigate possible variations of the BrO/SO$_2$ molar ratio as a function of, e.g., the distance to the plume centre.

As mentioned above, a volcanic gas plume can be expected to have an about Gaussian shaped angular gas distribution embedded in a flat, gas-free reference region. We retrieved the plume region by a Gaussian distribution fitted on the $SO_2$ dSCD distribution as a function of the zenith angles. The standard deviation range of the Gaussian distribution ($\alpha_{\mathrm{peak}} \pm \sigma_{\mathrm{Gauss}}$) was then defined as the plume region. The applied filters of the BrO/SO$_2$ retrieval were less strict than for the $SO_2$ emission flux retrieval: A scan was rejected from the further analysis only if the Gaussian fit failed to converge or if $\sigma_{\mathrm{Gauss}} < 5°$. Furthermore,
if $\sigma_{\mathrm{Gauss}}$ was rather large the such defined plume region may had overlapped with the reference region. To avoid this inconsis-





tency, if the such defined (Gaussian) plume region included more than 10 spectra, the plume region was instead defined as the angle range with the highest running mean value over 10 spectra for the $SO_2$ dSCD. The spectra associated to the plume region were spectroscopically added to one "added-plume-spectrum".

We highlight that it would be more consistent to fit the Gaussian distribution on the $SO_2$ VCD distribution as a function of the tangent of the zenith angles (instead of a fit on the $SO_2$ dSCD distribution as a function of the zenith angles). Nevertheless, the maximum possible effect would be that $\pm 1$ spectrum is included to the plume region. We used the fit in the dSCD–angle space for practical and historical reasons but encourage to fit in the VCD–tangent space instead for maximum consistency with the $SO_2$ flux retrieval.

The added-plume-spectra and added-reference-spectra per scan were used for a second iteration of the spectroscopic retrieval in 430 order to retrieve $SO_2$ and BrO dSCDs representative for the plume centre. From this set of scans, all scans with sufficiently reliable BrO fits (here: scans with $\chi_{BrO} < 2 \cdot 10^{-3}$) were used for a third iteration: the added-plume-spectra and added-reference-spectra of 4 consecutive scans were spectroscopically added and again $SO_2$ and BrO DOAS fits were applied. An $I_0$-correction was applied to these final data, what had not been done beforehand in order to save evaluation time.

The $SO_2$ and BrO dSCDs and the BrO/$SO_2$ molar ratios discussed in this manuscript refer to the results of the third spectro-435 scopic iteration (Figure 9). We highlight that these dSCDs are not absolutely calibrated for a background contamination (see above) because no reliable method for a absolute calibration of a background contamination with BrO has been developed. The interpretation of the BrO/$SO_2$ molar ratios thus tacitly assumes that a possible background contamination has the same BrO/$SO_2$ molar ratio as the main plume. For first investigations of this assumption and possible advances towards a correction of a BrO contamination see Wilken (2018).

We highlight that the retrieval of the BrO/$SO_2$ molar ratios is hardly affected by the numerous potential sources of systematic effects as it is the case for the $SO_2$ emission fluxes. For instance, the BrO/$SO_2$ molar ratio is not affected by assumptions on the air mass factor and on the plume height. In addition, the BrO/$SO_2$ molar ratio appears to be hardly susceptible to systematic effects in the radiative transport because the quantitative effects are similar for BrO and $SO_2$ and thus cancel in good approximation (Lübcke et al., 2014).

## 445  4  $SO_2$ and BrO time series at Masaya

In this section, we present the $SO_2$ and BrO time series retrieved from the NOVAC data. General volcanological observations of the lava lake activity suggested a separated discussion of three time intervals: (1) "prior to the lava lake elevation (until late 2015)", (2) "period of high lava lake activity (from December 2015 to May 2018)", and (3) "period of low lava lake activity (from May 2018 on)". The transition between these three activity periods were unfortunately coinciding by chance with the 450 two major data gaps in the NOVAC time series from September 9 to November 16 2015 and from March 21 to June 23 2018. The statistical analysis results discussed in the following, therefore, refers to the time intervals (1) March 2014 − September 2015, (2) November 2015 − March 2018, (3) June 2018 − March 2020. Accordingly, the analysis assumed that the volcanic activity was enhanced already in mid November 2015 or earlier.





**Table 4.** Main statistical properties of the spectroscopic results for Caracol station. Early BrO/SO$_2$ NOVAC observations between 2007–2009 are listed for completeness. For both time intervals the average BrO/SO$_2$ molar ratios peaked around March with monotonous decreasing trends else, indicating that the BrO/SO$_2$ molar ratios were also from 2007–2009 affected by a similar annual cyclicity.

| time interval | SO$_2$ emission fluxes (in $1000\,\text{t}\,\text{d}^{-1}$) | | | BrO/SO$_2$ molar ratios (in $10^{-5}$) | | | "BrO emission fluxes" |
| | daily means | daily variation | daily maxima | daily means | annual trend (in $10^{-5}\,\text{a}^{-1}$) | amplitude of annual cycle | (in $\text{kg}\,\text{d}^{-1}$) |
| --- | --- | --- | --- | --- | --- | --- | --- |
| *Apr 2007 − Jun 2007* | | | | $3.3 \pm 0.9$ | | | |
| *Sep 2008 − Feb 2009* | | | | $4.9 \pm 1.4$ | | | |
| (1) Mar 2014 − Oct 2015 | $1.0 \pm 0.2$ | $0.3 \pm 0.1$ | $1.8 \pm 0.4$ | $2.9 \pm 1.5$ | $-0.1 \pm 0.1$ | $\pm 1.4$ | $44 \pm 14$ |
| (2) Nov 2015 − Mar 2018 | $1.0 \pm 0.3$ | $0.3 \pm 0.1$ | $1.7 \pm 0.6$ | $4.8 \pm 1.9$ | $1.2 \pm 0.1$ | $\pm 1.8$ | $72 \pm 18$ |
| (3) Jun 2018 − Mar 2020 | $0.7 \pm 0.2$ | $0.2 \pm 0.1$ | $1.1 \pm 0.3$ | $5.5 \pm 2.6$ | $-0.8 \pm 0.2$ | $\pm 2.6$ | $56 \pm 18$ |

The consistent patterns in the gas data observed from November 2015 − March 2018 (see below) supports this assumption and
may furthermore indicate that the elevation of the lava lake level lagged by several weeks behind the actual onset of activity in the shallow magmatic system. The long-term averages of these time intervals were retrieved for intervals spanning over exact multiples of a year in order to avoid biases due to the seasonal modulation, namely September 1 2014 − September 1 2015, January 1 2016 − January 1 2018 (i.e. this was not affected whether the time intervals (1) and (2) were separated in December 2015 or already in November 2015), and January 1 2019 − January 1 2020.

**SO$_2$ and BrO dSCDs**

The data from Caracol station and Nancital station were in general in good agreement (correlation coefficients of 0.82 and 0.77 for daily averages of SO$_2$ and BrO dSCDs). The Caracol station observed in average higher dSCDs with a relative factor of $1.18 \pm 0.21$ for SO$_2$ and $1.06 \pm 0.24$ for BrO (when neglecting data with BrO dSCDs below $5 \cdot 10^{13}\,\frac{\text{molec}}{\text{cm}^2}$). Analogously, relative factors of $1.12 \pm 0.20$ and $0.99 \pm 0.19$ were observed for the daily averages of the SO$_2$ emission fluxes (see section 5
and Figure 10) and of the BrO/SO$_2$ molar ratios, respectively.

In the following, we discuss the typical variations observed by Caracol station. From March 2014 to September 2015, the SO$_2$ dSCDs varied between $1$–$3 \cdot 10^{18}\,\frac{\text{molec}}{\text{cm}^2}$ and the BrO dSCDs had daily maxima of about $1.5 \cdot 10^{14}\,\frac{\text{molec}}{\text{cm}^2}$ but with peaks of up to $3 \cdot 10^{14}\,\frac{\text{molec}}{\text{cm}^2}$. From November 2015 to March 2018 (period of high lava lake activity), the SO$_2$ dSCDs varied predominantly between $1$–$4 \cdot 10^{18}\,\frac{\text{molec}}{\text{cm}^2}$ but with 9% of the data varying between $4$–$8 \cdot 10^{18}\,\frac{\text{molec}}{\text{cm}^2}$ and the BrO dSCDs had doubled with daily
maxima of about $3 \cdot 10^{14}\,\frac{\text{molec}}{\text{cm}^2}$ with peaks of up to $6 \cdot 10^{14}\,\frac{\text{molec}}{\text{cm}^2}$. From June 2018 to March 2020, the SO$_2$ dSCDs were lower again and varied between $1$–$2 \cdot 10^{18}\,\frac{\text{molec}}{\text{cm}^2}$ and the BrO dSCDs had daily maxima of about $1.5 \cdot 10^{14}\,\frac{\text{molec}}{\text{cm}^2}$. In summary, the SO$_2$ and BrO dSCDs time series showed for the second time interval enhanced long-term averages but also a significantly larger variability.

Furthermore a Lomb-Scargle periodicity analysis indicated that the SO$_2$ dSCDs followed an annual cycle with pronounced







**Figure 9. a-c)** Time series of the differential slant column densities of $SO_2$ and BrO and calculated daily means of the $BrO/SO_2$ molar ratios in the gas plume emitted from Masaya (tick marks indicate first day of the particular month). The two NOVAC stations are indicated by the different colours. **c)** Best fits of the long-term pattern are given for three individual time intervals (orange lines). The yellow bands indicate the long-term averages and the standard deviations. **d)** Residual $BrO/SO_2$ time series when subtracting the best fits from the three individual parts of the $BrO/SO_2$ time series. **e)** Daily means of the $SO_2$ emission fluxes.





minima during January of each year (false alarm probability of $3 \cdot 10^{-211}$) and that the BrO dSCDs followed an annual cycle
($3 \cdot 10^{-213}$) with an additional semi-annual modulation ($2 \cdot 10^{-110}$).

**Patterns in the BrO/SO$_2$ time series**

Considering the whole time series from 2014–2020, the average BrO/SO$_2$ molar ratios were $(4.4 \pm 2.3) \cdot 10^{-5}$ and subject to
characteristic variations between $1$–$10 \cdot 10^{-5}$. The BrO/SO$_2$ molar ratios strongly differed between the three periods of volcanic
activity with average BrO/SO$_2$ molar ratios of $(2.9 \pm 1.5) \cdot 10^{-5}$, $(4.8 \pm 1.9) \cdot 10^{-5}$, and $(5.5 \pm 2.6) \cdot 10^{-5}$ (see yellow bars in
Figure 9c).

In addition to the variations described in the previous section, the BrO/SO$_2$ time series indicated an extremely significant
annual cycle with maxima in early March accompanied by a semi-annual modulation (indicated by a Lomb-Scargle analysis,
false alarm probability of $9 \cdot 10^{-74}$) as well as a varying long-term trend. These patterns were investigated for each of the three
time intervals separately by fitting linear trends plus a sinusoidal variation with a period of one year to the respective BrO/SO$_2$
time series. For all three time intervals the timing of the annual cycle remained basically the same but the average amplitude of
the cycle varies between the three time intervals being $1.4 \cdot 10^{-5}$, $1.8 \cdot 10^{-5}$, $2.6 \cdot 10^{-5}$, respectively. The accompanying linear
trends in the BrO/SO$_2$ time series were $(-0.07 \pm 0.11) \cdot 10^{-5}$, $(1.22 \pm 0.09) \cdot 10^{-5}$, $(-0.84 \pm 0.17) \cdot 10^{-5}$ per year for the three
time intervals (see Table 4). An extrapolation of the trends of the two earlier time intervals to December 11 2015, that is the
date of the lava lake elevation, implied an apparent step increase by $0.7 \cdot 10^{-5}$ in the average BrO/SO$_2$ molar ratios.

The residual patterns were investigated by subtracting the fitted variations (annual cycle and trend) from the respective time
series for the three time intervals. Most residual variations spanned between $\pm 2 \cdot 10^{-5}$ subject to a standard deviation of
$1.3 \cdot 10^{-5}$ and some outliers of up to $9 \cdot 10^{-5}$ (Figure 9d). A Lomb-Scargle periodicity analysis indicated a weak semi-annual
modulation with an amplitude of $0.5 \cdot 10^{-5}$ of the dominant annual periodicity with maxima in each March and September
(false alarm probability of $9 \cdot 10^{-16}$).

**SO$_2$ and minimum bromine emission fluxes**

For Caracol station and separated for the three time intervals (a) the mean daily averages of the SO$_2$ emission fluxes, (b) the
average daily variability, and (c) the averages of the daily maximum SO$_2$ emission fluxes are listed in Table 4. From March
2014 – March 2018, the daily means of the SO$_2$ emission fluxes were in general constant at $(1.0 \pm 0.3) \cdot 10^3 \, \mathrm{t} \, \mathrm{d}^{-1}$ with the
exception of December 2015 – February 2016 (i.e. in the three months after the elevation process) when they were enhanced at
$(1.3 \pm 0.3) \cdot 10^3 \, \mathrm{t} \, \mathrm{d}^{-1}$. Furthermore, a Lomb-Scargle analysis indicated a weak semi-annual cyclicity in the SO$_2$ emission fluxes
(false alarm probability of $1 \cdot 10^{-22}$). The product of the SO$_2$ emission fluxes and the BrO/SO$_2$ molar ratios $R_{\mathrm{BrO/SO_2}}$ allowed
the calculation of the apparent BrO emission fluxes $F_{\mathrm{BrO}} = F_{\mathrm{SO_2}} \cdot R_{\mathrm{BrO/SO_2}} \cdot \frac{M_{\mathrm{BrO}}}{M_{\mathrm{SO_2}}}$ (with the molar masses $M_i$). The according
apparent BrO emission fluxes would be 44, 72, and 56 $\mathrm{kg} \, \mathrm{d}^{-1}$ for the three time intervals. The apparent BrO emission fluxes
multiplied with $\frac{M_{\mathrm{Br}}}{M_{\mathrm{BrO}}} = 0.83$ can be considered as lower limits for the total bromine emission fluxes, because not all emitted
bromine would have been transformed into BrO. Arguably, the total bromine emission fluxes were at least a factor of 2 larger
than the derived apparent BrO emission fluxes (von Glasow, 2010; Roberts et al., 2014).



# 5 Discussion of SO$_2$ emission flux retrieval

**Intrinsic uncertainty in the SO$_2$ emission fluxes**

The simultaneous observation of basically the same volcanic gas plumes by two close NOVAC stations was a rare opportunity to retrieve empirically the lower limit of the uncertainty of the SO$_2$ emission fluxes. For ideal measurements, both stations would observe identical SO$_2$ emission fluxes, under real measurement conditions systematic as well as statistical deviations can be expected.

It is important to remark that the two stations usually did not recorded the exactly same plume but their telescopes pointed
at different times to the volcanic plume, with time differences of several minutes between their "simultaneous" observations. Pering et al. (2019) reported for SO$_2$ camera measurements at the crater rim, however, that the SO$_2$ emission fluxes frequently vary by more than 100% within minutes. We observed a similar variability when we analysed the SO$_2$ emission fluxes retrieved by the two stations with only several minutes between their observations. Accordingly, the higher the temporal resolution of the compared data, the larger is the expected scatter of the comparison.

We calculated the ratios of the SO$_2$ emission fluxes retrieved by Caracol station divided by the SO$_2$ emission fluxes retrieved by Nancital station using several temporal bin sizes. The relative factor and standard deviation of the scatter were $1.22 \pm 0.55$ for a 10-min binning, $1.19 \pm 0.40$ for a 1-hour binning, $1.13 \pm 0.21$ for daily means, and $1.11 \pm 1.15$ for weekly means (the 1-hourly data and the daily means are shown in Figure 10). Our observations confirmed the significant reduction in the scatter with increasing bin size. In contrast to that, we observed a rather persistent relative factor of 1.1–1.2 for all bin sizes, with
nevertheless a weakly decreasing trend as a function of the bin size.

The observed relative factor of 1.13 for daily means is relatively small in view of the uncertainties in the estimates of the meteorological conditions but also other measurement uncertainties. There are four obvious candidates which may have contributed to that factor: (i) wrong geometric parameters of the NOVAC stations, (ii) misestimations of the wind direction or the plume height, (iii) systematic deviations in the spectroscopic retrieval, and (iv) radiative transport effects.

(i) The most mundane cause for the observed offset would be wrong information on the viewing directions of the telescopes of the NOVAC instruments. For instance, w.r.t. a wind direction of $84°$ a variation of the scan plane orientation $\beta$ by $\pm 15°$ would result in a systematic miscalculation of the SO$_2$ emission fluxes by a factor of 0.92–1.01 for the Caracol station or 0.92–1.02 for the Nancital station, i.e. up to a relative factor of 1.11. Analogously, a misalignment of the zenith angle by as little as $\pm 5°$ can cause a systematic miscalculation of the VCDs (and thus the SO$_2$ emission fluxes) by a factor of 0.9–1.1 when the volcanic
plume is observed at $\pm 50°$. If both stations are affected, such apparently negligible misalignments of the zenith angle can cause a relative factor of about 1.2 between both stations.

(ii) A misestimation of the plume altitude can not only result in an absolute misestimation of the SO$_2$ emission fluxes but can also contribute to the observed relative factor because the stations are installed at different altitudes. For instance and for the particular conditions at Masaya, using a mean plume altitude of $1000 \, \mathrm{m}$ a.s.l. instead of $635 \, \mathrm{m}$ a.s.l. would cause a relative
factor of 1.09.

(iii) A more subtle source for the observed relative factor and scatter could be the relation between an underestimation of the



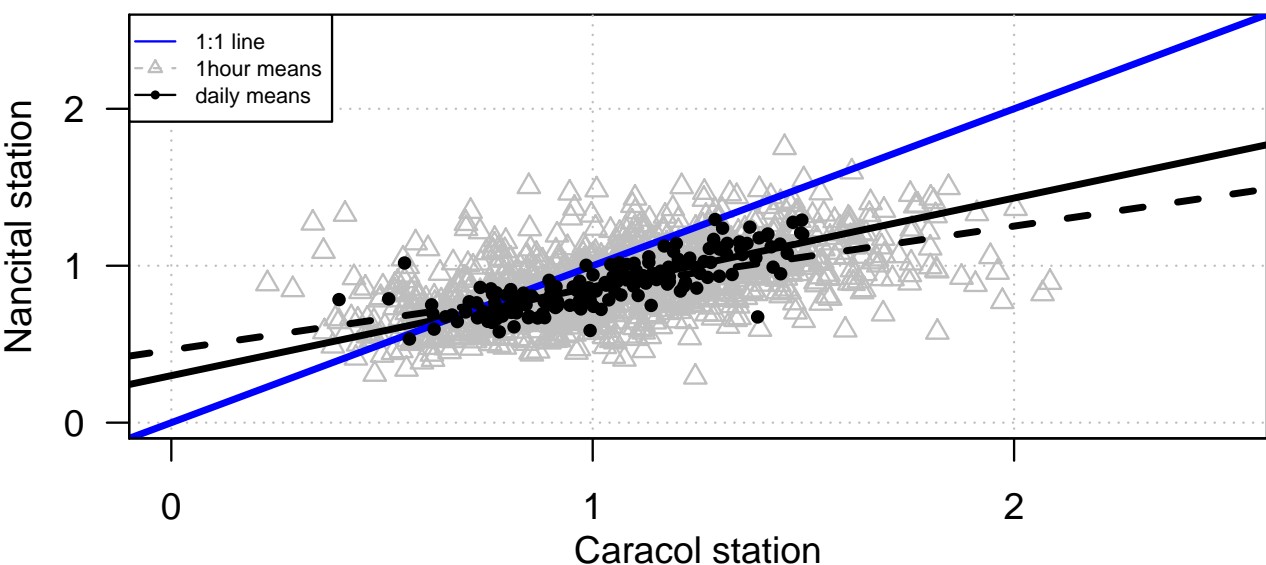

**Figure 10.** Comparison of the SO$_2$ emission flux estimates when both NOVAC stations observed volcanic plumes within the same time bin of hourly means (grey triangles) and daily means (black circles). Data compares the bin averages and only data for wind speeds larger than 5 m/s were considered.

SO$_2$ VCD and the absolute zenith angle: given a fixed SO$_2$ VCD, the larger the absolute zenith angle, the larger is the observed SO$_2$ dSCD, and thus the larger is the probability of a significant underestimation of the SO$_2$ VCD. Accordingly, if one of the instruments systematically more often records shallow plumes than the other instrument, this instrument would thus retrieve

systematically lower SO$_2$ emission fluxes. Both instruments, nevertheless, observed the volcanic plumes in average at the same (absolute) zenith angles and thus this possible source appears to be irrelevant here.

(iv) There could be significant deviations in the SO$_2$ emission fluxes recorded by the two stations due to different radiative transport effects. For Masaya, the radiative transport effects associated to the relative position of the sun were, however, presumably rather similar for both NOVAC stations because for March–October the sun was for most of the day close to the zenith.

Relative differences in the radiative transport caused, e.g. when there were systematically more clouds either to the North or the South of the NOVAC stations, could be nevertheless not ruled out as a source for a relative factor.

**Correlation of SO$_2$ emission fluxes and wind speeds**

We observed a strong correlation between the SO$_2$ emission fluxes and the wind speeds when none of our estimation approaches for the wind speed, the wind direction, or the plume height were applied (correlation coefficient of 82% when all wind speeds are considered and of 53% when only wind speeds larger than 10 m/s are considered, Figure 8e). This correlation was lower for






the calibrated data (correlation coefficient of 69% when all wind speeds are considered) and in particular basically vanished for wind speeds larger than 10 m/s (correlation coefficient of 19%, Figure 8f).

The $SO_2$ emission fluxes are of magmatic origin and thus no causal link to the meteorological conditions would be expected. There are three groups of possible causes for this observation: (1) a chance coincidence of shared long-term patterns (e.g. an annual cyclicity), (2) causal links between the wind speed and the "volcanic" (in contrast to "magmatic") gas emission flux, and (3) a systematically wrong calculation of the $SO_2$ emission fluxes. In the following the plausibility of these options is discussed.

(1) The wind speed followed a semi-annual cyclicity with strong maxima in January/February and weaker maxima in July. If the observed correlation is caused by a chance coincidence this would imply an annual cyclicity in the volcanic degassing behaviour with maxima in January/February. Such an annual cycle would be arguably caused by an astronomical forcing. The both best candidates, the solar irradiance and the Earth tidal potential, are indeed at Masaya minimum in December/January and June/July. Nevertheless, it is still far from obvious that these forcings can cause such a strong annual modulation of the $SO_2$ emission flux.

(2) There is indeed a plausible mechanism which links the wind speed and the $SO_2$ emission flux: Volcanic gas emissions often accumulate in the crater of Masaya. The larger the wind speed, the higher is the atmospheric turbulence and thus the lower is the accumulation. Accordingly and if the wind speed is subject to significant short-term fluctuations, over-proportionally much volcanic gas gets effectively released from the volcanic edifice to the atmosphere during high wind speed peaks. However, the observed correlation is based on long-term variations in the wind speeds but not on short-term fluctuations. While the temporal variability of our $SO_2$ time series could be partially caused by this mechanism, our wind data is insensitive for short-term effects and that causal link can be ruled out as a cause for the observed correlation. We highlight that this mechanism may partially explain the high variability in the $SO_2$ emission fluxes as observed by Pering et al. (2019).

(3) There is a number of possibilities how the observed correlation could be caused by systematics in the retrieval of the $SO_2$ emission fluxes: the plume height estimate could systematically depend (a) on the wind speed or (b) on the $SO_2$ emission flux, (c) the retrieval of the background $SO_2$ SCD, or (d) an observational bias caused by the applied filters.

(a) As discussed above, we expected and indeed observed a weak anti-correlation between the plume height and the wind speeds (Figure 8d) which can explain the observed correlation for wind speeds larger than 10 m/s (Figure 8f). We therefore conclude that this mechanism is one of the predominant causes of the observed correlation.

(b) The stronger the absolute volcanic gas emission fluxes (i.e. in particular of $H_2O$), the slower is the cooling of the volcanic plume due to in-mixing of air, and thus the higher is the effective plume height of the buoyant gas plume. Combined with the general expectation that the wind speed is larger with increasing height above ground, we conclude that the higher the $SO_2$ emission flux (when assuming that it is proportional to the absolute gas emission flux), the higher is the wind speed at plume propagation altitude. Using only wind speeds for a fixed altitude level to calculate the $SO_2$ emission fluxes, we can then expect an anti-correlation between the $SO_2$ emission flux estimates and the applied wind speed. The opposite effect has not been observed.

(c) Lübcke et al. (2016) reported for Nevado del Ruiz and Tungaragua that the probability of a background contamination is





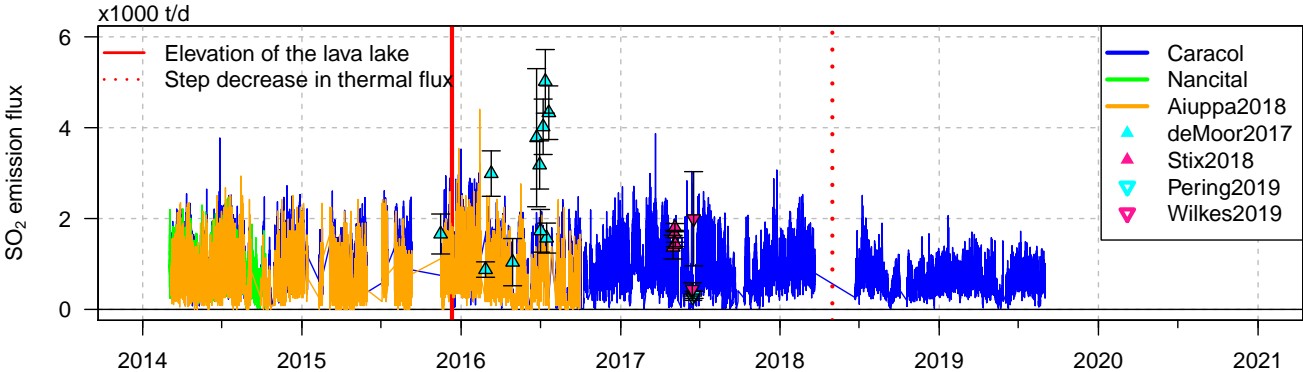

**Figure 11.** Comparison of the SO$_2$ emission fluxes reported in this and other studies.

higher for low wind speeds. Thus, at low wind speeds the SO$_2$ SCD (and hence the SO$_2$ emission flux) is more likely to be underestimated than at high wind speeds. Nevertheless, subtracting the propose absolute background SO$_2$ SCD had hardly an effect on the correlation coefficient indicating that the background contamination has not been the major cause.

(d) The stronger the observed plume shape deviates from an ideal Gaussian shape, the larger is the probability that the scan
gets rejected from the applied data filters. The plume shape is arguably better confined for larger wind speeds because then the relatively short time interval prior to the observation implies a smaller horizontal plume dispersion. Nevertheless, we would neither expect nor some data checks supported that such observation biases could have caused the observed correlation.

**Comparison with reported SO$_2$ emission fluxes**

For 2014–2017, Aiuppa et al. (2018) retrieved from the same NOVAC data and ERA-Interim data mean SO$_2$ emission fluxes of
$(700 \pm 400)\,\mathrm{t\,d^{-1}}$ subject to variations between $0$–$2600\,\mathrm{t\,d^{-1}}$ (Figure 11). Our and their SO$_2$ time series show a good agreement in relative variability but we observed considerably higher values with average relatively factors of $1.42 \pm 0.46$ (Figure 12). This relative factor can be perfectly explained by the combination of the deviations in (1) the SO$_2$ dSCD retrieval, (2) the plume height estimates, and (3) the wind speeds estimates, as detailed in the following.

(1) Aiuppa et al. (2018) used the standard NOVAC SO$_2$ dSCD retrieval whose fit range starts as low as $310\,\mathrm{nm}$. As motivated
above, we argue that they therefore may have underestimated the SO$_2$ dSCDs at Masaya by up to a factor of $1.25$ (or to be more precise: their underestimation relative to our underestimation was up to a factor of $1.25$, see Figure 5a+b).

(2) The different estimates in the plume height explain another relative factor of $\frac{374\,\mathrm{m}}{253\,\mathrm{m}} = 1.48$ (we applied in average a plume altitude of $756\,\mathrm{m}$ a.s.l. implying an average plume height of $374\,\mathrm{m}$ above Caracol station while Aiuppa et al. (2018) applied a constant plume altitude of $635\,\mathrm{m}$ a.s.l. implying a plume height of $253\,\mathrm{m}$ above Caracol station).

(3) Aiuppa et al. (2018) provided their wind data as an upload what allowed a direct comparison with our wind data. They interpolated the ERA-Interim data to the location of the volcano and used only data where the plume propagated in the proximity of the Caracol station (pers. comm. Santiago Arellano, Chalmers University of Technology). The seasonality in their wind speed





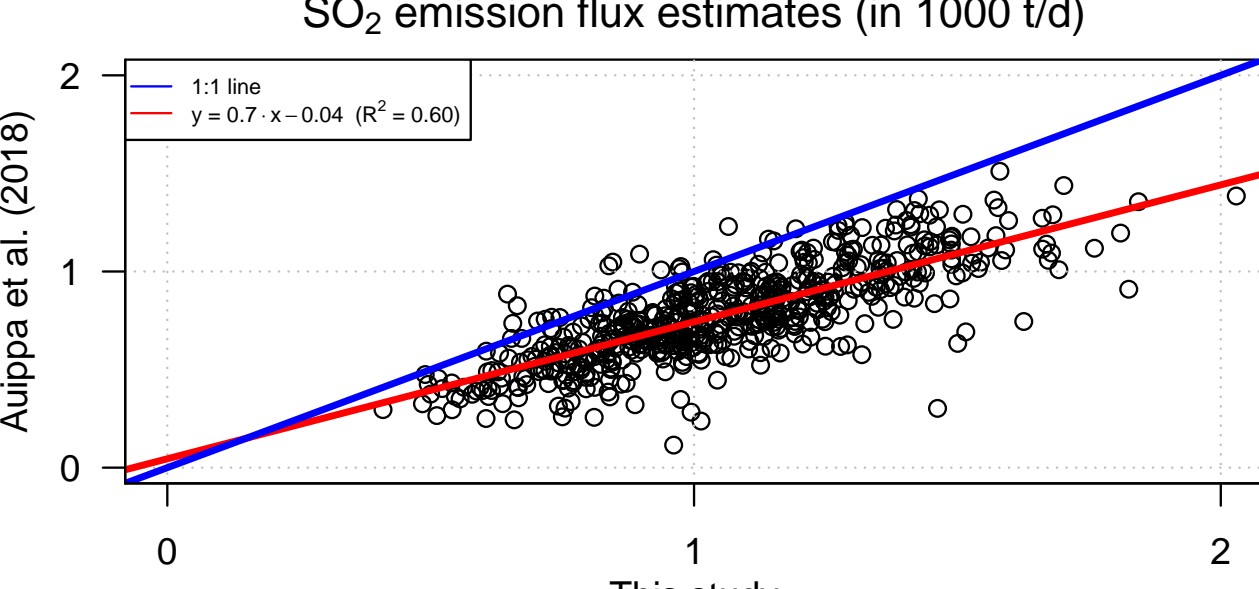

**Figure 12.** Comparison of the $SO_2$ emission fluxes from Caracol station reported in this study and by Aiuppa et al. (2018).

data is in good agreement with our data. The long-term ratio (from March 2014 to October 2016) between their wind speed
data (interpolated to 635 m a.s.l.) and our ERA-Interim data or our operational ECMWF reanalysis data (both interpolated to

700 m a.s.l.) was 1.02 or 1.28, respectively. We remark that in contrast to that actually a factor of less than 1 would be expected
because of their lower retrieval altitude of 635 m a.s.l. instead of our 700 m a.s.l. Following the expectation that the operational
ECMWF reanalysis data are the more accurate estimates, we argue that they overestimated the wind speed in average by more
than a factor 1.28 (see also Figure 8c).

For a complete record, there are further deviations between both retrievals which manifests predominantly in the extended

filtering for unstable measurement conditions in our retrieval (see section 3).

We highlight nevertheless that the conditions at Masaya are rather an exception than the rule. Most NOVAC stations are usu-
ally more than 4 km away from the volcanic edifice and their altitudes are usually more than 1 km below the altitude of the
volcanic summit. In consequence, a given absolute uncertainty in the plume height of, e.g. 100 m, results usually in relative
uncertainties in the plume height of less than 10%. Accordingly, for other volcanoes the uncertainty in the $SO_2$ emission fluxes

may be dominated by other sources of uncertainty. Similar considerations holds for the proposed weak anti-correlation of the
plume height and the wind speed.

Besides the NOVAC long-term time series, the $SO_2$ emission fluxes of Masaya were also determined episodically by short-
term (at most several weeks) measurement campaigns. From 1976–2010, the $SO_2$ emission fluxes varied between $(300 \pm 100)$
and $(2100 \pm 900)\,\mathrm{t\,d^{-1}}$ with all-time averages of roughly $800\,\mathrm{t\,d^{-1}}$ (Nadeau and Williams-Jones, 2009; Martin et al., 2010;





de Moor et al., 2013). Since 2014, $SO_2$ emission fluxes spanning between 1000–5000 t d$^{-1}$ were reported, determined via DOAS traverse measurements (de Moor et al., 2017) or via $SO_2$ camera measurements (Stix et al., 2018; Pering et al., 2019; Wilkes et al., 2019) (Figure 11). Those campaign data matches in general well within our observed range of $SO_2$ emission fluxes, with the exception of most of the June 2016 data from de Moor et al. (2017).

**Critical assessment of our $SO_2$ emission flux retrieval**

This paragraph summarises the extensions implemented in our retrieval as well as a set of possible future advances which have not yet been investigated. Furthermore, the justifications for some retrieval steps introduced in section 2 of this manuscript are motivated. The main findings are summarised in Table 5.

*1. Spectroscopic retrieval:* We documented the possibility for an underestimation of the $SO_2$ dSCDs when the $SO_2$ DOAS fit range is not chosen appropriately (Figure 5a+b). For strongly degassing volcanoes, we recommend to use a fit ranges which

starts at least at 314 nm (see Table 3). Furthermore, we encourage to investigate the possibility of a hybrid retrieval using an interpolation of the dSCDs retrieved from two or more fit ranges. Another source of possible errors can be a missing $I_0$-correction of the absorption cross section of a strongly absorbing gas species. We highlight that, nevertheless, the $I_0$-correction appears to be relevant to reduce the fit errors but usually of negligible importance for the accuracy of the retrieved $SO_2$ dSCD. For instance, even for $SO_2$ dSCDs of about $4 \cdot 10^{18} \frac{molec}{cm2}$ the difference in the retrieved dSCDs was usually less than 1% but the

peak-to-peak range of the residual structures were reduced by about 10–15%. Because precision is quite relevant for the BrO retrieval but not for the $SO_2$ retrieval, we apply the $I_0$-correction routinely to the final data of the BrO/$SO_2$ retrieval but not to the final data of the $SO_2$ flux retrieval. The reason for this is the pragmatic decision to save run time: the effective number of spectra is more than two orders of magnitude lower for the BrO/$SO_2$ retrieval than for the $SO_2$ flux retrieval—and so is the run time required for the $I_0$-correction. Nevertheless, we encourage to use the $I_0$-correction when aiming for a spectroscopically

optimum retrieval.

*2. Filter unstable conditions:* We documented unstable measurement conditions for a significant amount of the scans. We recommend to filter for unstable conditions but our filters should be understood as first proposals. A logical advance would be for instance the additional check via a two-modal Gaussian fit or to apply filter thresholds which more dynamically adjusts to the conditions of the investigated NOVAC station. Another filter whose potential is clearly not yet exhausted is the absolute $SO_2$

background calibration—neither w.r.t. its spectroscopically optimisation nor in the use of its results. Here, we need to highlight that these filters for unstable conditions have been applied only in the $SO_2$ flux retrieval but not transferred to the BrO/$SO_2$ retrieval. The investigation of such a filtering in the BrO/$SO_2$ retrieval is a logical extension of the current retrieval.

*3. Wind conditions:* Lacking measurement data for the wind conditions, the best proxy for wind data are usually weather model data. We compared the wind conditions proposed by the ECMWF ERA-Interim data ($1°$x$1°$ resolution) with operational

ECMWF reanalysis data (up to $0.125°$x$0.125°$ resolution). We documented that the ERA-Interim data proposed for Masaya were in average systematically larger wind speeds with deviations of up to 30% for wind speeds of 20 m/s (or respectively 15 m/s) and wind directions which were $11°$ further to east-northeasterly (in contrast to easterly) than both, the operational ECMWF reanalysis data and the triangulation results. We hesitated, however, to exclusively use the operational ECMWF re-



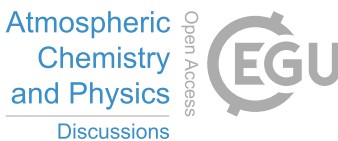

**Table 5.** Applied and possible future advances in the SO$_2$ and BrO analysis. See text for details.

| Status | Description |
| --- | --- |
| **Spectroscopic retrieval** | |
| done | $\lambda_{\text{lower limit}} > 314\,\text{nm}$ required for strong emissions |
| to do | always apply $I_0$-correction?! |
| **Filter unstable conditions** | |
| done | SO$_2$ fluxes: 5 filters summaries in Table 2 |
| | BrO/SO$_2$: 2 filters see section in "3. Methods" |
| to do | further optimise the filter conditions |
| | apply more filters on BrO/SO$_2$ retrieval |
| **Wind conditions** | |
| done | ERA-Interim data ($1°\text{x}1°$) as consistent long-term data |
| | base but calibrated with operational data ($0.14°\text{x}0.14°$) |
| to do | investigate a direct use of operational data |
| **Plume height estimate** | |
| done | plume height retrieved via triangulation |
| | plume height as function of wind speed |
| to do | optimise the triangulation algorithm |
| **SO$_2$ emission flux versus wind speed** | |
| *strong correlation observed for un-calibrated data, not expected!* | |
| done | no more correlation for $> 10\,\text{m/s}$ with our calibrations |
| to do | improve calibration for $< 10\,\text{m/s}$ |
| | establish such checks as benchmark for good estimates |
| **Instrument line function** (only a side note) | |
| done | provide empirical evidence for long-term stability |
| to do | direct retrieval from the recorded spectra |





analysis data due to the frequent jumps in the model set-up. As a cautious compromise, we calibrated the ERA-Interim data
such that they match the operational ECMWF reanalysis data in the long-term average and used these calibrated ERA-Interim
data in all further evaluation steps (see Figure 8b+c). We encourage, nevertheless, a comprehensive investigation of the jumps
in the operational data set-up with the possible finding that an exclusive use of the operational data is the best available proxy
for the wind data.

*4. Plume height estimate:* The triangulation results documented a standard deviation of about 100 m which corresponds to a
relative error of the plume height estimate of 30–40%. As long as no temporally resolved information on the plume height is
available, this has to be seen as a fundamental lowest limit for the uncertainty of the retrieved $SO_2$ emission fluxes at Masaya.
Furthermore, the retrieved mean plume height deviated just 100 m from the plausible best guess used by Aiuppa et al. (2018)
but this deviation in the applied plume height resulted directly in a deviation by a factor of 1.5. While these numbers are ex-
treme for Masaya and presumably less drastic for most other NOVAC volcanoes, it is obvious that the estimate of the plume
height is one of the major intrinsic sources of uncertainty in the $SO_2$ emission flux retrieval. Furthermore, we observed a weak
anti-correlation between the wind speed and the plume height, which is also expected because of the buoyancy of the initially
hot gas plume. Ignoring this relationship could cause a spurious correlation of the $SO_2$ emission fluxes with the wind speed
(see below). We highlight that the applied triangulation algorithm has been rather simple and several advances are desired, e.g.
a filter when both instruments simultaneously see different plumes (see "wings" in Figure 7).

*5. Correlation of $SO_2$ emission flux and wind speed?:* We observed a strong correlation between the original ERA-Interim
wind data and the $SO_2$ emission fluxes when these were calculated without our proposed retrieval advances (82% when all
wind speeds are considered, Figure 8e). This correlation is weaker when our retrieval advances are applied (69% when all wind
speeds are considered) and basically vanishes for wind speeds larger than 10 m/s (then only 19%, Figure 8f). As mentioned
above, this apparent correlation is most likely caused systematics in the $SO_2$ emission flux calculation and namely the igno-
rance of the variations in the plume height. Correlation checks like this should be used to validate under which measurement
conditions the applied assumptions are justified. Considering Figure 8f, we highlight that our proposed retrieval advances were
able to correct this spurious correlation only for high wind speeds larger than 10 m/s.

*6. Instrument line function:* We retrieved the instrument line function (ILF) from a mercury emission spectrum recorded prior
to the installation of the instrument in the field (this is the standard approach for NOVAC data). The ILF varies, however, in
general with temperature and due to ageing and such variations of the ILF could be another major limitation of the accuracy
of gas data from NOVAC (and presumably of most automated measurement platforms). A frequent recording of the ILF could
reduce ILF-related uncertainties, but this is not always feasible on each location. Another approach would the retrieval of the
ILF directly from the recorded spectra. Such retrievals have been developed, e.g. for satellite data (Sun et al., 2017), and are for
example available in the QDOAS software package (http://uv-vis.aeronomie.be/software/QDOAS/). However, as today none
of those retrievals has been optimised for the specifications of NOVAC instruments (i.e. rather low quality of recorded spectra
and no active temperature stabilisation). First steps in this direction have been made by Kleinbek (2020) using the HeiDOAS
software package (currently under development by Udo Frieß, University of Heidelberg). Nevertheless, we highlight that both
instruments enjoyed a surprisingly good long-term stability which may indicate that also their ILFs were rather stable (see





variations of their wavelength-to-pixel calibration in Figure B1). Furthermore, Dinger (2019) investigated the effect of the vari-
ations in the ILF for a NOVAC instrument installed at Nevado del Ruiz. That exemplary study concluded that for the $BrO/SO_2$
molar ratios the ILF-related uncertainties are an order of magnitude smaller than the typical measurement error. While such
exemplary findings can not be adopted directly for other instruments, this has been another hint that the ILF-related effects
may be in reality not as problematic as they could be.

*7. Network design:* The triangulation results gave a rare opportunity to validate the use of weather model data as a proxy for
the meteorological conditions at a volcano. While similar results could be retrieved also by a single NOVAC station using the
optional "flux measurement mode" (Galle et al., 2010), this mode has hardly been used in the past indicating that maintaining
two rather autarkic stations is apparently more likely to happen than actively scheduling the NOVAC measurements. Accord-
ingly, it could be rather beneficial to install two NOVAC stations close to each other in order to retrieve the wind direction and
plume height directly. While there are of course financial and maintenance limitations in adding another station, we highlight
that there is a significant number of NOVAC volcanoes with at least three NOVAC stations where a re-positioning of one of the
stations may be beneficial in the long run. As an even further advance, McGonigle et al. (2005) demonstrated that installing
three instruments in the main plume direction would also allow a direct retrieval of the wind speed.

## 6   Discussion of $SO_2$ and BrO time series

**Correlations between gas data and meteorology**

We investigated the NOVAC data and ERA-Interim data for correlations. For this purpose the daily means of the $SO_2$ dSCDs,
of the BrO dSCDs, of the $BrO/SO_2$ molar ratios, and of the $SO_2$ emissions fluxes and the about noon-time ERA-Interim data
are compared (Figure 13).

A correlation analysis of the ERA-Interim parameters with each other indicates: (1) The barometric pressure is basically not
correlated to any of the other parameters. (2) All remaining ERA-Interim parameters (except the wind direction) are correlated
with the total cloud cover (absolute correlation coefficients between 35–56%). This is presumably mainly a manifestation of
the shared general seasonality of the weather conditions with extrema roughly in March and in October where the total cloud
cover represents the seasonality apparently most clearly. (3) As expected, the three water related parameters (water vapour
concentration, relative humidity, total cloud cover) are strongly correlated and the atmospheric water vapour concentration
correlates with the temperature. (4) The ozone mixing ratio is anti-correlated with the water vapour concentration ($-55\%$),
however, this is presumably first of all the shared seasonality. (5) The wind speed and the wind direction are correlated ($-44\%$).
A correlation of the NOVAC parameters with each other indicates: (1) The variability in the $BrO/SO_2$ time series originates
almost exclusively from the variability in the BrO dSCDs (81%) and not at all from the variability in the $SO_2$ dSCDs ($-15\%$).
(2) The correlation between the $SO_2$ and BrO dSCDs was far from proportional (41%) indicating that these two parameters are
sufficiently independent from each other (i.e. the BrO data is an independent proxy for magmatic or atmospheric processes).
(3) The $SO_2$ emission fluxes were only relatively weakly correlated with the daily average of the $SO_2$ dSCDs in the plume
centre (33%). This can be explained by the two processes which presumably predominantly control the variability in the $SO_2$





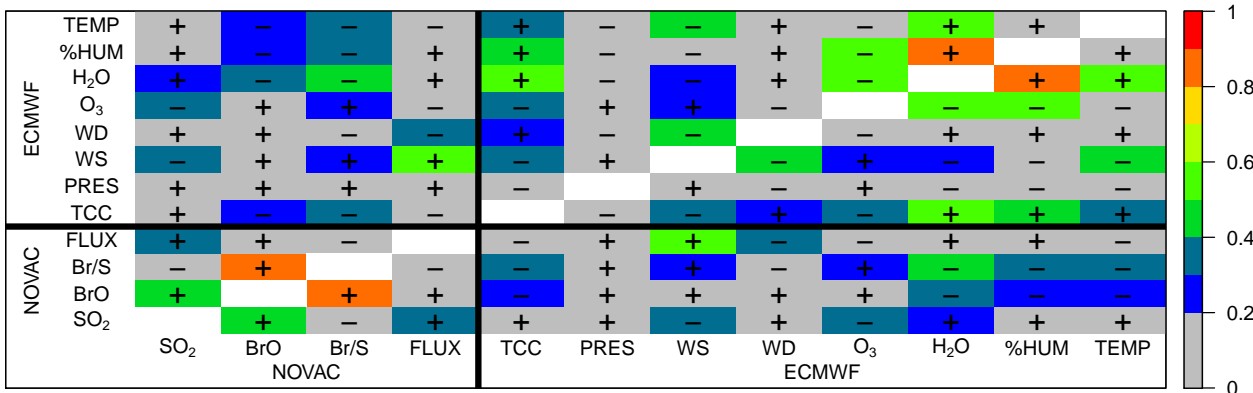

**Figure 13.** Correlation matrix between the different NOVAC parameters and the original ECMWF ERA-Interim parameters. The colour bar indicates the absolute value of the correlation coefficient and the plus and minus signs indicate the sign of the correlation coefficient. The abbreviated parameters are from left to right: daily means of $SO_2$ dSCDs, BrO dSCDs, $BrO/SO_2$ molar ratios, and $SO_2$ emission fluxes, and about noon-time time series of total cloud cover, pressure, wind speed, wind direction, ozone mixing ratio, water vapour concentration, relative humidity, and ambient temperature. The auto-correlation pixels are removed for better readability.

dSCDs: On the one hand, strong long-term variations in the $SO_2$ emission flux should manifest proportionally in the long-term means of the $SO_2$ dSCDs, but on the other hand, the variability of the $SO_2$ dSCDs in the plume centre is also significantly controlled by the horizontal plume dispersion and thus the wind speed (see also Figure 13). Given that the $SO_2$ emission fluxes

of Masaya have been basically constant for several years, the observed absence of such a correlation hints towards the latter reasoning.

A cross correlation between the NOVAC data and the ERA-Interim data indicates two strong correlations: (1) A correlation between the $SO_2$ emission fluxes and the wind speed (57%) and (2) an anti-correlation of the $BrO/SO_2$ molar ratios with the water vapour concentration ($-47\%$). As explained above, this correlation between the $SO_2$ emission fluxes and the wind speed

is most likely predominantly an artefact because in this correlation analysis the original ERA-Interim data have been used for consistency within the comparison of the meteorological data. The correlation is basically vanishing (19%) when the wind speeds are calibrated and only wind speeds larger than $10\,\mathrm{m/s}$ are considered (see Figure 8e+f).

We highlight that the $BrO/SO_2$ molar ratios are at most weakly correlated with the other meteorological parameters (except the water vapour concentration). In particular, the correlation coefficient w.r.t. the wind speed of 25% and w.r.t. the ozone

mixing ratio of 21% were remarkably small. The correlations between the $BrO/SO_2$ molar ratios and the three highlighted meteorological parameters are discussed in the next three paragraphs.





**BrO/SO$_2$ and atmospheric humidity**

The oxidation of bromide ions (Br$^-$) to BrO in a volcanic gas plume is an autocatalytic process, thus it is plausible that the HBr $\rightarrow$ Br$^-$ $\rightleftharpoons$ BrO formation rate in a volcanic gas plume should be positively correlated with the Br$^-$ concentration in the aerosol

phase. A slower BrO formation rate also implies a lower BrO equilibrium level because the equilibrium level of BrO/Br$_{total}$ is reached once the BrO formation rate is equalled by the rate of the BrO destruction mechanisms.

A higher humidity level could cause a smaller Br$^-$ concentration and thus a slower BrO formation rate, as supported by model simulations and experimental results (Rüdiger et al., 2018, and pers. comm. with Stefan Schmitt). As a remark, H$^+$ concentration in the aerosol phase (i.e. its pH-value) should be affected similarly, however, it can be expected that the H$^+$

concentration far exceeds the Br$^-$ concentration and thus this effect on the pH-value is presumably negligible.

The observed anti-correlation between the BrO/SO$_2$ molar ratios and the humidity supports this hypothesis for the rather humid conditions at Masaya. Accordingly, the BrO conversion at Masaya is humidity-limited in summer and autumn when the atmospheric humidity is rather high while this mechanism is much weaker in spring when the atmospheric humidity is in its annual minimum.

**BrO/SO$_2$ and wind conditions**

The NOVAC stations at Masaya were located in close proximity to the volcanic edifice, thus they almost exclusively observed volcanic gas plumes with an atmospheric plume age between 2–8 min (see Figure 14 where the calibrated wind data were used). Furthermore, almost all outliers in the BrO/SO$_2$ time series were associated with plume ages larger than 10 min or with measurements when the plume had allegedly not transacted the scan planes at all.

The BrO/SO$_2$ molar ratio apparently reached a maximum within the first 2 min after the release from the volcanic edifice, decreased to a slightly lower value within the 3rd minute, and remained on this long-term equilibrium level for at least the first 20 min. We highlight that the very young plumes, whose observation proposed the early peak in the BrO/SO$_2$ molar ratio, were observed almost exclusively in spring when by coincidence also the atmospheric humidity is minimum and the wind speeds are maximum (Figure 14b). This early peak may thus not necessarily imply a "fundamental overshoot" in the BrO formation but

could be explained entirely as a manifestation of higher BrO equilibrium level at times of relatively low atmospheric humidity or enhanced ozone in-mixing.

The BrO/SO$_2$ molar ratios were not correlated with the plume altitude for March–October 2014.

**BrO/SO$_2$ and ozone mixing ratio**

The bromide to BrO conversion requires ozone and its destruction is catalysed by BrO. If insufficient atmospheric ozone is

mixed into the volcanic plume, the BrO formation stops. The amount of ozone mixed into the plume depends on the ambient ozone background concentration and on the degree of turbulent mixing. A comparison of the BrO/SO$_2$ data with the ERA-Interim ozone time series (Figure 3) does not allow a detailed investigation of the chemical processes in the volcanic gas plume. Nevertheless, the ERA-Interim data allows an investigation of the Br$^-$ to BrO conversion in the context the temporal variations



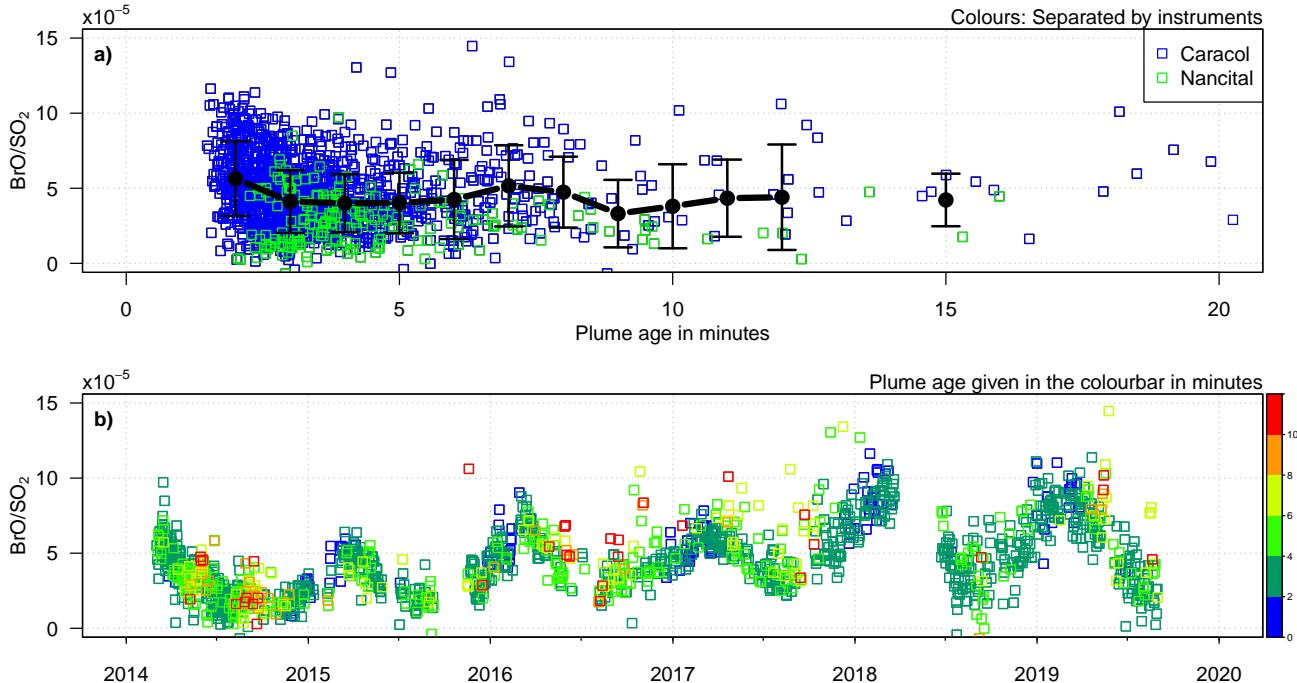

**Figure 14.** BrO/SO$_2$ molar ratios in the gas plume of Masaya depending on the plume age. The wind data imply for some data that the plumes did not transacted the scan planes; these data are excluded from the plot. The red-coloured data span plume age between 10–20 min.

in the general ozone availability.

On the one hand, the observed correlation coefficients between the BrO/SO$_2$ molar ratios and the atmospheric ozone mixing ratio and the wind speeds were both rather small, indicating that the BrO conversion is not predominantly controlled by the background ozone mixing ratio or the air in-mixing rate. On the other hand, the maxima in the BrO/SO$_2$ molar ratios coincide with the observed maxima in the wind speeds as well as in the ozone mixing ratio. Accordingly and despite of the low general correlation coefficient, strong Br$^-$ to BrO conversion rates may be nevertheless only possible for relatively large wind speeds

and/or ozone background concentrations.

**BrO/SO$_2$ and magmatic processes**

The elevation of the lava lake level was most likely caused by the arrival of juvenile (thus gas-rich) magma in the shallow system. Long-term variations in the gas data were thus for the presented time interval most likely linked to the magma dynamics connected to the lava lake.

The elevation of the lava lake level did not result in significant long-term changes in the SO$_2$ emissions fluxes but in a step increase in the BrO/SO$_2$ molar ratios. This change in the gas composition change were thus caused by variations in the volcanic





bromine emissions rather than in the sulphur emissions, either because the juvenile magma had a higher initial Br/S volatile ratio than the older magma or because the older magma was already relatively bromine-poor, where the latter possibility would indicate that bromine degassed earlier than sulphur from the (old) magma at Masaya.

The increasing trend in the BrO/SO$_2$ molar ratios from November 2015 – March 2018 could indicate the general degassing evolution of the hypothetical batch of juvenile magma since its arrival in the shallow magmatic system in late 2015. The increasing BrO/SO$_2$ molar ratios would thus indicate that bromine degasses later than sulphur from this juvenile magma.

The decrease in the lava lake activity in mid 2018 resulted in a significant decrease in the SO$_2$ emission fluxes while the BrO/SO$_2$ molar ratios hardly changed. This decrease of the SO$_2$ emission fluxes indicates a major change in the physico-

chemical conditions of the magma (e.g. in the pressure or temperature at the degassing depth) which is also plausible considering the superficial changes observed. The about constant BrO/SO$_2$ molar ratios would then imply that the bromine and sulphur partitioning from the magmatic melt phase to the magmatic gas phase were independent of the physico-chemical conditions in the magma, at least for this magma and this time.

The decreasing trend since June 2018 could indicate either that the bromine content of the juvenile magma became progres-

sively exhausted (while there were still massive amounts of sulphur solved) or that the relative cooler conditions results in a relatively enhancing bromine solubility in the magma.

## 7 Conclusions

This study contributes to three independent fields of research: a comprehensive discussion of a reliable retrieval of SO$_2$ emission fluxes from ground-based remote sensing data, a dataset for the bromine chemistry in volcanic gas plumes unique in its temporal

coverage and resolution, and an investigation of the BrO/SO$_2$ molar ratio as a proxy for magmatic processes.

**SO$_2$ emission flux retrieval**

An important conclusion of our study is the reminder that calculating reliable SO$_2$ emission fluxes requires a careful investigation of the local conditions. This holds true not only for their accuracy but also for the patterns in the data.

We reported suggestions for the retrieval of SO$_2$ emission fluxes from ground-based remote sensing data and retrieved SO$_2$

emission fluxes which are in average a factor of 1.4 larger than those retrieved by Aiuppa et al. (2018) from the same spectroscopic data. This factor is an accumulation of three major differences between the two retrieval approaches: the SO$_2$ fit range, the wind speed estimate, and the plume height estimate. (1) The different choices of the SO$_2$ fit ranges (our range starts at 314 nm, theirs starts at 310 nm) causes a relative factor of 1.25, indicating their systematic underestimation of the rather strong SO$_2$ SCDs in Masaya's gas plume. (2) Both studies estimated the wind speeds based on ERA-Interim data but we calibrated

those wind speeds to the local conditions by using the higher resolved operational ECMWF reanalysis data. In consequence, our estimates for the wind speeds are in average a factor of 0.8 smaller than theirs. (3) Aiuppa et al. (2018) assumed a plume height fixed at Masaya's summit altitude while we used a dynamic estimate of the plume height based on our triangulation results and the observed weak dependency on the wind speed. In consequence, our estimates for the plume height were in





average a factor of 1.5 larger than theirs.

When it comes to spurious patterns, we observed a strong correlation between the $SO_2$ emission fluxes and the wind speeds when several of our retrieval extensions are not applied (correlation coefficient of 82% when all wind speeds are considered and of 53% for wind speeds larger than 10 m/s). We discussed that no such correlation is expected and that it is most likely an artefact, e.g., due to the assumed fixed plume height. In consequence, the $SO_2$ emission fluxes would then falsely inherit patterns from the variability of the wind speeds and thus conclusions drawn from the variability of the $SO_2$ emissions fluxes

would be only of limited reliability. Using our retrieval, this correlation was reduced in general (69%) and in particular basically vanished for wind speeds larger than 10 m/s (19%). Another conclusion is thus that low wind speeds can result in rather unreliable results.

   We encourage for future publications on $SO_2$ emission fluxes to state detailed information on the used $SO_2$ emission retrieval algorithm. The investigation strategy presented in this study may provide a framework for that task. We nevertheless highlight

the large set of further possible advances which can be still applied and highlight that the choice and setting of the filters may vary significantly for different volcanoes.

**Atmospheric bromine chemistry**

   We observed an extremely significant annual cyclicity in the $BrO/SO_2$ time series. This annual cyclicity is most likely a manifestation of the meteorological seasonality. In particular, an anti-correlation (coefficient of $-47\%$) has been observed

between the $BrO/SO_2$ molar ratios and the atmospheric water vapour concentration. In contrast to that, no clear correlation has been observed between the $BrO/SO_2$ molar ratios and the atmospheric ozone mixing ratio (coefficient of 21%) or the wind speed (coefficient of 25%). A comparison of the $BrO/SO_2$ molar ratio and the atmospheric age of the volcanic plume suggests that the $BrO/SO_2$ reached in the long-term average maximum values within the first 2 min after the release from the volcanic edifice, dropped to a lower level within the 3rd minute, and remained at this level for at least the next 20 min. The apparent

enhancement prior to the 3rd minute could be explained by an observational bias. We conclude that the BrO formation rate at Masaya may be partly controlled by the rather high ambient humidity with higher humidity leading to dilution of the bromide concentration in the aerosol phase, and thus a lower BrO conversion rate.

**Volcanological findings**

   We observed a complementary sensitivity of the $SO_2$ emission fluxes and the $BrO/SO_2$ molar ratios on magmatic processes.

The long-term trend of the $SO_2$ emission fluxes was hardly affected by the initial lava lake level elevation but dropped in mid 2018, when the lava lake activity ceased, to significantly lower $SO_2$ emissions fluxes. In contrast to that, the $BrO/SO_2$ molar ratios doubled due to the lava lake level elevation but showed only a weak response to the reduced lava lake activity since mid 2018. Accordingly, the combination of $SO_2$ emission fluxes and $BrO/SO_2$ molar ratios is highly recommended for monitoring. When corrected for the annual cyclicity, we observed an about linearly increasing trend in the $BrO/SO_2$ molar ratios during the

period for high lava lake activity (November 2015 until March 2018) and an about linearly decreasing trend in the $BrO/SO_2$ molar ratios since May 2018. The isolated interpretation of these observation did not provide clear information on, e.g., the





degassing order of sulphur and bromine of the juvenile magma at Masaya. The provided data may help to double-check and enhance models on the magmatic processes at Masaya.

*Competing interests.* The authors declare that they have no conflict of interest.

*Acknowledgements.* We express our thanks to Santiago Arellano for critically reading our manuscript and for helpful discussions. FD wants to thank Stefan Schmitt (IUP Heidelberg) for fruitful discussions on the bromine chemistry in volcanic gas plumes.

## Appendix A: Absolute calibration of background SCD

We checked for an $SO_2$ contamination of the background by applying the absolute calibration algorithm described by Lübcke et al. (2016). This algorithm performs $SO_2$ DOAS fits where the recorded added-reference-spectrum is used as the measure-
ment spectrum and the solar atlas provided by Chance and Kurucz (2010)—convoluted with the instrument line function—is used as the reference spectrum. Such a fit results in large residual spectroscopic structures because the solar atlas does not contain information on the instrument characteristics. These characteristics can, nevertheless, be determined from the fit residual structures via a principal component analysis applied on a time series of the residual structures. Hereby, it is important that the principal components do not contain structures caused by an interference with the major absorbers in the investigated wave-
length range. Accordingly, we used only those residual structures for the principal component analysis (1) where the retrieved $SO_2$ SCD was smaller than two times the (individual) $SO_2$ fit error, (2) where the solar elevation angle was $> 30°$ to avoid large tropospheric ozone columns, and (3) where the $SO_2$ fit had a $\chi^2 < 0.1$ to avoid potentially problematic spectra. We retrieved a unique fixed first principal component for the total 6 years time series and added it as a pseudo-absorbers to the DOAS fit. This second iteration of the DOAS retrieval gave the absolute $SO_2$ SCD of the added-reference-spectrum (see histograms of these
results in Figure 8a). We highlight that the second principal component for the total 6 years time series explain only 1% of the residual structures and thus adding also this component to the fit scenario would not have improved the spectroscopic retrieval.

## Appendix B: Additional supportive figures

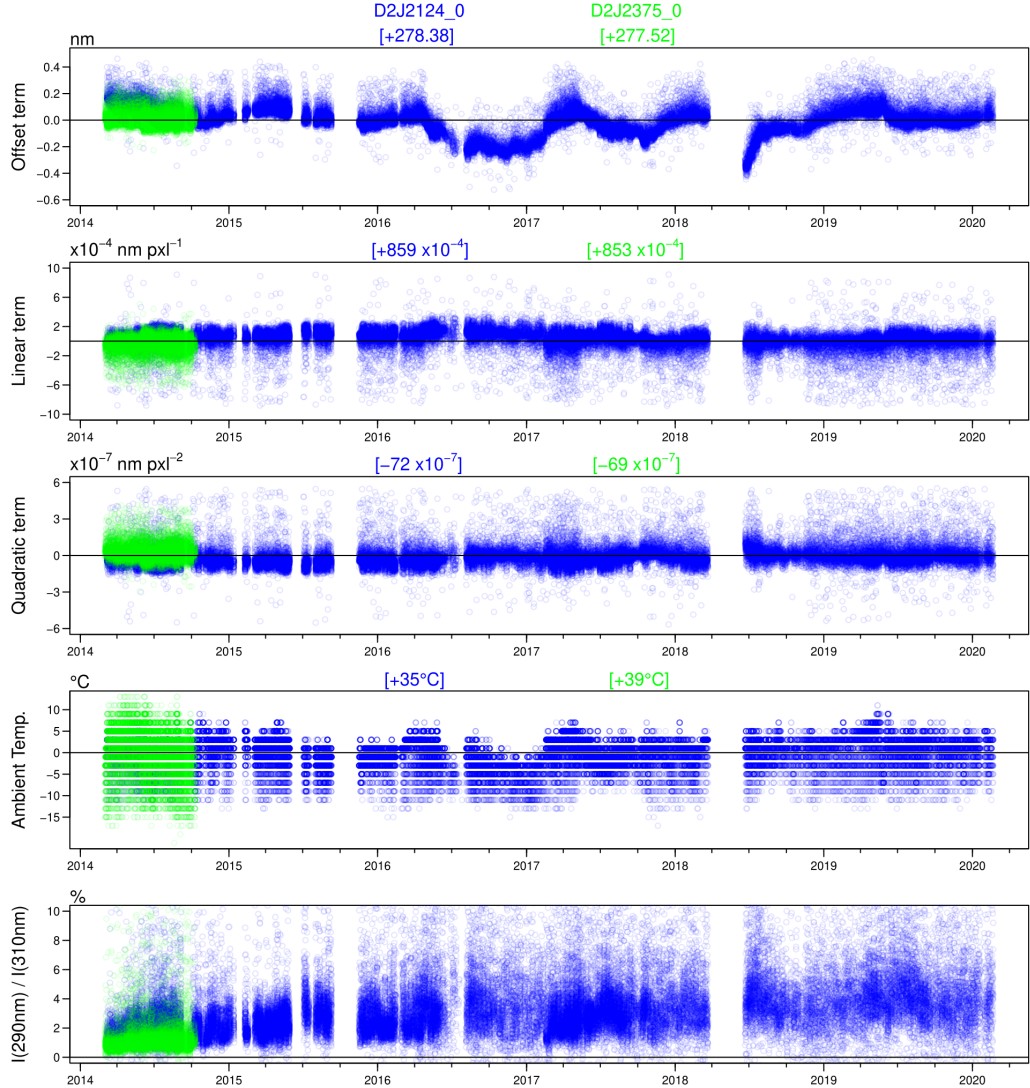

**Figure B1.** Variation of the long-stability of the instruments and ambient temperature. For each parameter, the total values are given by the rough estimate for the mean value (given for each instrument in blue and green above the particular panel) and the variation shown in the plots. The zero lines are chosen arbitrarily and should not be confused with mean values. **a-c)** All spectra have been calibrated by matching their Fraunhofer lines with the Fraunhofer lines of a solar-atlas and the wavelength calibration has been given by a calibration polynomial of 2nd order (see e.g. Dinger et al., 2019, for details). The three panels give for each scan the three coefficients of the wavelength calibration polynomial. The variability is already as displayed rather low and but is actually much lower (most of the indicate scatter is predominantly caused by the first scans in the morning when the temperatures significantly lower than for the rest of the day). **d)** Variation of the ambient temperature. **e)** Ratio of the intensity at around 290 nm and 310 nm (each time average over 10 channels) as a proxy for the magnitude and variation of the stray light.





**Figure B2.** SO$_2$ distribution retrieved **a)** from the scan starting at 2014-03-07 19:16 UTC recorded at Nancital station and **b-d)** from the scan starting at 2014-03-07 19:18, 2014-03-07 19:36, and 2014-03-19 16:25 UTC recorded at Caracol station.

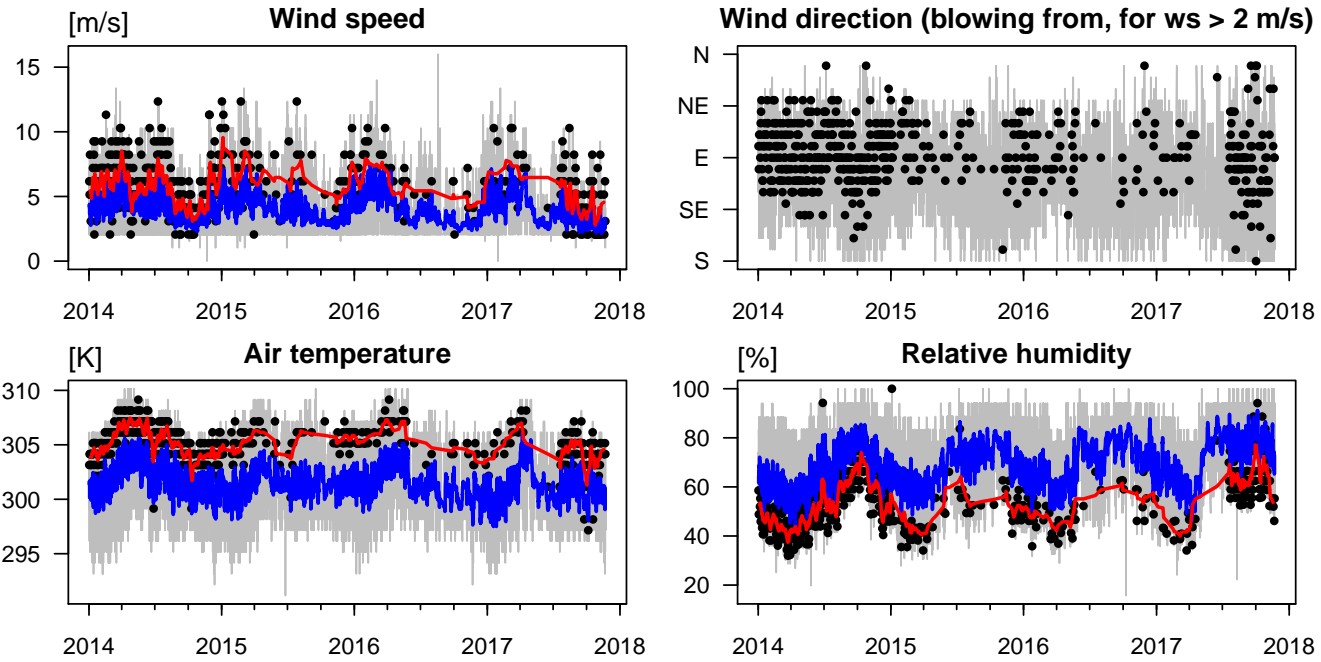

**Figure B3.** Meteorological conditions retrieved from a ground-based station at Managua airport (15 km north of Masaya volcano). **Grey lines:** hourly data. **Blue lines:** sliding average over the horly data (±2 weeks window). **Black dots:** around noon (18:00 UTC) data. **Red lines:** same sliding average but over the around noon data.

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
