# Peer review of "SO2 and BrO emissions of Masaya volcano from 2014–2020"

_Atmospheric Chemistry and Physics, 2020_

## Referee Comment (RC1) · Tom D Pering (Referee) · 2 Nov 2020

**Manuscript Summary**

This manuscript details the changes in $SO_2$ and BrO emissions at Masaya using an unusually long degassing dataset, which in combination with re-analysis meteorology the authors use to investigate the trends and associations of data, whilst being rigorous in their retrieval methodologies.

**General Comments**

This is a well written manuscript with a generally high level of presentation throughout. It is logically structured and is easy to read. My one major comment is on the treatment of statistics in this piece, which reads as a little bit muddled (and confusing in places), with several elements lacking. These elements need to be improved before the manuscript can be published, particularly as some of the conclusions and discussion rely on some of the statements made about trends through time or difference between values, yet differences between times (BrO/$SO_2$ ratios for example at the different phases of the lava lake arrival and the activity) are only stated in a qualitative manner. More statistical rigour is needed. I outline this below in other general comments and in the specific comments.

The abstract is very long. I found it difficult to follow exactly what the key purpose but importantly the major conclusions and discoveries were. Needs shortening. Further comments below.

A very thorough discussion of methodology and sources of error throughout. Substantial and rigorous. Excellent.

A minor thing, I found it difficult that some Figures and Tables were presented before aspects of the main text, which explain some of the formation of the Figures, some rejigging to make sure that this doesn't happen would be great.

Treatment of statistics:

Part 1: Correlation vs. Regression. It is difficult to see what form of analysis has been performed, as frequent reference to 'correlation coefficients' are made, and yet the resultant number is provided as a percentage (with negatives occasionally). Correlation coefficients are presented in the -1 to 1 format (as you do in one point in the manuscript). The use of the percentage here throughout is confusing as we could commonly use the regression coefficient in this manner, i.e., an $R^2$ of 47% (47% of the variation in one variable can be counted for by another). The correlation would be reported as 0.47. This is where the confusion arises. Have you conducted regression and are providing an $R^2$ and calling it a correlation coefficient? Or have you conducted correlation and are formatting it incorrectly. If you were using Pearson's correlation, then the actual correlation value for an $R^2$ of 47% would be ~0.69. This is an important distinction, and it is important that the reader has confidence in the actual statistical technique used – correlation or regression.

Further, regarding regression. Any trend identified can then come with a p-value, is there a significant trend through time? So, where you identify a trend in the manuscript we also need to see the associated p-value to see whether this is the case.

Part 2: In parts of the manuscript where you are comparing significant differences between variables (between different phases for example), you need to statistically test this, i.e., avoiding the qualitative terminology currently used. After determining normality of the data, we can then use a variety of techniques for two-variables (t-test variations / Mann Whitney etc.) and others for three+ variables (Anova / Kruskal-Wallis) dependent on circumstances. This would allow rigorous interpretation of differences and back up the points you make in the manuscript.

Overall, the only major comment being the treatment of statistics I consider that this manuscript would be acceptable after minor revisions. The authors will need to be careful that the results of the additional statistical analysis match with the framing of the discussion.

Reviewed by Dr Tom D. Pering

**Specific Comments**

- Line 19: What is an 'extremely significant' annual cyclicity? Do you mean statistically significant? There are no degrees of 'extremeness' beyond this.
- Line 21: Correlation is not measured as a percentage; it is a standardized set of values between -1 and 1. So what is the -47% signifying? Is this regression?
- Line 57: I would say in situ methods are not able to retrieve bulk gas emissions, suggest removing 'may, however'.
- Line 65: Why are chlorine and fluorine compounds 'obvious candidates'? Needs more detail here.
- Line 77-79: Needs evidence, why is it the best accessible proxy for volcanic processes? References? Examples? The next paragraph (lines 80-95) then goes on to say that interpretation is difficult, so those two sections don't tie in together. Based on the subsequent paragraph a combined DOAS + MultiGas approach would seem far simpler. If by accessible in line 79, do you just mean that you can just use one instrument? If so, tailor that sentence in that manner.
- Line170-175: Where was this data acquired from? Needs a link or detail.
- Line 216: What does 'hardly affected' mean? Is that the 10% figure at the end of the sentence? If so rephrase to use this value. 'Hardly' isn't quantifiable.
- Line 226: Retrieval of the background $SO_2$ slant column. Content fine, but it might be helpful to the reader to summarise the 'four approaches' into a Table.
- Line 302-304. Needs more context, why the 'actually'?
- Line 351: Coefficient – this isn't a correlation coefficient, do you mean 0.89? Latter fit suggests regression? Confusing statistical phrasing. See general comment.
- Figure 8: (d), you highlight a relationship between wind speed and plume height. What is the regression coefficient (the percentage model fit)? What is the p-value? Is it a significant fit? It is also unclear whether the fir is on the grey dots or the black dots.
- Figure 8: (e and f), this is unclear, did I miss in the text why you have split this up into 0-5, 5-10, and 10+ ms blocks [I note that I see this stipulated in text following the Figure but question remains for 5-10]? There appears (not tested) to be abroad relationship between flux and wind speed? So why separated? Needs justification. Same comment regarding statistical terminology.
- Line 375: OK, what does weak mean? What is the regression coefficient ($R^2$ value)? What is the associated p-value? Is this a statistically significant relationship? The scatter plot looks like a smudge of points.
- Line 380: Its only a best guess if you present the model with some statistical rigour, which it is not currently.
- Lines 384-400: Explanation here makes broad sense, but I wonder why you did not use a low flux threshold instead of wind speed? For example, omitting below 0.1 x 1000 t/d?
- Table 4 – Part 1: How is daily variation measured? Is this a standard deviation? Or range? Or iqr? And in each case how is the error determined? Is this 1 standard deviation? Particularly important to clarify.

- Table 4 – Part 2: annual trend and amplitude of cyclicity. How did you measured the trend? Can you have a significant trend of -0.1 with an error the size of the trend itself? P-values? Significance? How did you calculate the amplitude of the cyclicity?
- Line 461 and 462: Correlation coefficients listed here. Confirm that this is indeed what you have.
- Line 470-473: OK, significant variability and different averages. You need to test this statistically, see general comments details.
- Figure 9c: Orange label is for linear trend. But none is indicated? One would expect obvious linear trend in d (your residual plot) therefore. But it isn't obvious. Is this correct? Also how was your annual cycle determined?
- Line 479-481: See general comment on significant difference.
- Line 482: Remove the word 'extremely'.
- Line 486: What does 'basically the same' mean? The values afterwards look different to me.
- Line 490: Significance of the trend?
- Line 493: How did you determine outliers?
- Line 556: Rephrase to remove 'basically vanished'.
- Line 553-558: See general comments on correlation.
- Line 580: see previous comment, anti-correlation needs proof.
- Line 718: What does 'basically not correlated' mean in quantifiable terms?
- Line 715 onwards: Correlation terminology, see general comments.
- Line 787: Reference should probably be made to the Aiuppa paper here (already cited in this manuscript), which talks about this subject exactly.
- Lines 787-806: Interesting analysis, but framing of this will depend on the reassessment of statistics in the manuscript.

**Technical Comments**

- Line 10: Correct 'We make plausible'. Sentence needs shortening and clarification.
- Line 15-18: Shorten, too much distance between the mention of 'former periods' and what those periods are.
- Line253 and 293 and 531 and 744: Don't use 'w.r.t' use with regards to.
- Line 295: 'There is also a significant number of scans'
- Line 299. 'Gaussian distribution' singular.
- Line 571-572: Rephrase needed.
- Line 577 – There *are* a number of possibilities
- Line 718: Remove 'basically'.
- Line 741: Alter phrasing away from 'basically vanishing'
- All graphs, check the 2 is subscripted in $SO_2$.

---

## Referee Comment (RC2) · Christoph Kern (Referee) · 11 Nov 2020

Review

Dinger F, Kleinbek T, Dörner S, Bobrowski N, Platt U, Wagner T, Ibarra M, and Espinoza E, SO2 and BrO emissions of Masaya volcano from 2014–2020, submitted to Atmospheric Chemistry and Physics for discussion.

Summary

This manuscript discusses Differential Optical Absorption Spectroscopy (DOAS) measurements of sulfur dioxide (SO2) and bromine monoxide (BrO) performed at Masaya volcano, Nicaragua. The measurements stem from two monitoring stations of the Network for Observation of Volcanic and Atmospheric Change (NOVAC). The scanning

[Figure]

DOAS instruments at these stations scan the sky from one horizon to the other while collecting scattered solar radiation. When this radiation intersects the volcanic plume, partial absorption occurs at ultraviolet (UV) wavelengths specific to the gas species – in this case SO2 and BrO.

A large part of the manuscript deals with the development of a methodology for consistent analysis of the continuous DOAS data which spans the 2014-2020 time period with only two short interruptions. The authors emphasize the need for applying data quality filters to remove unreliable DOAS scans from the analysis and introduce several corrections for what they deem to be systematic errors in modeled meteorology and measurement geometry parameters.

The resulting time series of SO2 emission rate and BrO/SO2 molar ratio are presented in Figure 9 and appear to show that SO2 emissions averaged around 1000 metric tons per day (t/d) in 2014 - May 2018, then dropped slightly to around 700 t/d. The BrO/SO2 ratio had a significant annual periodicity with relative maxima occurring in March of each year, as well as a general increase in the ratio occurring around the time of the appearance of a lava lake in late 2015. Finally, the implications of these observations are discussed (1) for atmospheric chemistry occurring in the volcanic plume and (2) for volcanic processes occurring at Masaya during the observation period. The authors hypothesize that the seasonal trend of BrO/SO2 may stem from a dilution of bromide in the aerosol phase during more humid times of the year slowing the formation of BrO, and that the drop in SO2 emissions in 2018 was caused by an overall drop in lava lake activity around this time.

General comments

This manuscript provides a wealth of information on methodology for analyzing scanning DOAS measurements of volcanic gas plumes. The authors discuss two different meteorology models (ECMWF ERA-Interim and Operational ECMWF Reanalysis) and compare these with each other and data from a ground-based meteorology station in

Managua. Various methods for filtering and averaging are considered, and finally a correction is developed and applied to the ERA-Interim data. Next, the authors assess methods for retrieving SO2 column densities from the spectral data, developing filters for rejecting data of questionable quality, testing spectroscopic retrievals in multiple wavelength regions, and exploring four different approaches for correcting and/or filtering data associated with contaminated clear-sky reference spectra. The calculation of SO2 emission rates from column densities follows standard methods, but additional filters are then introduced to remove data collected in unstable wind conditions from further consideration. Further testing the reliability of their retrievals, the authors check for and find a significant correlation of SO2 emission rate with wind speed and develop an empirical correction for plume height which appears to improve the quality of their results. Finally, an operational method for retrieving BrO/SO2 ratios representative for each DOAS scan is developed. In the discussion section, all of these topics are again brought up, this time in the context of a critical assessment of each retrieval step and a detailed discussion of potential error sources. All of these topics are well-motivated, described in detail, and appear to be robust, therefore providing a valuable resource for researchers analyzing scanning DOAS data.

At the same time, the other aspects of this manuscript are lacking detail in my opinion, especially the purported link between the measurement results and volcanic processes. Cited extensively throughout this manuscript, a study by Aiuppa et al (2018) developed a fairly detailed conceptual model of degassing at Masaya specifically for the time period examined here, yet there is almost no mention of their conclusions (other than to say that the re-analysis performed here yielded overall higher SO2 emission rates). The authors should consider a much more thorough discussion of this existing degassing model, noting where new information might be added based on their measurements, and work out the volcanological implications of their results compared to those presented in this previous work.

Overall, I wonder whether Atmospheric Chemistry and Physics is the best venue for

dissemination of this manuscript. In its current form, readers primarily interested in the reactive halogen chemistry or volcanic degassing processes will likely find it difficult to identify the information relevant to them. Sections dealing with development, testing, and refinement of measurement strategies make up approximately 2/3 of the manuscript, while the presentation and discussion of results is fairly minor in comparison. The manuscript is already quite lengthy, so this situation might be mitigated by moving significant portions of the methodology to a supplement where those interested in these aspects could find all the details, while expanding on the sections dealing with atmospheric processes and especially volcanology. However, another option might be to move this manuscript to a technical journal such as Atmospheric Measurement Techniques. In this case, the technical information could remain in the body of the manuscript, and less details would be expected in the discussion of atmospheric chemistry and volcanic processes.

Regardless of the path chosen by the authors, I am highly supportive of the content being published for use by the scientific community. In the following section and an attached annotated PDF, I list specific comments which I hope will help the authors in making improvements to their manuscript.

Specific Issues

Abstract - Pending the decision on how to proceed with the publication of this manuscript, the abstract should be revised and shortened to highlight only the key aspects of the study.

L199 – Please clarify how the quality filter based on spectral intensity works. Does the filter only consider the SO2 fit region when rejecting over- or underexposed spectra? Or does it consider the entire spectrum? It would seem to me that spectra could still be used if they are not overexposed in the fit region. Also, I'd be concerned that removing individual spectra from a scan based on their intensity might lead to preferential removal of plume spectra. In my experience, plume spectra often appear brighter than

background sky spectra during clear-sky conditions due to the increased scattering of solar radiation on aerosols in the plume. This will often cause the plume to appear brighter than the sky. The result of this filter could therefore be a removal of some or all the spectra associated with the plume itself. Of course spectra that are oversaturated in the fit region cannot be evaluated, but in such a case the entire scan would need to be discarded, not just the oversaturated plume spectra.

L380 – It didn't become clear to me whether the triangulated plume height was used for the retrieval of SO2 emission rates in cases where it could be determined, i.e. when both scanners detected the plume. Regardless of what was done, one interesting test would be to restrict the dataset to only those emission rates derived using triangulated plume heights. For this subset, does the correlation of emission rate with wind speed disappear? If this correlation really is an artefact of assuming incorrect plume heights, wouldn't we expect it to disappear in cases where the plume height can be measured reliably?

L455 and throughout – I had a hard time understanding the descriptions of volcanic activity presented by the authors. Descriptions such as 'elevation of the lava lake' seem a bit misleading. Aiuppa et al (2018) reported that there was not a lava lake at Masaya prior to December 2015. So do you mean 'appearance of a lava lake' here? I recommend that, when describing volcanic processes, the nomenclature from Aiuppa et al (2018) and other previous studies be adapted as much as possible and those studies be referenced so that it's clear you are referring to the same events. Here, I also don't understand what is meant by 'actual onset of activity in the shallow magma system'. What type of activity? What do you consider the shallow magma system? Do you mean the shallow magma reservoir identified by Rymer et al. (1998) and Williams-Jones et al. (2003)? See Aiuppa et al (2018) for more details on this.

Table 5 – In this section of the manuscript and the accompanying table, a number of suggestions are made for further improving upon the framework utilized by the authors to evaluate the scanning DOAS data. It's understood that there is always room for

improvement, but since one of the central points of this study is to discuss the ideal methodology for analyzing the DOAS data, it seems a bit contradictory to make so many suggestions for improvement beyond what is recommended here. And in many cases, particularly in the table, the suggestions either seem to contradict earlier statements or are too vague for the readers to act upon. For example, I understood that the I0 correction did not improve the SO2 retrieval noticeably. Why then is it recommended that this should always be applied? Similarly, suggestions like 'further optimize filters', 'apply more filters', 'optimize triangulation algorithm', or 'improve calibration' don't provide actionable information. I recommend that this section either be carefully revised such that only specific, actionable suggestions are given, or that this section be removed entirely and these suggestions for possible future investigation simply be given in the methods section.

L782 – The lack of correlation between BrO/SO2 and background ozone concentrations or wind speed is interesting. Others (myself included) have suggested that in-mixing of atmospheric oxidants could be a relevant process limiting BrO formation. For example, this might explain the spatial heterogeneity of BrO/SO2 we observed at Mount Pagan volcano (Kern and Lyons 2018). Here, you write that "The observed correlation coefficients between the BrO/SO2 molar ratios and the ozone mixing ratio and the wind speeds were both rather small, indicating that the BrO conversion is not predominantly controlled by the background ozone mixing ratio or the air in-mixing rate." However, I wonder if in-mixing might be more important than it may seem here. As you point out, the in-mixing rate would likely depend on wind speed, with higher wind speeds leading to greater turbulence and more efficient mixing. But at the same time, a higher wind speed would also reduce the plume transport time from the vent to the DOAS scanning plane, so the measurements would be made in a younger plume. The measurements suggest that the BrO/SO2 ratio is approximately equal, regardless of wind speed. Is it therefore possible that the plume is indeed evolving more quickly (with faster halogen activation) in the high wind / efficient mixing scenario, and that this is the reason that the BrO/SO2 ratio is approximately the same despite the plume being younger?

L786ff – This section needs major revisions in my opinion. As I stated in the general comments above, I strongly recommend the authors take a closer look at the literature available on Masaya's volcanic system and activity. The very first sentence in this section claims that the elevation (= appearance?) of the lava lake was likely caused by the arrival of juvenile magma in the shallow system. How do you know? I don't see how this follows from the measurements presented in the study – in fact I think you could argue the opposite could be true given the observation that the SO2 emission rate hardly increased in this period. At the same time, Aiuppa et al (2018) make some relatively convincing arguments for a shallow intrusion which may have caused a rejuvenation of pre-existing shallow magma. If this is what you are arguing, then please be sure to cite their paper as well as the original references upon which their observations rely. When citing previous work, please also be sure to use the original nomenclature when describing parts of the volcano's plumbing system or types of activity. For example, please clarify what is meant by "shallow system". Other concerns I have for this section include:

- You distinguish between juvenile magma and older magma. Where is the juvenile, where is the older magma? Are they moving up or down? Mixing? Convecting?

- It's not clear that bromine and sulfur would degas at the same depth/pressure. Therefore, it's not clear to me that bromine would have degassed earlier than sulfur from old magma. Isn't it possible that sulfur (and additional brome) are being supplied from juvenile magma and being added to the gas emitted from the older magma?

- Rather than speaking of earlier or later degassing, maybe it's best to describe things in terms of pressure or depth?

- What was the observed 'decrease in lava lake activity in mid 2018? What exactly decreased? How was this determined? I assume this observation is also taken from other studies? Please be sure to cite them.

- Couldn't the decrease in the SO2 emissions simply be caused by depletion of sulfur

in the magma? It's not clear to me why it requires a change in pressure or temperature at degassing depth.

- I don't think the lack of change in BrO/SO2 in 2018 implies that Br and S partitioning are independent of the physio-chemical conditions in the magma. I guess it's also possible that changing conditions combined with varying degrees of depletion of these elements in the magma could lead to a relatively stable ratio. (such an example is actually given in the next sentence).

Overall, I think this section would benefit greatly if the discussion occurred within the framework of a conceptual model for the degassing behavior. Aiuppa et al. (2018) actually present such a model, and this model could be used if desired. In that case, it would be interesting to highlight which information this study adds to the existing framework, which (if any) new observations do not fit well into the exiting model, and which new insights (beyond those already discussed by others previously) can be gleamed from these measurements.

Minor Corrections

Please see the attached annotated PDF for additional comments, suggestions, and minor corrections to the text. I'd like to thank the authors and journal editors for the opportunity to review this manuscript.

Please also note the supplement to this comment:
https://acp.copernicus.org/preprints/acp-2020-942/acp-2020-942-RC2-supplement.pdf

**Supplement:**

[revised manuscript text omitted]

---

## Author Comment (AC1) · 21 Mar 2021

We thank Tom Pering for his comprehensive review of our manuscript "SO$_2$ and BrO emissions of Masaya volcano from 2014–2020". In particular, we welcome his critical checks on our statistical methods and results. In the following, we reply on his specific comments paragraph-wise. If not stated differently, the line numbers and figure numbers refer to the originally submitted manuscript.

**1 General Comments**

**Tom Pering, *Manuscript Summary*:**
**This manuscript details the changes in SO2 and BrO emissions at Masaya using an unusually long degassing dataset, which in combination with re-analysis meteorology the authors use to investigate the trends and associations of data, whilst being rigorous in their retrieval methodologies.**

**Tom Pering, *General Comments* (general part 1):**
**This is a well written manuscript with a generally high level of presentation throughout. It is logically structured and is easy to read. My one major comment is on the treatment of statistics in this piece, which reads as a little bit muddled (and confusing in places), with several elements lacking. These elements need to be improved before the manuscript can be published, particularly as some of the conclusions and discussion rely on some of the statements made about trends through time or difference between values, yet differences between times (BrO/SO2 ratios for example at the different phases of the lava lake arrival and the activity) are only stated in a qualitative manner. More statistical rigour is needed. I outline this below in other general comments and in the specific comments.**
We welcome the reviewer's focus on statistical rigour. Several of his recommendations helped to make our statements more comprehensible. Nevertheless, there might be misunderstanding on the side of the reviewer regarding our notation or methods. In particular, we consider our statistical analysis sufficiently comprehensive as we will argue in detail below.

**Tom Pering, *General Comments* (general part 2):**
**The abstract is very long. I found it difficult to follow exactly what the key purpose but importantly the major conclusions and discoveries were. Needs**

**shortening. Further comments below.**
**Change:** We shortened the abstract by about 25 %. In particular, we removed most of the qualitative volcanological interpretations because these have not been the major conclusions of this study.

**Tom Pering,** *General Comments* **(general part 3):**
**A very thorough discussion of methodology and sources of error throughout. Substantial and rigorous. Excellent.**
We are glad that this is appreciated.

**Tom Pering,** *General Comments* **(general part 4):**
**A minor thing, I found it difficult that some Figures and Tables were presented before aspects of the main text, which explain some of the formation of the Figures, some rejigging to make sure that this doesn't happen would be great.**
Our original manuscript was prepared and optimised for the final two-column format of ACP (including the correct positions of the figures). We will take care of the correct placing of figures and tables relative to the text in the final type-setting process.

**Tom Pering,** *General Comments* **(Treatment of statistics, Part 1):**
**Correlation vs. Regression. It is difficult to see what form of analysis has been performed, as frequent reference to 'correlation coefficients' are made, and yet the resultant number is provided as a percentage (with negatives occasionally). Correlation coefficients are presented in the -1 to 1 format (as you do in one point in the manuscript). The use of the percentage here throughout is confusing as we could commonly use the regression coefficient in this manner, i.e., an $R^2$ of 47% (47% of the variation in one variable can be counted for by another). The correlation would be reported as 0.47. This is where the confusion arises. Have you conducted regression and are providing an R2 and calling it a**

**correlation coefficient? Or have you conducted correlation and are formatting it incorrectly. If you were using Pearson's correlation, then the actual correlation value for an R$^2$ of 47% would be 0.69. This is an important distinction, and it is important that the reader has confidence in the actual statistical technique used — correlation or regression.**

Our statistical analysis was based almost exclusively on Pearson correlation co-efficients (the $R^2$-values in Figures 5 and 12 — Figures 11 and B1 in the revised manuscript — being the only exceptions). We consistently called these coefficients "correlation coefficients" and never used the term "regression". Nevertheless, we agree with the reviewer that the notation would be less ambiguous when the correlation coefficients are stated as pure decimal numbers (nonetheless because we use the "%" exhaustively elsewhere in the manuscript for "amounts/fractions of data").

**Change:** We formatted all correlation coefficients presented in the manuscript in the -1 to +1 format, where we explicitly added also the "+"-sign in order to minimise further ambiguity.

**Tom Pering, *General Comments* (Treatment of statistics, second paragraph):**
**Further, regarding regression. Any trend identified can then come with a p-value, is there a significant trend through time? So, where you identify a trend in the manuscript we also need to see the associated p-value to see whether this is the case.**

All reported fits came with total p-values $< 2.22 \cdot 10^{-16}$ (i.e. the machine epsilon of the used computer). The p-values for most regressors were as well $< 2.22 \cdot 10^{-16}$.

On the specific question on a trend through time: We separated our time series in three time intervals (motivated also by the general volcanological observations) and retrieved and reported different trends (including standard errors for the linear trend regressor) for those adjacent time intervals. The purpose was to identify differences between those three time intervals. For the sake of clearness, we did not investigated further trend constituents (beyond a linear and a sinusoidal term) within the specific

time intervals.
**Change:** We added the p-values to all statement of fits.

**Tom Pering, *General Comments* (Treatment of statistics, Part 2):**
**In parts of the manuscript where you are comparing significant differences between variables (between different phases for example), you need to statistically test this, i.e., avoiding the qualitative terminology currently used. After determining normality of the data, we can then use a variety of techniques for two-variables (t-test variations / Mann Whitney etc.) and others for three+ variables (Anova / Kruskal-Wallis) dependent on circumstances. This would allow rigorous interpretation of differences and back up the points you make in the manuscript.**
A general remark to our statistical analysis and reporting of the statistical result:
We applied linear regression analyses (based on ordinary least squares) to estimate models constituting, e.g., of a linear trend parameter and a sinusoidal part. (The other statistical method applied was the Lomb-Scargle periodicity analysis.) For those linear regressions, we have a large set of statistical test results available, e.g., the standard error, the t-value, the p-value for each regressor, the F-statistics etc. We decided to report the estimated value $x$ and its standard error $e$, that is $(x \pm e)$, which we consider best practise. In particular, we consider the reported format $(x \pm e)$ as a comprehensive notation of a t-test (where the reader can easily derive the explicit t-value).

**Tom Pering, *General Comments* (last paragraph):**
**Overall, the only major comment being the treatment of statistics I consider that this manuscript would be acceptable after minor revisions. The authors will need to be careful that the results of the additional statistical analysis match with the framing of the discussion.**
The analysis has already been carried out with rigour on the statistical significance.

Adjusting the notation of the statistics will thus not change the framing of the discussion.

**2 Specific Issues**

**Tom Pering, *Specific issues, Line 19*: What is an 'extremely significant' annual cyclicity? Do you mean statistically significant? There are no degrees of 'extremeness' beyond this.**
We aimed to highlight that the confidence in this observation is basically 1 (false alarm probability of $9 \cdot 10^{-74}$, see Line 484).
**Change:** We changed from *extremely significant annual cyclicity* to *annual cyclicity*, i.e. omitting on purpose also the redundant word "significant".

**Tom Pering, *Specific issues, Line 21*: Correlation is not measured as a percentage; it is a standardized set of values between -1 and 1. So what is the -47% signifying? Is this regression?**
It is the Pearson correlation coefficient.
**Change:** See our reply on (Treatment of statistics, Part 1) above. (Specifically, we replaced $-47\,\%$ by $-0.47$.)

**Tom Pering, *Specific issues, Line 57*: I would say in situ methods are not able to retrieve bulk gas emissions, suggest removing 'may, however'.**
**Change:** We changed the text in the revised manuscript as recommended.

**Tom Pering, *Specific issues, Line 65*: Why are chlorine and fluorine compounds 'obvious candidates'? Needs more detail here.**
**Change:** We now state in the revised manuscript: *Other obvious candidates are chlorine and fluorine compounds due to their relatively high abundance*, which we think are obvious reasons.

**Tom Pering, *Specific issues, Line 77-79*: Needs evidence, why is it the best accessible proxy for volcanic processes? References? Examples? The**

**next paragraph (lines 80-95) then goes on to say that interpretation is difficult, so those two sections don't tie in together. Based on the subsequent paragraph a combined DOAS + MultiGas approach would seem far simpler. If by accessible in line 79, do you just mean that you can just use one instrument? If so, tailor that sentence in that manner.**

We never assumed that the $BrO/SO_2$ molar ratio is per se the "best accessible" proxy for volcanic processes (in fact, we made the potential problems with $BrO/SO_2$ analysis transparent in the next paragraphs) but we stated that the database for this gas proxy is most likely already the second largest just behind $SO_2$ emission flux data. Furthermore, we emphasise that this is also a direct consequence of the fact that the $BrO/SO_2$ molar ratios can be derived rather cost-efficient via remote-sensing (a lot of data for minimum invested resources). Accordingly, we have to disagree that a DOAS + MultiGas approach is simpler as this adds another instrument (the MultiGas) which brings in addition the potential problems of in-situ methods discussed further above in the manuscript.

**Change:** We now state in the revised manuscript: *In consequence, although BrO is not on the list of the most desired plume constituent species, time series of the $BrO/SO_2$ molar ratios in volcanic gas plumes are the easiest accessible remote-sensing gas proxy for volcanic processes so far (besides the $SO_2$ emission fluxes).*

**Tom Pering,** *Specific issues, Line 170-175*: **Where was this data acquired from? Needs a link or detail.**

The data sources was given in the original manuscript in Line 128/129 (same paragraph as for the ECMWF data). In the revised manuscript, the statement is now given in the Appendix A in Line 903/04.

**Tom Pering,** *Specific issues, Line 216*: **What does 'hardly affected' mean? Is that the 10% figure at the end of the sentence? If so rephrase to use this value. 'Hardly' isn't quantifiable.**

"Hardly" referred to an underestimation of 3% which we consider — while systematic and thus not "insignificant" — minor to the scatter of this comparison (see Figure 5b — Figure B1b in the revised manuscript).

**Change:** We specified in the revised version of the manuscript: *to be hardly affected by saturation effects up to SO$_2$ SCDs of $1 \cdot 10^{18} \frac{molec}{cm^2}$ (3% underestimation, see Figure B1b) and still of acceptable accuracy at SO$_2$ SCDs of $3 \cdot 10^{18} \frac{molec}{cm^2}$ (9% underestimation).*

**Tom Pering, *Specific issues, Line 226*: Retrieval of the background SO2 slant column. Content fine, but it might be helpful to the reader to summarise the 'four approaches' into a Table.**
**Change:** We added Table 4 in the revised manuscript.

**Tom Pering, *Specific issues, Line 302-304*: Needs more context, why the 'actually'?**
In the sentence above we introduced that we used 'a' Gaussian distribution. But 'actually' we used two.
**Change:** We now state in the revised manuscript: *In order to provide an automated test of the "Gaussian shape assumption", we fitted two Gaussian distributions to the SO$_2$ VCD distribution, one with a fixed $b = 0$ and one with a free $b$*

**Tom Pering, *Specific issues, Line 351*: Coefficient — this isn't a correlation coefficient, do you mean 0.89? Latter fit suggests regression? Confusing statistical phrasing. See general comment.**
We confirm that this is a Pearson correlation coefficient.
**Change:** See our reply on (Treatment of statistics, Part 1).

**Tom Pering, *Specific issues, Figure 8 (d)*: you highlight a relationship between wind speed and plume height. What is the regression coefficient (the**

**percentage model fit)? What is the p-value? Is it a significant fit? It is also unclear whether the fir is on the grey dots or the black dots.**

The fit was based on the black dots. The Pearson correlation coefficient is $-0.25$ (or $R^2 = 0.06$ and p-value of $2.3 \cdot 10^{-10}$). The fit is thus significant but superposed by a much stronger scatter.

**Change:** We rephrased the text, see our reply on the next but one reviewer's comment.

**Tom Pering,** *Specific issues, Figure 8 (e and f)***: this is unclear, did I miss in the text why you have split this up into 0-5, 5-10, and 10+ ms blocks [I note that I see this stipulated in text following the Figure but question remains for 5-10]? There appears (not tested) to be abroad relationship between flux and wind speed? So why separated? Needs justification. Same comment regarding statistical terminology.**

We tested the correlation of the calculated $SO_2$ emission fluxes and the wind speed for three wind speed regimes. The wind speed intervals were chosen arbitrarily yet comprehensively as regimes of low, intermediate, and high wind speeds (0–5, 5–10, 10+ m/s). We don't understand the reviewer's comment on the "broad relationship between flux and wind speed" — this finding was discussed exhaustively in the manuscript.

**Tom Pering,** *Specific issues, Line 375***: OK, what does weak mean? What is the regression coefficient (R2 value)? What is the associated p-value? Is this a statistically significant relationship? The scatter plot looks like a smudge of points.**

The regression coefficient is $R^2 = 0.06$, thus the 'weak' anti-correlation coefficient is $-0.25$, and the p-value was $2.3 \cdot 10^{-10}$.

**Change:** We rephrased and extended the paragraph in the revised manuscript:

*The comparison of the triangulated plume height with the wind speed (calibrated as explained above) confirmed such a causal link between the plume height and the wind*

*speed (correlation coefficient of $-0.28$ when considering all wind speeds and of $-0.25$ when considering only wind speeds larger than 5 m/s). We retrieved for the linear relationship of $H_s + A_s = a_0 - a_1 \cdot v_{calibrated}$ a best fit (when $H_s$ and $A_s$ measured in m and $v_{calibrated}$ measured in m/s) for $a_0 = (902 \pm 12)\,m$ and $a_1 = (12.2 \pm 1.5)\,s$ (when all wind speeds were considered, F-statistics of 64.7, p-value = $3.2 \cdot 10^{-15}$) or $a_0 = (909 \pm 18)\,m$ and $a_1 = (13.1 \pm 2.0)\,s$ (when only wind speeds larger than 5 m/s were considered, F-statistics of 41.6, p-value = $2.3 \cdot 10^{-10}$).*

*As a remark, we retrieved similarly well matching fits also for a quadratic relationship of $H_s + A_s = a_0 - a_1 \cdot (v_{calibrated})^2$ with a best fit for $a_0 = (860 \pm 8)\,m$ and $a_1 = (7.7 \pm 1.0) \cdot 10^{-4}\,s^2/m$ (when all wind speeds were considered, F-statistics of 64.8, p-value = $3.1 \cdot 10^{-15}$) and $a_0 = (850 \pm 10)\,m$ and $a_1 = (6.8 \pm 1.1) \cdot 10^{-4}\,s^2/m$ (when only wind speeds larger than 5 m/s were considered, F-statistics of 38.9, p-value = $8.3 \cdot 10^{-10}$).*

*We chose to use the linear relationship retrieved for winds speeds larger than 5 m/s for dynamic estimates of the plume height as a function of the wind speed, i.e. we applied a $H_s$ retrieved via $H_s + A_s = 909\,m - 13.1\,s \cdot v_{calibrated}$ as the estimate for the plume height in the calculation of the $SO_2$ emission fluxes.*

**Tom Pering,** *Specific issues, Line 380***: Its only a best guess if you present the model with some statistical rigour, which it is not currently.**
This "best guess" was the significant result of the linear regression described above. The F-statistics of those tests are now given in the text.
**Change:** See our reply on the previous reviewer comment.

**Tom Pering,** *Specific issues, Line 384-400***: Explanation here makes broad sense, but I wonder why you did not use a low flux threshold instead of wind speed? For example, omitting below 0.1 x 1000 t/d?**
We argued that low wind speeds can cause a significantly wrong estimation of the $SO_2$ flux — with the amount of deviation being possibly independent of the $SO_2$ degassing

strength. Accordingly, the interpretation of low wind speeds should be avoided and not the interpretation of low $SO_2$ emission fluxes.

**Tom Pering,** *Specific issues, Table 4 — Part 1*: **How is daily variation measured? Is this a standard deviation? Or range? Or iqr? And in each case how is the error determined? Is this 1 standard deviation? Particularly important to clarify.**
The daily variations are based on the standard deviations of the individual days. All given errors are standard deviations.
**Change:** We state now in the caption of Table 5: *Main statistical properties of the spectroscopic results for Caracol station. Early BrO/SO$_2$ NOVAC observations between 2007–2009 are listed for completeness. The daily variations are based on the standard deviations of the single days. The given errors are standard deviations, except for the annual trend and the amplitude of the annual cycle for the BrO/SO$_2$ molar ratios were the errors refer to the standard regression error.*

**Tom Pering,** *Specific issues, Table 4 — Part 2*: **annual trend and amplitude of cyclicity. How did you measured the trend? Can you have a significant trend of -0.1 with an error the size of the trend itself? P-values? Significance? How did you calculate the amplitude of the cyclicity?**
The trend and the amplitude were determined simultaneously by a linear regression (see Figure 9c — Figure 8c in the revised manuscript). We consider a trend of $(-0.1 \pm 0.2)$ as not significant (based, e.g., on the results a t-test). We see this statistical finding as equally important as the significant trends for the other periods.
**Change:** We added the standard errors of the amplitude (as derived via the linear regression), and removed the $\pm$ sign. Remark: we had added the $\pm$ in order to clarify that this is a proper amplitude and not a peak-to-peak value but this should be clear already by the term "amplitude".

**Tom Pering,** *Specific issues, Line 461-462*: **Correlation coefficients listed here. Confirm that this is indeed what you have.**
Confirmed.

**Tom Pering,** *Specific issues, Line 470-473*: **OK, significant variability and different averages. You need to test this statistically, see general comments details.**
See our reply on (Treatment of statistics, Part 2).

**Tom Pering,** *Specific issues, Figure 9c*: **Orange label is for linear trend. But none is indicated? One would expect obvious linear trend in d (your residual plot) therefore. But it isn't obvious. Is this correct? Also how was your annual cycle determined?**
The orange lines give the results of the model $y(t) + y_0 + \alpha \cdot t + \beta \cdot \sin(2 \cdot \pi \cdot t/(364.24 days) + \gamma)$. The regressors for the linear trends are given in the text. We did not plot the linear trend by a separate line, in order to keep the plot more tidy.

**Tom Pering,** *Specific issues, Line 479-481*: **See general comment on significant difference.**
See our reply on (Treatment of statistics, Part2).

**Tom Pering,** *Specific issues, Line 482*: **Remove the word "extremely".**
**Change:** We changed the text in the revised manuscript as recommended.

**Tom Pering,** *Specific issues, Line 486*: **What does 'basically the same' mean? The values afterwards look different to me.**
This sentence deals with the phase parameter of the annual cyclicity. The phase did not changed between the 3 time intervals, that is the sinusoidal always peaks in mid

February (see Figure 9c– Figure 8c in the revised manuscript).

**Change:** We specified this by replacing the term by *timing of the annual cycle* by *phase of the annual cycle*.

**Tom Pering,** *Specific issues, Line 490***: Significance of the trend?**
The significance can be derived directly from the reported standard errors.
**Change:** We added the standard errors for the estimated amplitude.

**Tom Pering,** *Specific issues, Line 493***: How did you determine outliers?**
For this qualitative statement: by eye. This statement was just meant to give an overview of the data and its variations. Please note, that we did not treat these "outliers" differently than other data points in the analysis.

**Tom Pering,** *Specific issues, Line 556***: Rephrase to remove 'basically vanished'.**
We consider "basically vanished" a matching qualitative description of a correlation coefficient of +0.19, in particular in contrast to the "former" correlation coefficient of +0.69. The sentence reads: *This correlation was lower for the calibrated data (correlation coefficient of +0.69 when all wind speeds are considered) and in particular basically vanished for wind speeds larger than 10 m/s (correlation coefficient of +0.19, Figure 8f).*
Remark: We are aware of the fact that we call elsewhere a correlation coefficient of 0.25 "weak" — i.e. there is some arbitrariness in qualitative scales, what we consider nevertheless appropriate.

**Tom Pering,** *Specific issues, Line 553-558***: See general comments on correlation.**
See our reply on (Treatment of statistics, Part 1).
**Tom Pering, *Specific issues, Line 580*: see previous comment, anti-correlation needs proof.**

We confirm that this refers to a negative Pearson correlation coefficient.

**Tom Pering, *Specific issues, Line 718*: What does 'basically not correlated' mean in quantifiable terms?**

**Change:** We simplifed the sentence by removing *basically*.

**Tom Pering, *Specific issues, Line 715 onwards*: Correlation terminology, see general comments.**

See our reply on (Treatment of statistics, Part 1).

**Tom Pering, *Specific issues, Line 787*: Reference should probably be made to the Aiuppa paper here (already cited in this manuscript), which talks about this subject exactly.**

As stated in the abstract, we focussed on the retrieval method and description of the time series rather than volcanological interpretations. Nevertheless, we agree with the reviewer that a somewhat deeper comparison with the volcanological model presented by Aiuppa et al. (2018) is appropriate. We therefore extended our discussion on volcanological findings.

**Change:** We extended and reformulated the paragraphs on volcanology in the discussion which reads now (Lines 800–837 in the revised manuscript):

*BrO/SO$_2$ and SO$_2$ emission fluxes and magmatic processes*

[revised manuscript text omitted]

**Tom Pering, *Specific issues, Line 787-806*: Interesting analysis, but framing of this will depend on the reassessment of statistics in the manuscript.** We don't consider a reassessment of the statistical methods/results required. Thus no change of the framing is required.

**3 Technical Comments**

**Tom Pering, *Technical Comments, Line 10*: Correct 'We make plausible'. Sentence needs shortening and clarification.**
**Change:** We removed the corresponding sentence from the abstract (along with other shortenings of the abstract).

**Tom Pering, *Technical Comments, Line 15-18*: Shorten, too much distance between the mention of 'former periods' and what those periods are.**
We consider the original sentences as a rather condensed form of the specific content. We don't see how the text regarding this content can be further shortened. Nevertheless, we moved these sentences to the first paragraph of the abstract because this is where the abstract deals the first time with the time interval(s).
**Change:** The sentences were moved to the first paragraph of the abstract.

**Tom Pering, *Technical Comments, Line 253 and 293 and 531 and 744*: Don't use 'w.r.t' use with regards to.**
**Change:** We replaced *w.r.t.* by *with respect to* at all 10 positions in the manuscript.

**Tom Pering, *Technical Comments, Line 295*: 'There is also a significant number of scans'**
**Change:** Adjusted as recommended.

**Tom Pering, *Technical Comments, Line 299*: 'Gaussian distribution' singular.**
**Change:** Adjusted as recommended.

**Tom Pering, *Technical Comments, Line 571-572*: Rephrase needed.**
**Change:** We changed the text to read now: *[The larger the wind speed, the higher*

*is the atmospheric turbulence and thus the lower is the accumulation.] Accordingly, over-proportionally much volcanic gas could effectively get released from the volcanic edifice to the atmosphere during peaks in the wind speed (if the wind speed is subject to significant short-term fluctuations).*

**Tom Pering,** *Technical Comments, Line 577*: **There are a number of possibilities.**
**Change:** Adjusted as recommended.

**Tom Pering,** *Technical Comments, Line 718*: **Remove 'basically'.**
**Change:** Adjusted as recommended.

**Tom Pering,** *Technical Comments, Line 741*: **Alter phrasing away from 'basically vanishing'**
**Change:** We changed the text to read: *The correlation became insignificant (+0.16) when the wind speeds are calibrated and only wind speeds larger than 10 m/s are considered (see Figure 7e+f).*

**Tom Pering,** *Technical Comments, all graphs*: **check the 2 is subscripted in $SO_2$.**
**Change:** We took care of this issue.

**4 Additional Comment by the authors**

There has been a major update from GNU R 3.6 to GNU R 4.0. Besides many other changes, GNU R 4.0 contains now an improved algorithm for rounding to decimals (see a description of the issue on https://cran.r-project.org/web/packages/round/vignettes/ Rounding.html).

The statistical analysis in our manuscript was performed with GNU R 3.6 for the originally submitted manuscript but with GNU R 4.0 for the current version of the manuscript. As expected, most numerical results are identical although whenever there were numerical thresholds applied as data filter, some data points are now no more rejected by the filter while other data points are now no more rejected. In consequence, some data points appears now (or are missing now, respectively) in the plots (and analyses) of the time series. Nevertheless, these minor numerical changes did not changed any of our major findings.

You may see this behaviour prominently in Figure 13 which looks as in the original manuscript except that there is now a "blue" correlation coefficient of $-0.2004209$ for pressure vs. $H_2O$ which was originally (absolutely) smaller than $|0.2|$ and thus was "grey".

---

## Author Comment (AC2) · 21 Mar 2021

We thank Christoph Kern for his comprehensive and constructive review of our manuscript "SO$_2$ and BrO emissions of Masaya volcano from 2014–2020". In the following, we reply on his specific comments paragraph-wise. If not stated differently, the line numbers and figure numbers refer to the originally submitted manuscript.

**1   Summary and General Comments**

**Christoph Kern,** *Summary***:**

This manuscript discusses Differential Optical Absorption Spectroscopy (DOAS) measurements of sulfur dioxide (SO2) and bromine monoxide (BrO) performed at Masaya volcano, Nicaragua. The measurements stem from two monitoring stations of the Network for Observation of Volcanic and Atmospheric Change (NOVAC). The scanning DOAS instruments at these stations scan the sky from one horizon to the other while collecting scattered solar radiation. When this radiation intersects the volcanic plume, partial absorption occurs at ultraviolet (UV) wavelengths specific to the gas species — in this case SO2 and BrO.

A large part of the manuscript deals with the development of a methodology for consistent analysis of the continuous DOAS data which spans the 2014-2020 time period with only two short interruptions. The authors emphasize the need for applying data quality filters to remove unreliable DOAS scans from the analysis and introduce several corrections for what they deem to be systematic errors in modeled meteorology and measurement geometry parameters.

The resulting time series of SO2 emission rate and BrO/SO2 molar ratio are presented in Figure 9 and appear to show that SO2 emissions averaged around 1000 metric tons per day (t/d) in 2014 - May 2018, then dropped slightly to around 700 t/d. The BrO/SO2 ratio had a significant annual periodicity with relative maxima occurring in March of each year, as well as a general increase in the ratio occurring around the time of the appearance of a lava lake in late 2015. Finally, the implications of these observations are discussed (1) for atmospheric chemistry occurring in the volcanic plume and (2) for volcanic processes occurring at Masaya during the observation period. The authors hypothesize that the seasonal trend of BrO/SO2 may stem from a dilution of bromide in the aerosol phase during more humid times of the year slowing the formation of BrO, and that the drop in SO2 emissions in 2018 was caused by an overall drop in lava lake activity around this time.

**Christoph Kern,** *General Comments* **(Part 1):**

**This manuscript provides a wealth of information on methodology for analyzing scanning DOAS measurements of volcanic gas plumes. The authors discuss two different meteorology models (ECMWF ERA-Interim and Operational ECMWF Reanalysis) and compare these with each other and data from a ground-based meteorology station in Managua. Various methods for filtering and averaging are considered, and finally a correction is developed and applied to the ERA-Interim data. Next, the authors assess methods for retrieving SO2 column densities from the spectral data, developing filters for rejecting data of questionable quality, testing spectroscopic retrievals in multiple wavelength regions, and exploring four different approaches for correcting and/or filtering data associated with contaminated clear-sky reference spectra. The calculation of SO2 emission rates from column densities follows standard methods, but additional filters are then introduced to remove data collected in unstable wind conditions from further consideration. Further testing the reliability of their retrievals, the authors check for and find a significant correlation of SO2 emission rate with wind speed and develop an empirical correction for plume height which appears to improve the quality of their results. Finally, an operational method for retrieving BrO/SO2 ratios representative for each DOAS scan is developed. In the discussion section, all of these topics are again brought up, this time in the context of a critical assessment of each retrieval step and a detailed discussion of potential error sources. All of these topics are well-motivated, described in detail, and appear to be robust, therefore providing a valuable resource for researchers analyzing scanning DOAS data.**
We are thankful that this is appreciated.

**Christoph Kern, *General Comments* (Part 2):**
**At the same time, the other aspects of this manuscript are lacking detail in my opinion, especially the purported link between the measurement results and volcanic processes. Cited extensively throughout this manuscript, a study by**

**Aiuppa et al (2018) developed a fairly detailed conceptual model of degassing at Masaya specifically for the time period examined here, yet there is almost no mention of their conclusions (other than to say that the re-analysis performed here yielded overall higher SO2 emission rates). The authors should consider a much more thorough discussion of this existing degassing model, noting where new information might be added based on their measurements, and work out the volcanological implications of their results compared to those presented in this previous work.**

We understand the reviewer's motivation, which leads him to suggest to undertake a more comprehensive discussion of the degassing model by Aiuppa et al. (2018). However, the emphasis of this manuscript is on data reporting and it is not easy to say to which extent new information about the model can be gained from our data. Therefore and considering the fact that our manuscript is already quite lengthy, we decided not to attempt a comprehensive volcanological interpretation of the time series from this study. The short paragraphs on volcanology should be rather considered as a first outlook on a possible future use of the reported time series for volcanological studies. Nevertheless, we agree that Aiuppa et al. (2018) presented the state of the art of the knowledge on the ongoing Masaya activity cycle and thus we adjusted our discussion on volcanology in order to compare it with their model.

**Change:** We extended and reformulated the paragraphs on volcanology in the discussion which reads now (Lines 800–837 in the revised manuscript):

*$BrO/SO_2$ and $SO_2$ emission fluxes and magmatic processes*

[revised manuscript text omitted]

Furthermore, we removed most of the volcanological findings from the abstract in order to highlight that these are not considered as the strongest conclusions of our study.

**Christoph Kern, *General Comments* (Part 3):**
**Overall, I wonder whether Atmospheric Chemistry and Physics is the best venue for dissemination of this manuscript. In its current form, readers primarily interested in the reactive halogen chemistry or volcanic degassing processes will likely find it difficult to identify the information relevant to them. Sections**

**dealing with development, testing, and refinement of measurement strategies make up approximately 2/3 of the manuscript, while the presentation and discussion of results is fairly minor in comparison. The manuscript is already quite lengthy, so this situation might be mitigated by moving significant portions of the methodology to a supplement where those interested in these aspects could find all the details, while expanding on the sections dealing with atmospheric processes and especially volcanology. However, another option might be to move this manuscript to a technical journal such as Atmospheric Measurement Techniques. In this case, the technical information could remain in the body of the manuscript, and less details would be expected in the discussion of atmospheric chemistry and volcanic processes.**

While we understand the reviewers rational about the choice of the journal, we are still convinced that ACP is a good choice as we want to detail in the following.

1) We consider the reporting of the $SO_2$ and BrO time series as the core information of our study (that is why we decided to submit to ACP rather than to, e.g., AMT). In addition, we compiled and compared the meteorological data and compared the gas data with the meteorological data in order to test hypotheses on several chemical and physical processes, which may or may not alter the bromine chemistry in the volcanic plumes. We are convinced that ACP is an appropriate journal for these data and that the analyses (alone) are sufficient for a publication in ACP.

2) During our study, we developed the new retrieval tools discussed in this manuscript. The description and discussion of these tools constitutes now about 1/2 to 2/3 of the manuscript. In our opinion, such an amount of information should not be attached as an appendix. We considered to split the study in two manuscripts, one on the tools and one on the data, although both parts would be highly entangled (the tool manuscript would be validated with the data and the data paper needs to partially reiterate the tools in detail) and a large amount of redundancy would be unavoidable. In view of the vast number of publications per year, we decided that a single, though long manuscript would be more convenient for the reader than two (in total longer) manuscripts.

3) Finally, we highly value "open access" journals (which limits the number of journals which come into question), in particular because a large fraction of the likely readers reside in developing countries, where access to literature behind paywalls is limited. Moreover, open access literature somewhat loosens the past imperative to publish in a specific set of journals in order to reach a specific audience. As far as we know, ACP is regularly read by other remote-sensing gas volcanologists and thus a good journal for our study.

**Change:** We moved two further paragraphs to the appendix ("Ground-based data from Managua airport" and "Choice of the wavelength range in the $SO_2$ DOAS fit").

**Christoph Kern, *General Comments* (Part 4):**
**Regardless of the path chosen by the authors, I am highly supportive of the content being published for use by the scientific community. In the following section and an attached annotated PDF, I list specific comments which I hope will help the authors in making improvements to their manuscript.**

We like to thank Christoph Kern for this general support to publish our manuscript. We also highly welcome his many minor corrections and suggestions found in his attached annotated PDF. We applied most of them to our manuscript.

**Change:** See our reply on the *Minor corrections* at the end of this document.

**2   Specific Issues**

**Christoph Kern, *Specific Issues* (Part 1):**
**Abstract - Pending the decision on how to proceed with the publication of this manuscript, the abstract should be revised and shortened to highlight only the key aspects of the study.**
**Change:** We shortened the abstract by about 25%. In particular, we removed most of the qualitative volcanological interpretations because these have not been the major conclusions of this study.

**Christoph Kern, *Specific Issues* (Part 2):**
**L199 — Please clarify how the quality filter based on spectral intensity works. Does the filter only consider the SO2 fit region when rejecting over- or under-exposed spectra? Or does it consider the entire spectrum? It would seem to me that spectra could still be used if they are not overexposed in the fit region. Also, I'd be concerned that removing individual spectra from a scan based on their intensity might lead to preferential removal of plume spectra. In my experience, plume spectra often appear brighter than background sky spectra during clear-sky conditions due to the increased scattering of solar radiation on aerosols in the plume. This will often cause the plume to appear brighter than the sky. The result of this filter could therefore be a removal of some or all the spectra associated with the plume itself. Of course spectra that are oversaturated in the fit region cannot be evaluated, but in such a case the entire scan would need to be discarded, not just the oversaturated plume spectra.**
We are thankful for this comment. The overexposure and underexposure filters were applied on the total spectrum (as we wrote in the manuscript: *any channel*). The underexposure filter was, however, wrongly described in Table 2 (but correctly in the main text). We corrected for that in the revised manuscript.
Discard single spectra vs. discard the whole scan: While it is true that discarded

spectra may sometimes be associated with the plume, we also observed frequently discarded spectra outside of the "plume region". A major reason for over- and under-exposure is presumably the change in the cloud cover and thus independent from the position of the plume. Discarding a whole scan only because a single spectrum is under- or overexposured would in our opinion unnecessarily reduce the data coverage. As stated in Table 2, we nevertheless discard scans which encompass less than 30 of the effective 43 scan spectra (51 initial scan spectra minus the 8 spectra with zenith angles beyond $\pm 76°$ which we unconditionally discarded anyway. The choice of 30 was to some extent arbitrary (though it should be significantly larger than 20, see text) and could be adjusted to the meteorological conditions at a specific volcano.

**Change:** We corrected Table 2 for the underexposure filter condition. Furthermore. we added the following remark in Line 213–221 in the revised manuscript:

*A remark on the overexposure filter: It was chosen as described above in order to assure that BrO DOAS fits were not degraded by saturation effects. For the sole retrieval of the $SO_2$ emission fluxes, it may be sufficient to check for overexposure exclusively in the $SO_2$ fit range. Nevertheless, we aimed for the same data base for both, the $BrO/SO_2$ molar ratios and the $SO_2$ emission fluxes in order to assure a consistent comparison of both time series. Further arguments for applying the overexposure filter on the overall spectrum are: (1) Overexposure in the spectrum indicates significant variations in the intensity of the back-scattered solar radiations during a scan (caused presumably by variations in the cloudiness of the sky). Accordingly, the overexposure filter would (conveniently?!) reject those times with unstable meteorological conditions. (2) The saturation of any pixel of the charge-coupled device detector may cause the additional photo-electrons to spill over to other pixels and thus could lead to the degradation of the entire spectrum.*

**Christoph Kern, *Specific Issues* (Part 3):**
**L380 — It didn't become clear to me whether the triangulated plume height**

**was used for the retrieval of SO2 emission rates in cases where it could be determined, i.e. when both scanners detected the plume. Regardless of what was done, one interesting test would be to restrict the dataset to only those emission rates derived using triangulated plume heights. For this subset, does the correlation of emission rate with wind speed disappear? If this correlation really is an artefact of assuming incorrect plume heights, wouldn't we expect it to disappear in cases where the plume height can be measured reliably?**

As the reviewer expected, a smaller correlation coefficient between the $SO_2$ emission fluxes and the wind speed can be observed when using the triangulation results instead of the parametrised plume heights.

We aimed to provide a time series which is consistently derived in the same way for all years. Therefore we used the triangulation results as well as the ECMWF re-analysis data exclusively for the calibration of the meteorological data.

**Change:** We added Figure D1 to the appendix and we added in the revised manuscript to the paragraph on *Correlation of SO₂ emission fluxes and wind speeds* (Line 390 ff.):
*For March–October 2014, the SO₂ emission fluxes can be calculated alternatively via the triangulation results, i.e. using the triangulated plume height and plume propagation direction instead of the parametrised plume height and the wind direction from ECMWF. We calculated the SO₂ emission fluxes accordingly, while using only data with triangulated plume altitudes below 1200 m a.s.l. in order to be consistent with the above explained parametrisation approach (and again in order to avoid the influence of the artificial "wings"). For these alternative SO₂ emission flux estimates, the correlation between the SO₂ emission fluxes and the wind speeds were significantly lower and completely vanished for wind speeds larger than 10 m/s (correlation coefficient of +0.05 and +0.02 for the two NOVAC stations, see Figure D1).*

**Christoph Kern, *Specific Issues* (Part 4a):**
**L455 and throughout — I had a hard time understanding the descriptions of**

**volcanic activity presented by the authors. Descriptions such as 'elevation of the lava lake' seem a bit misleading. Aiuppa et al (2018) reported that there was not a lava lake at Masaya prior to December 2015. So do you mean 'appearance of a lava lake' here? I recommend that, when describing volcanic processes, the nomenclature from Aiuppa et al (2018) and other previous studies be adapted as much as possible and those studies be referenced so that it's clear you are referring to the same events.**

We consider it a matter of volcanological debate whether the lava lake has formed ("formation", the description by Aiuppa et al., 2018) or whether the lava lake has been there already for quite some time but with a lave lake level too deep to see it from the rim or via satellite observations. More specifically, there is to our knowledge no observational evidence which interpretation is correct (but consider the elevation hypothesis more plausible). Nevertheless, and as we decided to focus this study not on volcanology, we agree that we should adapt the nomenclature used in the literature, not because we necessarily agree with the nomenclature but for the sake of unambiguity.

**Change:** We changed all parts in the manuscript which refer to the change of the lava lake in December 2015 such that is now reads "lava lake appearance" or like-wise.

**Christoph Kern, *Specific Issues* (Part 4b):**
**Here, I also don't understand what is meant by 'actual onset of activity in the shallow magma system'. What type of activity? What do you consider the shallow magma system? Do you mean the shallow magma reservoir identified by Rymer et al. (1998) and Williams-Jones et al. (2003)? See Aiuppa et al (2018) for more details on this.**

A discussion on the volcanological system can be found in the vary last part of the revised manuscript.

**Change:** We reformulated this paragraph which in the revised manuscript: *"In this section, we present the SO$_2$ and BrO time series retrieved from the NOVAC data.*

*There were two major data gaps in the $NOVAC$ time series from September 9 to November 16 2015 and from March 21 to June 23 2018. The statistical analysis results discussed in the following, therefore, refer to the time intervals (1) March 2014 − September 2015, (2) November 2015 − March 2018, (3) June 2018 − March 2020.*

*This separation in three time series is also in good agreement with the three episodes of general volcanological observations of the lava lake activity: (1) "prior to the lava lake formation (until late 2015)", (2) "period of high lava lake activity" (from November 2015 to October 2018, where the thermal activity started at latest on November 15 and the lava lake visualised on December 15, INETER, 2015a, b, Aiuppa et al., 2018), and (3) "period of low lava lake activity (from October 2018 on)" (Smithsonian Institution, 2018). It has been reported that Masaya was already relatively calm before May 2018 (Smithsonian Institution, 2018), indicating that the major transition from high to low activity may have happened somewhen during the data gap from March–June 2018. The minor discrepancy in separation with respect to the time span from June–October 2018 was not elaborated in the following."*

**Christoph Kern, *Specific Issues* (Part 5):**
**Table 5 — In this section of the manuscript and the accompanying table, a number of suggestions are made for further improving upon the framework utilized by the authors to evaluate the scanning DOAS data. It's understood that there is always room for improvement, but since one of the central points of this study is to discuss the ideal methodology for analyzing the DOAS data, it seems a bit contradictory to make so many suggestions for improvement beyond what is recommended here. And in many cases, particularly in the table, the suggestions either seem to contradict earlier statements or are too vague for the readers to act upon. For example, I understood that the I0 correction did not improve the SO2 retrieval noticeably. Why then is it recommended that this should always be applied? Similarly, suggestions like 'further optimize filters',**

**'apply more filters', 'optimize triangulation algorithm', or 'improve calibration' don't provide actionable information. I recommend that this section either be carefully revised such that only specific, actionable suggestions are given, or that this section be removed entirely and these suggestions for possible future investigation simply be given in the methods section.**
We stated those options for further improvements and also in order to clarify what we did and what we did not, i.e. to be transparent on possible flaws in our analysis (however, we also motivated in the text that these flaws presumably don't alter our main findings). We consider them valuable information in order to assess the performance of the retrieval algorithm. Furthermore, we don't see any major drawbacks in keeping them there.

**Christoph Kern,** *Specific Issues* **(Part 6):**
**L782 — The lack of correlation between BrO/SO2 and background ozone concentrations or wind speed is interesting. Others (myself included) have suggested that in-mixing of atmospheric oxidants could be a relevant process limiting BrO formation. For example, this might explain the spatial heterogeneity of BrO/SO2 we observed at Mount Pagan volcano (Kern and Lyons 2018). Here, you write that 'The observed correlation coefficients between the BrO/SO2 molar ratios and the ozone mixing ratio and the wind speeds were both rather small, indicating that the BrO conversion is not predominantly controlled by the background ozone mixing ratio or the air in-mixing rate.' However, I wonder if in-mixing might be more important than it may seem here. As you point out, the in-mixing rate would likely depend on wind speed, with higher wind speeds leading to greater turbulence and more efficient mixing. But at the same time, a higher wind speed would also reduce the plume transport time from the vent to the DOAS scanning plane, so the measurements would be made in a younger plume. The measurements suggest that the BrO/SO2 ratio is approximately equal, regardless of wind speed. Is it therefore possible that the plume is**

**indeed evolving more quickly (with faster halogen activation) in the high wind / efficient mixing scenario, and that this is the reason that the BrO/SO2 ratio is approximately the same despite the plume being younger?**
Our statement was exclusively based on the empirical correlation results. Using the phrase "not predominantly", we aimed to highlight that no clear causal link from an ozone related process to $BrO/SO_2$ molar ratios could be retrieved from this particular time series. This should not be confused with the possible hypothesis that "ozone related processes could be rejected as a major parameter in the BrO conversion in the gas plume of Masaya". Our statement just highlights that there appears to be other, more important parameters involved at this time and location; in particular the processes linked to the absolute humidity (at least according to the correlation analysis).

**Christoph Kern, *Specific Issues* (Part 7):**
**L786ff — This section needs major revisions in my opinion.**
First of all, we would like to highlight that this part of our manuscript is rather independent from any previous or subsequent part of the manuscript.
Our discussion in the original manuscript on volcanological findings was aimed at providing some possible interpretations on magmatic processes without claim of completeness nor rigour. The reviewer would like to see a more critical assessment of our data in context of the existing literature. While we still would prefer this manuscript to focus not too much on volcanology, we agree that some more rigour is appropriate. Therefore, we entirely reworked our discussion on volcanology, linked it more closely to the literature and removed hints on any interpretation which is not rigourously backed by our data. In the following, we address the reviewer's individual comments by the specific text parts we added to that section.
**Change:** Please find the reworked discussion in the revised manuscript in Lines 800–837 and paragraph-wise in our replies on the following reviewer comments.

**Christoph Kern, *Specific Issues* (Part 7a):**
**As I stated in the general comments above, I strongly recommend the authors take a closer look at the literature available on Masaya's volcanic system and activity.**
We summarised the literature on Masaya's volcanic system and activity as following (in addition to the information given in the introduction section):
*Aiuppa et al. (2018) suggested a model, based on their data and past studies, that the (re)appearance of the lava lake on the surface was most likely caused by the enhanced magma convection supplying $CO_2$-rich gas bubbles from minimum equivalent depths of 0.36–1.4 km. They proposed that this elevated gas bubble supply destabilized the shallow Masaya magma reservoir (<1 km). The model is not completely new, already Rymer et al. (1998) and Williams-Jones et al. (2003) proposed that the Masaya cyclic degassing crises are caused by convective replacement of dense, degassed magma by gas-rich vesicular magma in the shallow (<1 km depth) plumbing system. Their ideas were based on results of periodic gravity surveys, and they also argued such convective overturning is not necessarily triggered by intrusion of fresh (gas-rich) magma but may simply be initiated by degassing/crystallisation (and consequent sinking) of shallow resident magma. The data from Aiuppa et al. (2018) seem to confirm this model.*
**Change:** We added this paragraph to the revised manuscript (Line 801–809).

**Christoph Kern, *Specific Issues* (Part 7b):**
**The very first sentence in this section claims that the elevation (=appearance?) of the lava lake was likely caused by the arrival of juvenile magma in the shallow system. How do you know? I don't see how this follows from the measurements presented in the study — in fact I think you could argue the opposite could be true given the observation that the SO2 emission rate hardly increased in this period. At the same time, Aiuppa et al (2018) make some relatively convincing arguments for a shallow intrusion which may have caused a rejuvenation of**

**pre-existing shallow magma. If this is what you are arguing, then please be sure to cite their paper as well as the original references upon which their observations rely. When citing previous work, please also be sure to use the original nomenclature when describing parts of the volcano's plumbing system or types of activity. For example, please clarify what is meant by 'shallow system'.**

We agree that our proposed hypothesis was not backed sufficiently by our data. Concerning the reviewer's question mark on "elevation (=appearance?)", this is indeed meant (and now reformulated) this way, see our reply on "Specific Issues (Part 4a)".

**Change:** We removed that hypothesis from the manuscript and linked our discussion closely to the framework provided by Aiuppa et al. (2018).

**Christoph Kern, *Specific Issues* (Part 7c):**
**Other concerns I have for this section include:**
**- You distinguish between juvenile magma and older magma. Where is the juvenile, where is the older magma? Are they moving up or down? Mixing? Convecting?**

The nomenclature was over-simplified and not comprehensive in the original manuscript.

**Change:** We adopted the nomenclature by Aiuppa et al. (2018).

**Christoph Kern, *Specific Issues* (Part 7c):**
**- It's not clear that bromine and sulfur would degas at the same depth/pressure. Therefore, it's not clear to me that bromine would have degassed earlier than sulfur from old magma. Isn't it possible that sulfur (and additional brome) are being supplied from juvenile magma and being added to the gas emitted from the older magma?**

In the original manuscript, the interpretation of the linear trends was done assuming that there was a one-time arrival of juvenile magma. Our criticised interpretation was

addressing the increasing trend in the BrO/SO$_2$ molar ratios *after* the one-time arrival of the juvenile magma. Accordingly, we proposed the interpretation that bromine was degassing predominantly later than sulphur.
**Change:** This not rigourously backed interpretation was removed from the revised manuscript.

**Christoph Kern, *Specific Issues* (Part 7d):**
**- Rather than speaking of earlier or later degassing, maybe it's best to describe things in terms of pressure or depth?**
We agree.
**Change:** We adopted this perspective in the revised manuscript.

**Christoph Kern, *Specific Issues* (Part 7d):**
**- What was the observed 'decrease in lava lake activity in mid 2018'? What exactly decreased? How was this determined? I assume this observation is also taken from other studies? Please be sure to cite them.**
We cited in the introduction section: *Masaya's most recent lava lake cycle started in late 2015 when a lava lake appeared (incandescence observed since November 2015, INETER, 2015a, b; Aiuppa et al., 2018) and started to cease in October 2018 when Masaya's thermal activity decreased to relatively low levels (Smithsonian Institution, 2018).*
We are not aware of further publications of this decrease in thermal activity.

**Christoph Kern, *Specific Issues* (Part 7e):**
**- Couldn't the decrease in the SO2 emissions simply be caused by depletion of sulfur in the magma? It's not clear to me why it requires a change in pressure or temperature at degassing depth.**
Our reasoning was motivated by the assumption that a depletion in SO$_2$ should manifest in a continuously decreasing degassing behaviour rather than in a step decrease.

In contrast, a step increase in the degassing behaviour could be the consequence of a fast change in the physico-chemical conditions of the magma.

Nevertheless, we considered it more appropriate to remove our incomplete discussion from the text — or rather — we replaced it by *"But a step increase in the BrO/SO$_2$ molar ratios can be noted after September 2015 (happening somewhen between September–November 2015, covered by a data gap). This change in the gas composition was thus caused by variations in the volcanic bromine emissions rather than in the sulphur emissions, similar to the change in CO$_2$/SO$_2$ molar ratios noted by Aiuppa et al. (2018), which respectively was caused mainly by the variation of the CO$_2$ emission flux. Those authors interpret these observations as evidence for supply of CO$_2$-rich gas bubbles, sourced by enhanced magma transport and degassing at a depth $>$(0.36–1.4) km. Following their interpretation and assuming that BrO is somehow an indicator for bromine emissions, that would mean that also bromine is degassing below that depth or something, which leads to an enhanced transformation of HBr into BrO."*

**Change:** We added this paragraph to the revised manuscript (Line 813–820).

**Christoph Kern, *Specific Issues* (Part 7e):**
**- I don't think the lack of change in BrO/SO2 in 2018 implies that Br and S partitioning are independent of the physio-chemical conditions in the magma. I guess it's also possible that changing conditions combined with varying degrees of depletion of these elements in the magma could lead to a relatively stable ratio. (such an example is actually given in the next sentence).**

The original text reads: *The decrease of the SO$_2$ emission fluxes indicated a major change in the physico-chemical conditions of the magma [..] The [in the meanwhile] about constant BrO/SO$_2$ molar ratios would then imply that the bromine and sulphur partitioning [..] were independent on the physico-chemical conditions in the magma.*

Besides the given interpretation for the about constant BrO/SO$_2$ molar ratios, there are two further possibilities: Second, another physico-chemical process, which does

not significantly impact the $SO_2$ degassing but the bromine degassing, change as well. Or third, the degassing regime change entirely by the arrival of a new batch of magma. While the third possibility can not be excluded, there are also no supporting observations for this interpretation. A similar reasoning holds for the second possibility. Nevertheless, we considered it more appropriate to remove our incomplete discussion from the text.
**Change:** We remove our incomplete discussion from the text.

**Christoph Kern, *Specific Issues* (Part 7e):**
**Overall, I think this section would benefit greatly if the discussion occurred within the framework of a conceptual model for the degassing behavior. Aiuppa et al. (2018) actually present such a model, and this model could be used if desired. In that case, it would be interesting to highlight which information this study adds to the existing framework, which (if any) new observations do not fit well into the exiting model, and which new insights (beyond those already discussed by others previously) can be gleamed from these measurements.**
We did this to some extent as detailed above.
In addition to the comparison with the literature already given above, we added:
*Aiuppa et al. (2018) further observed an increase in the SO2 degassing after the appearance of the lava lake at the surface, which is a further argument on their hypothesis for a faster shallow magma convection. Our data confirms an enhancement of the mean SO2 emission fluxes by 30% for the period from December 2015 to February 2016 when compared with the previous and subsequent degassing behaviour. The described observation of Aiuppa et al. (2018) ends with March 2017. The decrease in the lava lake activity in mid 2018 is therefore not described by those authors. We here report a significant decrease in the SO2 emission fluxes after June 2018 (happening somewhen between March?June 2018, covered by a data gap), while the BrO/SO$_2$ molar ratios hardly changed. This decrease of the SO$_2$ emission fluxes in time in connection with the decrease in the lava lake activity is consistent to the interpretation*

*that the convection of the magma inside the conduit below the upper reservoir has slowed down again after 2018 and an important further indicator to sustain this hypothesis could be additional $CO_2/SO_2$ molar ratios. Unfortunately no $CO_2/SO_2$ molar ratios are available to the authors by the time of writing of the manuscript.*

*An unchanged $BrO/SO_2$ ratio and a lower $SO_2$ emission flux would lead to lower bromine emission as well, if we assume a correlation of bromine emissions and amount of BrO. We might further speculate that the bromine emission and carbon emission are characterised again by a similar pattern, which would mean that we also see a decrease in the $CO_2$ emission flux.*

**Change:** We added this paragraph to the revised manuscript (Line 825–837).

**3  Minor Corrections**

**Christoph Kern,** *Minor Corrections***:**
**Please see the attached annotated PDF for additional comments, suggestions,
and minor corrections to the text. I'd like to thank the authors and journal editors
for the opportunity to review this manuscript.**
**Please also note the supplement to this comment: https://acp.copernicus.org/
preprints/acp-2020-942/acp-2020-942-RC2- supplement.pdf**
We thank Christoph Kern for this exhaustive list of suggested reformulations, requested
specifications, or raised questions.
**Change:** We adjusted the manuscript according to the larger part of the reviewers
minor corrections. In the following are only those reviewer comments listed where we
didn't fully apply the proposed changes.

**Line 51:** We consider this a remark rather than a change request.
**Line 54:** While the reviewer asks "why not?", the other review report took the diametrically different position and requested to change from our originally "may not" to "are
not". We now wrote "are usually not".
**Line 61 (first):** The reviewer suggests to remove that apparently redundant sentence.
We think explicitly stating this sentence may be of use to understand our reasoning.
**Line 96:** The reviewer suggests to introduce sub-heading in the introduction section.
We don't think that this is common practise but would do so if the editor insists.
**Line 106:** The reviewer suggests to replace "strongest degassing volcanoes" by "most
prodigious degassing volcanoes". We would like to keep our original adjective.
**Figure 2:** The direction of the arrow is correctly pointing with wind direction
**Line 150:** The reviewer suggests a shorter description of the wind data in order
to focus on the main takeaways. We consider our original, more comprehensive
description of potential use for the reader.
**Line 199:** The reviewer asks: **"Do the filters consider the SO2 SCD of spectra**

**adjacent to ones that have been removed from a scan? If not, I'd be concerned
that significant parts of the plume may often be filtered out from scans. Scat-
tering on aerosols in the plume will often make it appear brighter than the
background. This can lead to oversaturated spectra. If you just remove these
but keep the rest of the scan for your analysis, this might lead to a systematic
underestimation of SO2 in the scans unless you require that the SCD is low in
the region where spectra are being removed."**

Our reply: See our reply on Specific Issues (Part 2). Specifically, we agree that such
an effect could be possible and incorporating such an additional filter criterion may
further improve the retrieval. Nevertheless, we argue that this effect is rather not
significant for the reported time series for several reasons:

(1) We checked the efficiency of this filter and usually there are only few spectra
rejected and even if there are many spectra rejected, these are not necessarily located
in the plume region (see Figures 4 and D3 in the revised manuscript).

(2) On $SO_2$ emission fluxes: our Gaussian filter rejects those scans where the plume
centre is missing.

(3) On $BrO/SO_2$ molar ratios: On the one hand, if there is no spatial variation of the
$BrO/SO_2$ molar ratios along the plume cross section, the addressed filter would not
cause a systematic misestimation of the $BrO/SO_2$ molar ratio (though a decreased
precision). On the other hand, if there is a spatial variation of the $BrO/SO_2$ molar
ratios along the plume cross section, the addressed filter would cause a systematic
misestimation (presumably an overestimation) of the $BrO/SO_2$. Nevertheless, our
analysis of the plume age rather supports that there is no such spatial variation.

**Table 3 (second):** The $\in$-symbol allows for a maximum condensed format of the
numerical conditions. Even if the symbol may not be used in the most common way,
we think it reports the meaning of those statements in a comprehensive way.

**Line 215:** As the reviewer mentioned himself, Davis and McLaren (2020) actually
discuss a different regime.

**Line 252:** The third approach uses a solar-atlas spectrum, which is definitively not

measured under the same measurement conditions.

**Line 259:** We consider this a remark rather than a change request.

**Line 260:** We consider this a remark rather than a change request.

**Line 308:** The "should be negative by construction" is not meant to be a strict mathematical condition but highlights the fact that a dSCD should be always smaller than the absolute SCD.

**Line 314:** We didn't apply a positive threshold for this filter.

**Line 320+322+325:** We consider stating the statistics of potential use, even if it requires several reads to understand the condensed notation.

**Line 419:** The threshold was chosen that way to assure that the plume region always contains at least 3 spectra.

**Line 436:** The reviewer asks: **"Could you please clarify what you mean with 'background contamination with BrO'? Is this a problem that stems from possible plume spectra being included in the "added-reference-spectra"? But couldn't this be avoided by using the same methods for selection of the reference region as described in the SO2 flux section, i.e. filtering scans in which the reference appeared to be contaminated after e.g. fitting a solar atlas?"**

Our reply: We stated in the original manuscript: *"no reliable method for a absolute calibration of a background contamination with BrO has been developed"*. We added in the revised manuscript *"[... developed] (in contrast to the $SO_2$ retrieval)"*, in order to make this constrain more prominent. Furthermore, we specified in the revised manuscript: *possible [background contamination with BrO]*, whose reasons are understood to be the same as for $SO_2$ discussed further above in the manuscript.

**Figure 9:** We didn't plot the mean $BrO/SO_2$ molar ratios per time interval in order to keep the figure more tidy. The mean values can be easily estimated by eye or looked up in Table 5 (in the revised manuscript).

**Line 518+524:** The reviewer wonders if the comprehensibility of the paragraph could be improved. We think the original text states the details already in a comprehensible way. Unrelated to that request, we have to highlight that there was a typo (1.15 instead

of 0.15), the revised manuscript now reads: *"*$1.11 \pm 0.15$ *for weekly means"*.

**Line 526+533+536+601+673:** We consider "relative factor" the less ambiguous term for this comparison (besides that the reader needs to figure out what is the nominator and what is the denominator). In particular, the proposed terms "discrepancy" or "difference" may refer to an additive behaviour rather than a multiplicative behaviour. Nevertheless, we think it would be indeed beneficial to strictly define the term. Accordingly, we added a bracket in *"We calculated the ratios (called "relative factors" in the following) of the $SO_2$ emission fluxes retrieved by Caracol station divided by the $SO_2$ emission fluxes retrieved by Nancital station using several temporal bin sizes."*.

**Line 546:** We specified from *"(iii) systematic deviations in the spectroscopic retrieval"* (original manuscript, Line 529) under *"(iii) systematic deviations in a possible underestimation of the $SO_2$ dSCD"* (revised manuscript).

**Line 556:** We think "calibrated" is the more appropriate word.

**Line 563:** Our "semi-annual" refers to "twice a year", while the suggested "bi-annual" could ambiguously mean "once every two years".

**Line 605:** "Underestimation by 1.25" refers to "the values are only 80 % of the true values".

**Line 606:** We argued earlier in the manuscript that also our retrieval might slightly underestimated the true value (see Figure 5b). The specification in the brackets highlights this with maximum rigour.

**Table 5 (original manuscript):** We there suggested some possible further improvements of the retrieval. The reviewer requests for more concrete discussion of the suggested improvements, which was not the aim of this outlook.

**Line 670:** But there were two scanners only in 2014. For the other years this "worst case" scenario held indeed.

**Line 684 (second):** We consider the original formulation more appropriate.

**Line 742:** We consider "calibrated" here more appropriate than "corrected".

**Line 764:** This is why we wrote "allegedly": some assumption had to be wrong for those times, e.g. the wind data was wrong or the gas plume was rather old. In both

scenarios, those observations should/could not be used to retrieve information of the plume age.

**Line 785:** The reviewer asks: **"Just for the sake of argument, could it be possible that in-mixing of atmospheric ozone is actually a limiting factor in the formation of BrO? In-mixing of ambient air would be more efficient at high wind speed. This speeds up the chemistry, but also reduces the time that elapses before the plume is measured by the DOAS. On the other hand, in-mixing is less efficient at low wind speed, but more time elapses before the BrO/SO2 ratio is measured. Is it possible that this could explain the lack of significant correlation between wind speed and BrO/SO2?"**

Our reply: We think the in-mixing of atmospheric ozone could always be a limiting factor but for our specific time series, the correlation analysis does not support that this was the case at Masaya. We consider the reviewer's argument for the lack of significant correlation between the wind speed and the BrO/SO2 molar ratios plausible. We added this reasoning in the paragraph on "BrO/SO$_2$ and wind conditions".

**Line 788–806:** These comments were handled when the major comment (Part7) was reworked.

**Line 820:** The reviewer suggested: **"All three of these reasons why this estimate is higher than that of Aiuppa et al. were already presented farther up in the manuscript. I recommend removing them here and instead focusing on any implications that your higher SO2 emission rate might have on the conceptual model for degassing and other activity occurring at Masaya during this period."**

Our reply: We used the conclusion section to summaries the major results of this rather lengthy manuscript in order to assure that the reader will not miss them. The suggested review of the model by Aiuppa et al. (2018) was not one of the foci of this study.

**Figure B1 (original manuscript):** We consider the scaling as a good way to display the differences between the instruments. We agree that this may look somewhat unnecessary for the ambient temperature, we would like to keep it that way for the

sake of a consistent plotting format.

In addition, we applied the request for a quadric plot format for Figures 10 and 12. Nevertheless, we highlight that we did not presented those plots in the original manuscript in a quadric form for the sole reasons that this requirement in space appears to be excessive (while like-wise the plots would be too small if presented only in a single column when the manuscript is formatted in a two-column format.)

**4 Additional Comment by the authors**

There has been a major update from GNU R 3.6 to GNU R 4.0. Besides many other changes, GNU R 4.0 contains now an improved algorithm for rounding to decimals (see a description of the issue on https://cran.r-project.org/web/packages/round/vignettes/Rounding.html).

The statistical analysis in our manuscript was performed with GNU R 3.6 for the originally submitted manuscript but with GNU R 4.0 for the current version of the manuscript. As expected, most numerical results are identical although whenever there were numerical thresholds applied as data filter, some data points are now no more rejected by the filter while other data points are now no more rejected. In consequence, some data points appears now (or are missing now, respectively) in the plots (and analyses) of the time series. Nevertheless, these minor numerical changes did not changed any of our major findings.

You may see this behaviour prominently in Figure 13 which looks as in the original manuscript except that there is now a "blue" correlation coefficient of $-0.2004209$ for pressure vs. $H_2O$ which was originally (absolutely) smaller than $|0.2|$ and thus was "grey".